# Including vegetation dynamics in an atmospheric chemistry-enabled GCM: Linking LPJ-GUESS (v4.0) with EMAC modelling system (v2.53)

Matthew Forrest[1,2], Holger Tost[3], Jos Lelieveld[2,4], and Thomas Hickler[1,5]

[1]Senckenberg Biodiversity and Climate Research Centre (SBiK-F), Frankfurt am Main, Germany
[2]Atmospheric Chemistry Department, Max Planck Institute for Chemistry, Mainz, Germany
[3]Institute for Atmospheric Physics, Johannes Gutenberg University Mainz, Mainz, Germany
[4]Energy, Environment and Water Research Center, The Cyprus Institute, Nicosia, Cyprus
[5]Institute for Physical Geography, Goethe University, Frankfurt am Main, Germany

*Correspondence to:* Matthew Forrest (matthew.forrest@senckenberg.de)

**Abstract.**

Central to the development of Earth System Models (ESMs) has been the coupling of previously separate model types, such as ocean, atmospheric and vegetation models, to address interactive feedbacks between the system components. A modelling framework which combines a detailed representation of these components, including vegetation and other land surface processes, enables the study of land-atmosphere feedbacks under global change.

Here we present the initial steps of coupling LPJ-GUESS, a dynamic global vegetation model, to the atmospheric chemistry enabled atmosphere-ocean general circulation model EMAC. The LPJ-GUESS framework is based on ecosphysiological processes, such as photosynthesis, plant and soil respiration, ecosystem carbon, nitrogen and water cycling and includes a comparatively detailed individual-based representation of resource competition, plant growth and vegetation dynamics as well as fire disturbance. Although not enabled here, the model framework also includes crop and managed-land scheme, a representation of arctic methane and permafrost, and a choice of fire models; and hence represents many important terrestrial biosphere processes and provides a wide range of prognostic trace gas emissions from vegetation, soil and fire.

We evaluated a one-way, on-line coupled model configuration (with climate variable being passed from EMAC to LPJ-GUESS but no return information flow) by conducting simulations at three spatial resolution (T42, T63 and T85). These were compared to an expert derived map of potential natural vegetation and four global gridded data products: tree cover, biomass, canopy height and gross primary productivity (GPP). We also applied a post-hoc land use correction to account for human land use. The simulations give a good description of the global potential natural vegetation distribution although there are some regional discrepancies. In particular, at the lower spatial resolutions, a combination of low cold and low-radiation biases in the growing season of the EMAC climate at high-latitudes causes an underestimation of vegetation extent.

Quantification of the agreement with the gridded datasets using the normalised mean error (NME) averaged over all datasets shows that increasing spatial resolution from T42 to T63 improved the agreement by 10%, and going from T63 to T85 improved agreement by a further 4%. The highest resolution simulation gave NME scores of 0.63, 0.66, 0.84 and 0.53 for tree cover, biomass, canopy height and GPP respectively (after correcting tree cover and biomass for human-caused deforestation which

was not present in the simulations). These scores are just 4% worse on average than an offline LPJ-GUESS simulation using observed climate data and corrected for deforestation by the same method. However, it should be noted that the offline LPJ-GUESS simulation used a higher spatial resolution which makes the evaluation more rigorous, and that excluding GPP from the datasets (which was anomalously better in the EMAC simulations) gave 10% worse agreement for the EMAC simulation than the offline simulation. Gross primary productivity was best simulated by the coupled simulations, and canopy height the worst. Based on this first evaluation, we conclude that the coupled model provides a suitable means to simulate dynamic vegetation processes into EMAC.

## 1 Introduction

Simulation models are at the forefront of Earth systems research. Historically, such models were initially developed to simulate one component of the Earth system in isolation, such as ocean and atmospheric General Circulation Models (GCMs) or Dynamic Global Vegetation Models (DGVMs), with prescribed boundary conditions at the interfaces to other Earth system components. However, the interactions between Earth system components are dynamic, and representations of feedbacks are necessary to assess the functioning and response of the Earth system as a whole. To this end, models have increasingly been coupled to each other to provide dynamic multidirectional fluxes between models, as opposed to prescribing simple non-interacting boundary conditions. This approach has yielded Atmosphere-Ocean General Circulation Models (AOGCMs) which are utilised to understand the dynamics of the physical components of the climate system (Flato et al., 2013).

Interactive carbon cycles and dynamically changing vegetation have been recognised as important processes in the Earth system (Cox et al., 2000; Ciais et al., 2013). Consequently, more recent developments have seen AOGCMs extended to include biogeochemical cycles, most often the carbon cycle, to form a new category of model, Earth System Models (ESMs). These state-of-the-art models are the most comprehensive tools for modelling past and future climate change in which biogeochemical feedbacks play an important role, and for studying biosphere-atmosphere feedbacks explicitly (Flato et al., 2013). However, whilst all ESMs by definition have a carbon cycle, not all have truly dynamic vegetation or a nitrogen cycle. These processes change vegetation cover and structure in response to changing climate and fire activity, and thus models which do not include them miss key biosphere responses and corresponding feedbacks to the climate system (Wramneby et al., 2010).

To take the first steps towards constructing an ESM with dynamic vegetation, anthropogenic influences and fire, we have combined an atmospheric chemistry-enabled AOGCM, EMAC (Jöckel et al., 2010; Pozzer et al., 2011; Jöckel et al., 2016), with a DGVM, LPJ-GUESS (Smith et al., 2001; Sitch et al., 2003; Smith et al., 2014), in a single modelling framework. LPJ-GUESS has been widely applied and extensively evaluated, and has, in different model versions, been extended to include many terrestrial processes and used in over 200 ISI-listed publications[1]. At the time of writing LPJ-GUESS is being actively developed and there are on-going efforts to consolidate many previously independent innovations into the main model release.

---

[1]see *http://iis4.nateko.lu.se/lpj-guess/LPJ-GUESS_bibliography.pdf* for an up-to-date list of publications featuring LPJ-GUESS

This combination of active development, broad range of included processes and the flexible modelling framework design of the LPJ-GUESS source code makes it a good choice to provide the land surface component of an ESM. Furthermore, LPJ-GUESS has already been used in both a global ESM, EC-Earth (Weiss et al., 2014; Alessandri et al., 2017), and a regional ESM, RCA-GUESS (Wramneby et al., 2010; Smith et al., 2011; Zhang et al., 2014). In both modelling systems, sub-daily state variables from the atmospheric component and its land surface scheme are aggregated over one simulation day and provided to LPJ-GUESS (Weiss et al., 2014; Smith et al., 2011), and we follow a similar approach here.

EMAC (ECHAM/MESSy Atmospheric Chemistry)[2] originally combined the ECHAM atmospheric GCM (Roeckner et al., 2006) with the Modular Earth Submodel System (MESSy) (Jöckel et al., 2005) framework and philosophy. The model has since been extended to include expanded representations of atmospheric chemistry, a coupled ocean model with dynamic sea ice (Pozzer et al., 2011), ocean biogeochemistry (Kern, 2013), an alternative base model for the atmospheric circulation (Baumgaertner et al., 2016), regional downscaling via a two-way coupling (Kerkweg et al., 2018) with the COSMO weather forecast model (Baldauf et al., 2011) and a multitude of processes such as representations for aerosols, aerosol-radiation and aerosol-cloud (Tost, 2017) interactions and many more; all of which are integrated via the MESSy infrastructure.

By bringing together these two modelling systems, our intent is to produce a fully-featured ESM which benefits from the continuous development of all components. We plan to follow a step-wise model integration roadmap, whereby the coupling between LPJ-GUESS and EMAC is tightened in well-defined, consecutive steps and processes (such as land use) are included or enabled in a consecutive manner. This will allow us to assess the effects of one model on the other, and the effects of the inclusion of new processes, in a step-wise and logical fashion. For our first step, we have chosen to simulate and evaluate the vegetation produced when LPJ-GUESS is forced by EMAC-simulated climate, ie. a one-way coupling without the feedback from the land surface to the atmosphere.

Upon completion of the full model integration process (including bidirectional coupling which is not presented here), the trace gas emissions from LPJ-GUESS will form key inputs to the atmospheric chemistry representations in EMAC allowing for bi-directional chemical interactions of the surface with the atmosphere. Then the full model should become a powerful tool for investigating land-atmosphere interactions including: the methane cycle and lifetime and the atmospheric chemistry of reduced carbon; fire effects and feedbacks; future nitrogen deposition rates and fertilisation scenarios; ozone damage to plants; and the contribution of biogenic volatile organic compounds to aerosol load and, via cloud condensation nuclei activation, to cloud formation (e.g., precipitation cycles).

When evaluating the vegetation produced in the one-way coupled configuration employed here, there are potentially three sources of error that may contribute to data-model mismatch: poorly constrained parameters values and inadequate representation of the processes in LPJ-GUESS; biases in the climate produced by EMAC (which are expected to have some dependency

---

[2]www.messy-interface.org

on the spatial resolution, e.g. see Roeckner et al. (2006)) and missing processes in LPJ-GUESS (predominantly land use). The issue of missing land use was considered in the design of the evaluation method. To disentangle the mismatches resulting from LPJ-GUESS from those resulting EMAC, we consider a 'stand-alone' run of LPJ-GUESS in its standard configuration using observed climate data to assess LPJ-GUESS's implicit biases. To investigate resolution-dependent climate biases in EMAC, simulations with three spatial resolutions were performed and their performance relative to observed data were compared.

## 2 Methods

### 2.1 Model description

#### 2.1.1 The EMAC modelling system

The ECHAM/MESSy Atmospheric Chemistry (EMAC) model is a numerical chemistry and climate simulation system that includes sub-models describing tropospheric and middle atmosphere processes and their interaction with oceans, land and human influences (Jöckel et al., 2010, 2016). The historical starting point for the EMAC model was the ECHAM5 atmospheric model (Roeckner et al., 2006), but the original code has now been fully 'modularised' using the second version of the Modular Earth Submodel System (MESSy2) (Jöckel et al., 2010) including a comprehensive, but highly flexible infrastructure to the point that only the dynamical core and the runtime loop remain from the original code. The physical processes and most of the infrastructure have been split into 'modules' in accordance with the MESSy philosophy whereby such modules can be further developed to improve existing process representations, new modules can be added to represent new processes or alternative process representations, for example parameterised atmospheric convection (Tost et al., 2006), and modules can be selected at run time. EMAC has been extensively used for scientific applications of atmospheric chemistry and chemistry climate interactions from the surface to the mesosphere[3].

#### 2.1.2 The LPJ-GUESS DGVM (v4.0)

The following two paragraphs are modified from a standard LPJ-GUESS model description template which is freely available and copyright free[4]. At its core, LPJ-GUESS (Smith et al., 2001; Sitch et al., 2003; Smith et al., 2014) is a DGVM featuring an individual-based model of vegetation dynamics. These dynamics are simulated as the emergent outcome of growth and competition for light, space and soil resources among woody plant individuals and a herbaceous understorey in each of a number of replicate patches representing 'random samples' of each simulated locality or grid cell. Multiple patches (in this study 50) are simulated to account for the distribution within a landscape representative of the grid cell as a whole of vegetation stands differing in their histories of disturbance and stand development (succession). The simulated plants are classified into one of a number of plant functional types (PFTs) discriminated by growth form, phenological response, photosynthetic pathway ($C_3$ and $C_4$ herbaceous plants), bioclimatic limits for establishment and survival and, for woody PFTs, allometry and life history

---

[3]see *http://www.messy-interface.org/* for an up-to-date list of publications featuring MESSy
[4]http://web.nateko.lu.se/lpj-guess/resources.html

strategy. The standard LPJ-GUESS global PFT set containing 11 plant functional types (needle-and broad-leaved, deciduous and evergreen trees (all of which use $C_3$ photosynthesis), as well as two types of grass (one $C_3$ and one $C_4$) as described in Smith et al. (2014)) was used here. The simulations of this study were carried out in 'cohort mode,' in which, for woody PFTs, cohorts of individuals recruited in the same patch in a given year are identical, and are thus assumed to retain the same size and

form as they grow.

Primary production and plant growth follow the approach of LPJ-DGVM (Sitch et al., 2003), and includes an additional nitrogen limitation on photosynthesis (Smith et al., 2014). Canopy fluxes of carbon dioxide and water vapour are calculated by a coupled photosynthesis and stomatal conductance scheme based on the approach of BIOME3 (Haxeltine and Prentice, 1996).

The net primary production (NPP) accrued by an average individual plant each simulation year is allocated to leaves, fine roots and, for woody PFTs, sapwood, following a set of prescribed allometric relationships for each PFT, resulting in biomass, height and diameter growth (Sitch et al., 2003). Population dynamics (recruitment and mortality) are represented as stochastic processes, influenced by current resource status, demography and the life history characteristics of each PFT (Hickler et al., 2004, 2012). Forest stand destroying disturbances (such as wind throw and pest attacks) are simulated as a stochastic pro-

cess, affecting individual patches with an expectation of $0.01 \text{ yr}^{-1}$. Litter arising from phenological turnover, mortality and disturbances enters the soil decomposition cycle. Decomposition of litter and soil organic matter (SOM) pools follows the CENTURY scheme as described in Smith et al. (2014). Biogenic volatile organic compounds (BVOCs) are emitted from vegetation depending on vegetation type, leaf temperature, atmospheric $CO_2$ concentration and carbon assimilation (Arneth et al., 2007a, b). Soil hydrology follows Gerten et al. (2004).

Photosynthesis, respiration and hydrological processes operate on a daily time step and require daily temperature, precipitation and incident short wave radiation. However, monthly climate data may be provided, in which case the model interpolates daily values from the monthly values. In these circumstances, the number of precipitation days in the monthly periods may also be provided to disaggregate total precipitation into distinct rain events. In the case of unmanaged natural vegetation (as

simulated here), vegetation dynamics (such as establishment and mortality), disturbance, turnover of plant tissues and turnover between litter pools, and allocation of carbon and nitrogen to plant organs all occur on an annual basis. Simulation of wildfire is included via the GlobFIRM fire model (Thonicke et al., 2001), which simulates wildfires based on soil moisture (as a proxy for fuel moisture) and a minimum fuel (litter) threshold for burning.

All stochastic processes in LPJ-GUESS are implemented 'semi-stochastically' using a random number generator with a starting seed. This means that for a fixed starting seed, model runs with the identical settings produce identical results.

### 2.1.3 Overview of coupling implementation

The model coupling strategy employed here was to modify LPJ-GUESS such that it provides its functionality via a new sub-model in the MESSy framework. An important design priority was to maintain the integrity of the LPJ-GUESS source code by performing only minimal modifications and additions, in order to facilitate straightforward synchronising with the main
LPJ-GUESS trunk version in the future. This approach was successful, with only minor changes made to LPJ-GUESS infrastructure code and no changes to the scientific modules. For more details see Appendix A.

To provide appropriate climate forcing for LPJ-GUESS, EMAC calculates the daily mean 2 m temperature, daily mean net downwards shortwave radiation and the total daily precipitation at the end of the simulation day and provides it to LPJ-GUESS.
This is similar to the approach taken by others when coupling LPJ-GUESS to EC-Earth (Weiss et al., 2014; Alessandri et al., 2017) and RCA (Wramneby et al., 2010; Smith et al., 2011; Zhang et al., 2014). However in both these cases, daily soil moisture from the land surface model was also used to drive LPJ-GUESS (in this implementation LPJ-GUESS's internally calculated soil moisture was used). Atmospheric $CO_2$ concentration and nitrogen deposition are also provided on a daily basis from EMAC to LPJ-GUESS. Thus the LPJ-GUESS land-surface state is forced completely by the EMAC atmospheric state and
chemical fluxes.

In turn, LPJ-GUESS provides fractional vegetation cover, leaf area index, daily net primary productivity and average height of each PFT to EMAC. However, these values are not used by EMAC in the simulations presented here. Thus, we are demonstrating only a one-way coupling where the land surface state does not affect the atmospheric state. The boundary conditions
for the atmospheric model (in particular the surface energy and water fluxes) come from the pre-existing land surface representation. For an overview of the processes and feedbacks enabled in the EMAC configuration used here, as well as those to be included in future versions, please see Figure 1.

### 2.2 Simulation setup

In the coupled model, the vegetation produced by LPJ-GUESS within EMAC will be directly sensitive to biases in the climate produced by EMAC. It is well-known that these biases are dependent on spatial resolution; see Roeckner et al. (2006) for a study of the biases at different resolutions in ECHAM5 (the GCM upon which EMAC was original based). Thus the impact of the horizontal spatial resolution of the atmospheric simulation on the vegetation simulation is relevant when studying the coupled model setup. To investigate this, we performed three simulations for this evaluation, here denoted *T42*, *T63* and *T85*,
which used T42 (approximately $2.8° \times 2.8°$ grid cell size at the equator), T63 (approximately $1.9° \times 1.9°$ at the equator) and T85 (approximately $1.4° \times 1.4°$ at the equator) spectral resolutions respectively, but were otherwise identical in the considered processes, but use optimal (resolution dependent) "tuning" parameters. As LPJ-GUESS has no inter-gridcell interactions and no processes are gridcell size/spacing dependent, it has no direct sensitivity to the spatial resolution at which it is run. However,

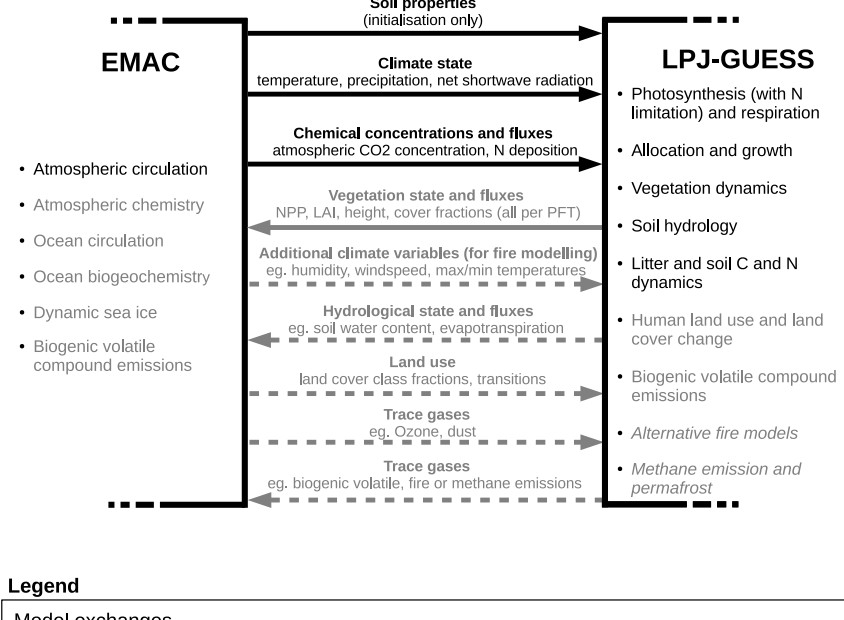

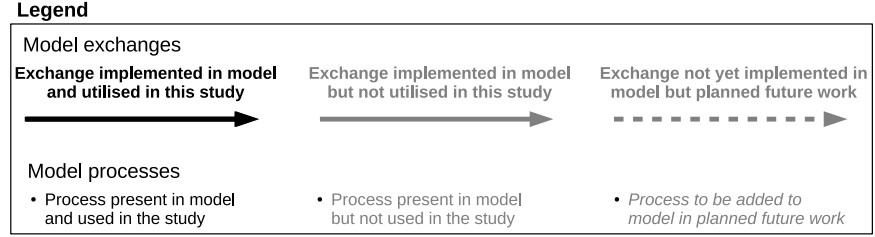

**Figure 1.** The main processes and exchanges in the coupled model framework. Processes/exchanges with normal black text/black solid arrows are included in the framework and used in the simulations presented here; processes/exchanges with normal grey text/grey solid arrows are included in the framework but not used in the simulations presented here; and processes/exchanges with italic grey text/grey dotted arrows are not included in the framework but planned in future work. All exchanges happen on a daily basis, except for soil properties which happen only during the initialisation phase.

in the coupled setup, LPJ-GUESS will be sensitive to spatial resolution via the climate data received from EMAC. Thus, the changes in the vegetation produced by the EMAC-coupled simulations at different resolution can only be due to changes in the EMAC produced climate (i.e. altered climate biases or climate aggregating). Whilst finer spatial resolution (such as T63 or higher) exhibit lower biases Roeckner et al. (2006) and so are generally preferred wherever possible; coarser resolutions (such as T42) may be used in situations where the large computational cost prohibits finer resolutions, such as long transient simulations of paleoclimate, factorial or sensitivity studies, or simulations including detailed atmospheric chemistry calculations.

The *T42*, *T63* and *T85* simulations are configured to be equilibrium simulations with boundary conditions corresponding approximately to the late 1990s and early 2000s in order to allow comparisons with the some global datasets from this period

(see section 2.3 below). $CO_2$ was constant and maintained at a level of 367 ppm (corresponding to concentration seen in around the year 1999) throughout the whole simulation. In LPJ-GUESS, nitrogen deposition rates were prescribed using data from Lamarque et al. (2013) for the decade 1990-1999 throughout. The applied EMAC model setup comprised the submodels for radiation (Dietmüller et al., 2016), clouds and convection, surface processes (see Jöckel et al., 2016), and 31 vertical hybrid pressure levels up to 10.0 hPa, representing a typical climate simulation. Note that for reasons of computational burden, the atmospheric chemistry calculations of which EMAC is capable were not activated in these simulations. The model was driven by constant solar (present day) conditions, with prescribed climatological SSTs (sea surface temperatures) and SIC (sea ice coverage) from the AMPI2 database (AMIP II Taylor et al., 2000). The climatologies are mean monthly values (so they include the annual cycle) for the years 1995-2000, but do not represent any specific year or include El Nino or La Nina event. Throughout the coupled *T42*, *T63* and *T85* simulations, LPJ-GUESS was driven exclusively by climate variables from EMAC, at no point were external climate datasets used.

As the simulations conducted here utilise only a one-way coupling, EMAC uses its standard land surface scheme which is taken from the ECHAM5 model and is described in detail by Roeckner et al. (2003). Prognostic surface and soil temperatures are calculated with a 5 layer soil model. For the hydrology component, a simple bucket model is assumed, and the water storage capacity is prescribed based on soil type data. A set of land surface data (vegetation ratio, leaf area index, forest ratio, background albedo) has been derived from a global 1 km-resolution dataset for the different horizontal resolutions of the ECHAM5 model (Hagemann, 2002). These data are used to prescribed a climatology of forest fraction (with a constant value) and of vegetation ratio and leaf area index (with a monthly temporal resolution). This prescribed land surface data is used in the model for the calculation of processes such as the interception of precipitation, the snow view in the case of snow-covered surfaces, and for evaporation (bare ground versus vegetated surfaces). Additionally, this data is used in the vertical diffusion scheme and to calculate the grid-mean surface albedo, which depends on a specified background albedo (provided as a constant input data field), a specified snow albedo (a function of temperature), the area of the grid cell covered with forest, the snow cover on the ground (function of snow depth and slope of terrain) and the snow cover on the canopy (Roesch et al., 2001).

To aid the interpretation of the EMAC simulations, we also performed an 'offline' LPJ-GUESS simulation using observed climate data from the CRUNCEP bias-corrected, re-analysis dataset (Wei et al., 2014) with a $0.5°$ spatial resolution. The simulation was performed using exactly the same code and parameter settings as the EMAC *T42*, *T63* and *T85* simulations, but code was compiled as a stand-alone model. The atmospheric $CO_2$ concentration and nitrogen deposition follow Smith et al. (2014) and the simulation is referred to as the *CRUNCEP* simulation.

In all model simulations a 500 years spin-up phase was used to allow the LPJ-GUESS vegetation to reach approximate equilibrium (confirmed by checking that net ecosystem exchange shows no systematic deviations from zero, see Appendix B). The coupled simulations used the online EMAC climate during spin-up, and the *CRUNCEP* simulations used the first 30 years (1901-1930) of the CRUNCEP dataset which were detrended and repeated. Simulations followed the standard LPJ-GUESS

procedure of starting with 'bare ground', ie. no vegetation and no C or N in the soil and litter pools. Having no plant available N present in the soil at the start of the simulation would inhibit and distort vegetation growth if N limitation was enabled. To overcome this, we followed the standard protocol, which is to run LPJ-GUESS for 100 years without N limitation but with normal N deposition to build up the N pools. After 100 years there is sufficient N in the pools, but the vegetation is inconsistent with the desired state as it has been growing without N limitation. Therefore, the vegetation is removed (and the C and N put into the litter pools), and the vegetation is allowed to regrow, this time with N limitation enabled, for a further 400 years. At that time, no significant trends in PFT extension and PFT height were obvious, but the vegetation shows interannual variability as expected.

For the *T42*, *T63* and *T85* simulations, an additional 50 years were simulated which were averaged to produce the plots shown here. In the *CRUNCEP* simulation, a further 113 years (1901-2013) were simulated using full CRUNCEP transient time series. The plots presented here show *CRUNCEP* output averaged over the years 1981-2010.

## 2.3   Model evaluation

Stand-alone LPJ-GUESS has a long history of development and has been evaluated in detail in previous work (some recent examples include modelled potential natural vegetation and forest stand structure and development (Smith et al., 2014), global net ecosystem exchange (NEE) variability (Ahlström et al., 2015) and the effect of $CO_2$ fertilisation (e.g. Medlyn et al., 2015)). Here we performed an initial evaluation focused on vegetation state variables relevant to the biophysical coupling between the land and atmosphere in the coupled model setup, in order to investigate how LPJ-GUESS responds when EMAC climate is used as the forcing data and to investigate any biases in the vegetation produced.

At this stage of model development we do not seek to precisely simulate the vegetation state of a particular year or exact period. Our atmospheric simulations are not nudged by meteorological data, but rather an unconstrained simulation based on climatological SSTs and SIC, so they do not correspond to a particular calendar period. Furthermore we prescribe a fixed atmospheric $CO_2$ concentration. Whilst we can't expect perfect agreement since (among other reasons) this is not a full transient simulation, the simulations should be sufficient to check if the model coupling is working as intended, and to gain some insight into biases that may be present when LPJ-GUESS is forced using EMAC climate. Instead, our goal with this evaluation is to perform steady state simulations where the climate and $CO_2$ forcing are constant and correspond approximately to conditions in the recent past. Thus, after 500 years of simulation, we can compare the equilibrium vegetation to satellite products based on observations in the early 2000s. Furthermore, it should be noted that the tree cover and biomass datasets (see section 2.3.1 below) reflect the biosphere as observed in the previous decade or so, and therefore inherently contains the considerable effect of human land use. This results in a conceptual mismatch between the Potential Natural Vegetation (PNV) as simulated by LPJ-GUESS and the observed biosphere state which is relevant when considering these comparisons. To quantify this ef-

fect, NME scores including a land use correction (see Appendix E for details) for these datasets are also included in Section 3.2.

Knowledge of EMAC biases is very useful for disentangling the causes of model-data disagreement in the simulated vegetation. To this end, we include bias plots of seasonal and annual biases in surface temperature, precipitation and net (plant-available) short wave radiation of the EMAC *T42*, *T63* and *T85* climate with respect to the CRUNCEP bias-corrected, reanalysis climate dataset in Appendix C.

### 2.3.1  Evaluation data sets

To provide a visual assessment of the structure and functioning of the vegetation cover at a level of detail relevant for studying interactions between the land surface and the atmosphere, we categorised the simulated vegetation into eight "megabiome" types and compared them to an expert-derived PNV map with equivalent categories. The classification of the simulated vegetation was based on Leaf Area Index (LAI) following Forrest et al. (2015). The expert-derived PNV data was taken from (Haxeltine and Prentice, 1996), regridded to the spatial resolution of the simulations using a largest area fraction algorithm and then aggregated into the eight megabiome classes (Smith et al., 2014; Forrest et al., 2015). It should be borne in mind that there are various sources of uncertainty affecting the classification of biomes in both the data and the model output, such as the somewhat subjective LAI threshold applied to the model data and the inherently subjective nature of expert classification. However, these uncertainties are to some extent minimised by the choice of broad megabiomes (see Forrest et al. (2015) for further discussion) and so, despite this lack of quantitative rigour, such classifications still provide a useful visual method for comparing vegetation cover.

For quantitative evaluation of the simulated vegetation we chose four vegetation state variables which are informative when evaluating ESMs/DGVMs, particularly with regard to the biophysical coupling and carbon flux between the land surface and the atmosphere: fractional coverage of trees; standing biomass; canopy height; and gross primary productivity (GPP). Fractional is coverage of trees is relevant both for the evaluation of stand-alone DGVMs (as it is result of both overall productivity and vegetation dynamics such as tree-grass competition and disturbance regimes) and for land surface schemes (as forested areas have different biophysical properties as non-forested area). To evaluate tree cover, collection 6 of the MOD44B MODIS tree cover (Dimiceli et al., 2015) was averaged between 2000 and 2015 and aggregated using simple averaging to an intermediate resolution of 0.05 degrees. As the MODIS tree cover layer does not include contributions from vegetation under 5m, we added an additional output variable to LPJ-GUESS which sums tree cover from tree individuals taller than 5m only. This variable was used for the comparison to MODIS tree cover.

Standing biomass is a key state variable in ESM and DGVMs as it is connected to productivity, carbon sequestration, evapotranspiration, vegetation cover, canopy height and other critical processes and variables. As such, it is a useful quantity for evaluating DGVM/ESM performance. We produce a near-global map of standing biomass combined two biomass datasets, one

tropical (Avitabile et al., 2016) and one northern temperate and boreal (Thurner et al., 2014). These dataset were aggregated to a common spatial resolution to approximately 25 km resolution using simple gridcell averaging and joined the maps (taking the average where they overlapped). Note that no data (non-forested) pixels in the original Thurner et al. (2014) dataset were set to zero to ensure consistency after the averaging procedure with the Avitabile et al. (2016) data. Furthermore, the Avitabile et al. (2016) datasets has no data for a part of the Sahara desert, Arabian peninsula and southern Australia, so no data values are present there.

Canopy height is highly relevant in a land-atmosphere context as it has a direct effect on atmospheric circulation through surface roughness length. To evaluate simulated canopy height, a 1 km tree canopy height map (Simard et al., 2011) was aggregated to an intermediate 10 km resolution by simple averaging (excluding no data values). For comparison, simulated canopy height is calculated from individual tree height (see Appendix D).

GPP is the critical quantity in earth systems modelling, both in terms of the planetary $CO_2$ budget and in terms of biosphere functioning. We used the global gridded GPP product of Beer et al. (2010) which upscales eddy flux covariance GPP measurements using gridded climate data and a selection of statistical and machine leaning techniques. This dataset provides an average annual value for the period 1995-2005 at $0.5°$ spatial resolution (which is a suitable 'intermediate-resolution' from which the data can be regridded to the simulation resolutions).

Taken together, these four quantities/data sets capture many of the key features of vegetation structure and functioning which affect biophysical land-atmosphere exchanges. The data sets were re-gridded from their intermediate resolutions to the simulation resolution using second-order conservative remapping (Jones, 1999; CDO (last), 2018). 'Conservative remapping' was initially developed to ensure that fluxes (such as energy and water) are conserved by the remapping processes (Jones, 1999), and is chosen here ensure that global area-weighted sums are conserved. 'Second-order' refers to a variant of the method which produces a smoother interpolation that 'first-order'(Jones, 1999).

### 2.3.2 Summary metric

To provide an overall summary metric of data-model agreement across the relevant spatial domain, the Normalised Mean Error (NME) is presented following the prescription and recommendations in Kelley et al. (2013). One point where we deviate from their approach is that we use NME for tree cover only, whereas Kelley et al. (2013) use Manhattan Metric (MM) and Square Chord Difference (SCD) to consider the proportions of tree, low-vegetation and bare cover simultaneously. This is because the land use correction represents deforestation by reducing tree cover in managed lands (see Appendix E), but it does not partition the area that was covered previously by trees into bare or low vegetation cover, which would be necessary for the MM or SCD. We therefore prefer not to benchmark bare or low vegetation cover in this study, and simply apply NME to tree cover.

As NME quantifies the absolute error in the model as compared to the data, the relative difference of the values for two models (compared to the same data set) can be considered the relative improvement of one over the other. For example, if one model yields a score of 0.8, and a second yields a score of 0.6, the second can be said to be 25% better than the first, since $0.6 - 0.8/0.8 = -25\%$, ie. a 25% reduction in absolute error.

It should be noted that the NME is rather different from a coefficient of correlation or a coefficient of determination. It does not attempt to derive a correlation but instead sums the differences between the model and the observation. It can be thought of as quantifying the deviation from the one-to-one line of perfect data-model agreement, rather than the deviation from a line of best fit. This means that is a rather direct and unforgiving metric, since every deviation of the model from the data is penalised

(uncertainty is not considered) and there is no possibility for the line of best fit to move to compensate for systematic biases. It also means the values are interpreted in the opposite direction to a correlation coefficient; an NME score of zero implies perfect agreement between observation and model, whereas an $r^2$ of zero would imply no correlation between the two. By the normalisation implicit in the method, using the mean value of the observations in place of the model gives an NME of 1.

**3    Results and Discussion**

**3.1    Spatial Patterns**

The simulations reproduced the global patterns of vegetation type well (Fig 2), although some regional discrepancies are visible. The most obvious mismatch between the simulations and the reconstructed megabiomes is the underestimation of the abundance of vegetation (in particular tundra) in the high northern latitudes. This is most apparent for the lowest resolution

(*T42*) EMAC simulation but improved with increasing spatial resolution, with the *T63* simulation being better substantially than *T42*. The EMAC simulation with the highest spatial resolution (*T85*) showed only a small tendency to underestimate high latitude vegetation, to a similar degree as the offline *CRUNCEP* simulation. Examination of the climate bias plots for temperature and radiation (Figs C2 and C3) reveals a high-latitude growing season low temperature bias and low plant available radiation bias at low resolution. This was somewhat mitigated at higher resolution as would be expected due to a better

representation of the synoptic scale systems in T63 and T85 (Roeckner et al., 2006). Correspondingly, the GPP simulated in this area (Fig. 3) confirms this by revealing a broad tendency to underestimate GPP above $50°$N in the *T42* simulation. This tendency lessens at higher resolution and is not seen in the offline *CRUNCEP* simulation. The consequences of this high-latitude underestimation of productivity at lower resolutions are also visible when comparing to observed tree cover (Fig. 4), biomass (Fig. 5) and canopy height (Fig. 6), showing that this issue affected both forested and non-forested vegetation types.

The extent of the temperate forest vegetation zones of the east coasts of the USA and China was underestimated in the EMAC simulations (Fig 2), which is not seen in the *CRUNCEP* simulation. Again, the underlying cause can be identified as insufficient GPP (Fig. 3), this time attributed to a negative bias in precipitation in the southern areas (Fig C1), and a negative

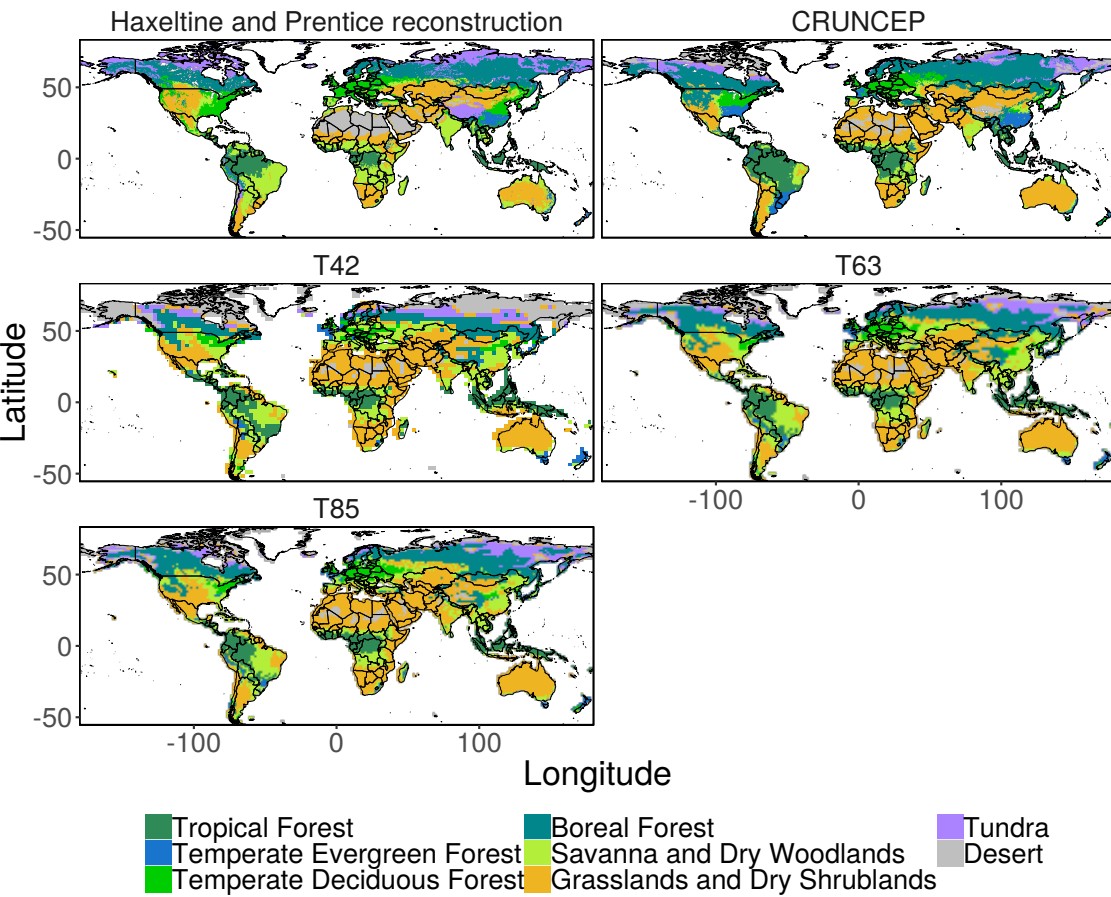

**Figure 2.** Distribution of PNV megabiomes simulated by LPJ-GUESS within EMAC (*T42*, *T63* and *T85*) and using observed climate data (*CRUNCEP*) compared to an expert-derived PNV map (Haxeltine and Prentice, 1996) following reclassification in (Smith et al., 2014; Forrest et al., 2015).

bias in the plant available radiation in the northern area of coastal China (Fig. C3). This underestimation led to reduced tree cover in the EMAC simulations compared to the *CRUNCEP* simulation (Fig. 4) and hence reduced temperate forest extent. The extent of the Savanna and Dry Woodlands vegetation type was also under-simulated in Australia and eastern Africa (Fig. 2) as a result of a negative precipitation bias, although this extent was also not well represented in the *CRUNCEP* simulation. The extent of the Sahara Desert was also underestimated in the EMAC and *CRUNCEP* simulations, as some areas grew sufficient grass cover to be classified as short grasslands. As this is present in the *CRUNCEP* simulation, it is not related to a climate biases, but rather the over-simulation of grasses in very arid regions by LPJ-GUESS.

The high productivity of the tropical rainforests was strongly underestimated by the EMAC simulations and to a lesser degree by the *CRUNCEP* simulation (Fig. 3). This manifests as an underestimation of the extent of the tropical forest vegetation type and tree cover, particularly the east of the Amazon rainforest and in Indonesia and Papua New Guinea (Fig 2 and 4), and as a wide-spread underestimation of biomass and canopy height across the region (Fig 5 and 6). The largest reductions in tropical forest extent coincide with large negative precipitation biases (Fig C1). However, the reasons for the more diffuse underestimation of productivity in the region are not immediately clear from examination of the bias plots, although some seasonal biases in precipitation and plant available radiation are apparent (Figs. C1 and C3). There is also a mild high-temperature bias in some areas (Fig. C2) which may depress productivity by the direct effects of inhibiting photosynthesis and raising plant respiration, and through the indirect effect of exacerbating water-stress conditions due to increasing evapotranspiration. As the *CRUNCEP* simulation also underestimated tropical forest GPP and biomass, there must also be issues with simulating tropical forests in LPJ-GUESS regardless of the climate data used.

The coupled model showed a tendency to overestimate GPP and, to some extent, biomass in the arid continental interiors in central Asia and the central North America (Figs. 3 and 5). This can be linked to an overestimation of both temperature and precipitation (Figs. C1 and C2). There is also an overestimation (small in magnitude but large as a relative fraction) of biomass in Europe and Eastern China, most likely due to human land use.

Considering tree cover specifically, the combined model produced reasonable global tree cover patterns (Fig. 4a.). However, regional discrepancies are starkly visible in the difference plots (Fig. 4b.) which can be attributed to three main sources. The most prominent of these are due to the fact that the simulation is of PNV (ie no human land use processes are included in the simulation) where the observations are of the current state of the planet and therefore include the impact human land use. This conceptual mismatch in the comparison can reasonably account for the large over-estimations of tree cover in Europe, China and temperate North America. The second possible reason for discrepancies in modelled tree cover compared to observed tree cover is poorly simulated productivity in the coupled model. This is most apparent in the underestimation of tree cover on the north-east coast of South America as discussed above. A final source of disagreement is due to the inevitable imperfect process representations in LPJ-GUESS. This is exemplified by the encroachment of forest cover into the Sahel in both the EMAC and CRUNCEP forced simulations. This discrepancy cannot be attributed to EMAC climate biases because it also appears in the *CRUNCEP* simulation, neither can it be due to human land use as as the PNV of the area is not forested (Fig. 2). In such cases the process representations and parameters values in LPJ-GUESS must be the cause. Another example is the general tendency to overestimate canopy height in arid and semi-arid areas (Fig. 6). Whilst this may be linked to an over-estimation of GPP, it may also be related to the lack of shrub PFTs and/or a low competitiveness of grass PFTs vs tree PFTs.

## 3.2 Summary metrics

The NME scores for all simulations and all evaluation variables are presented in Table 1. In the case of tree cover and biomass, the results are also presented with a land use correction (LUC) factor (see Appendix E). When considering the

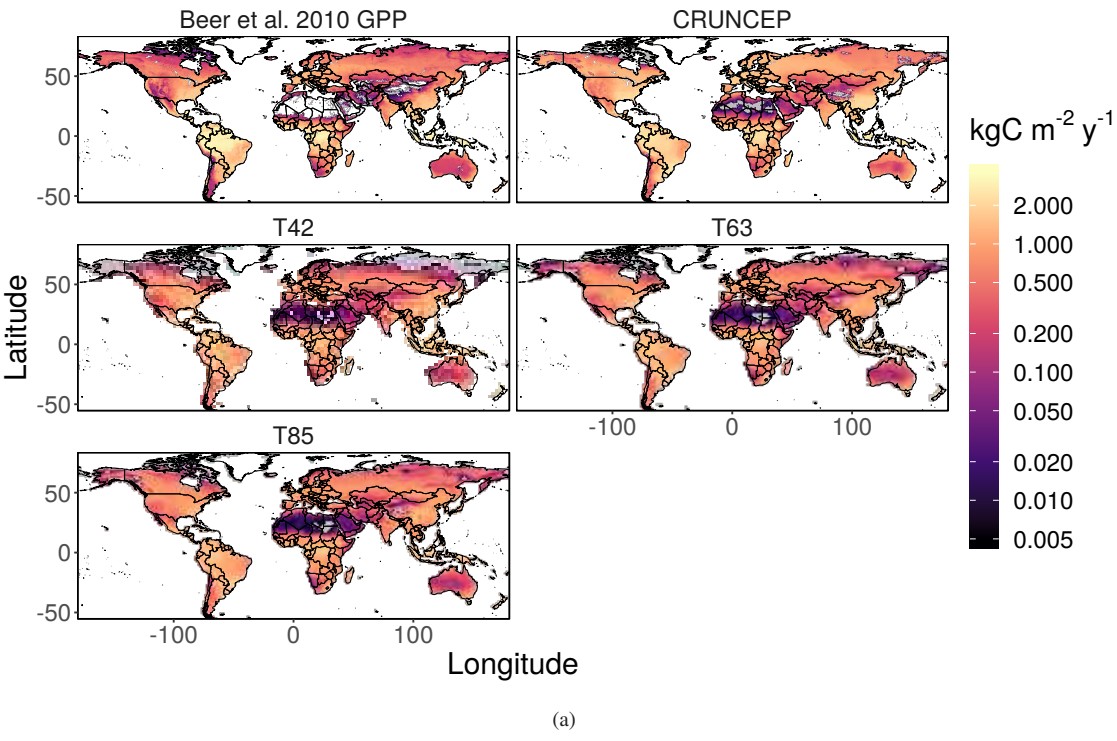

(a)

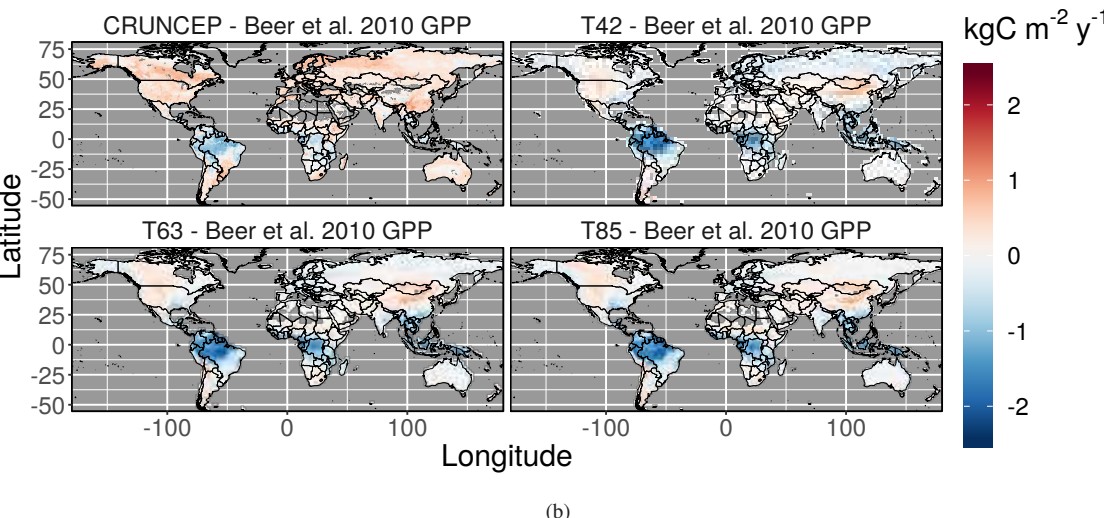

(b)

**Figure 3.** Comparison of GPP from the *T42*, *T63* and *T85* (EMAC climate) and *CRUNCEP* (observed climate) simulations with the gridded GPP product from Beer et al. (2010), a) absolute values and b) the difference between simulations minus observations. In the upper panel the colour scale in has been log-transformed and grey areas denote GPP values less than 5 gC m$^{-2}$.

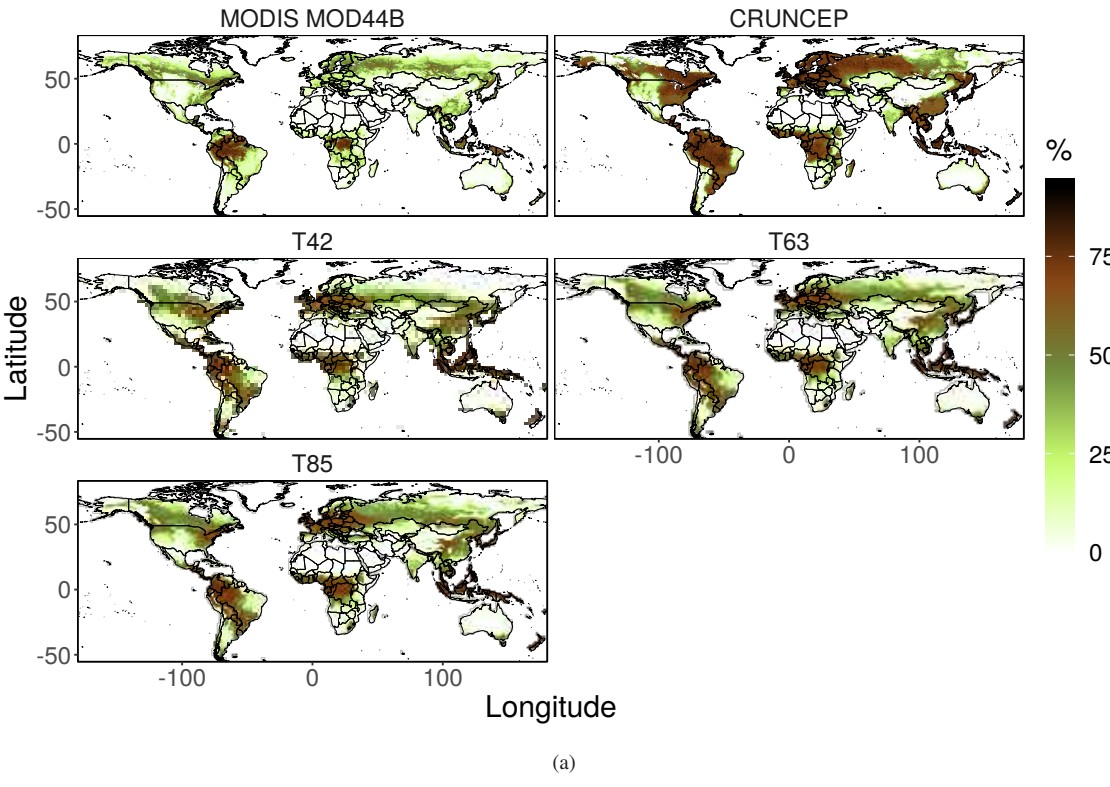

(a)

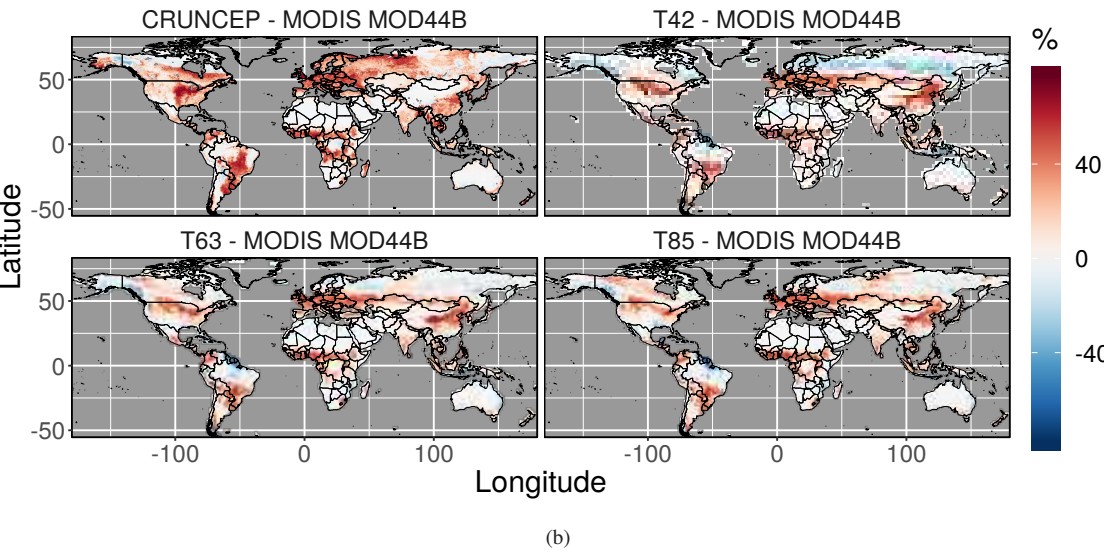

(b)

**Figure 4.** Comparison of tree cover from the *T42*, *T63* and *T85* (EMAC climate) and *CRUNCEP* (observed climate) simulations with observed tree cover from Dimiceli et al. (2015), a) absolute values and b) the difference between simulations minus observations.

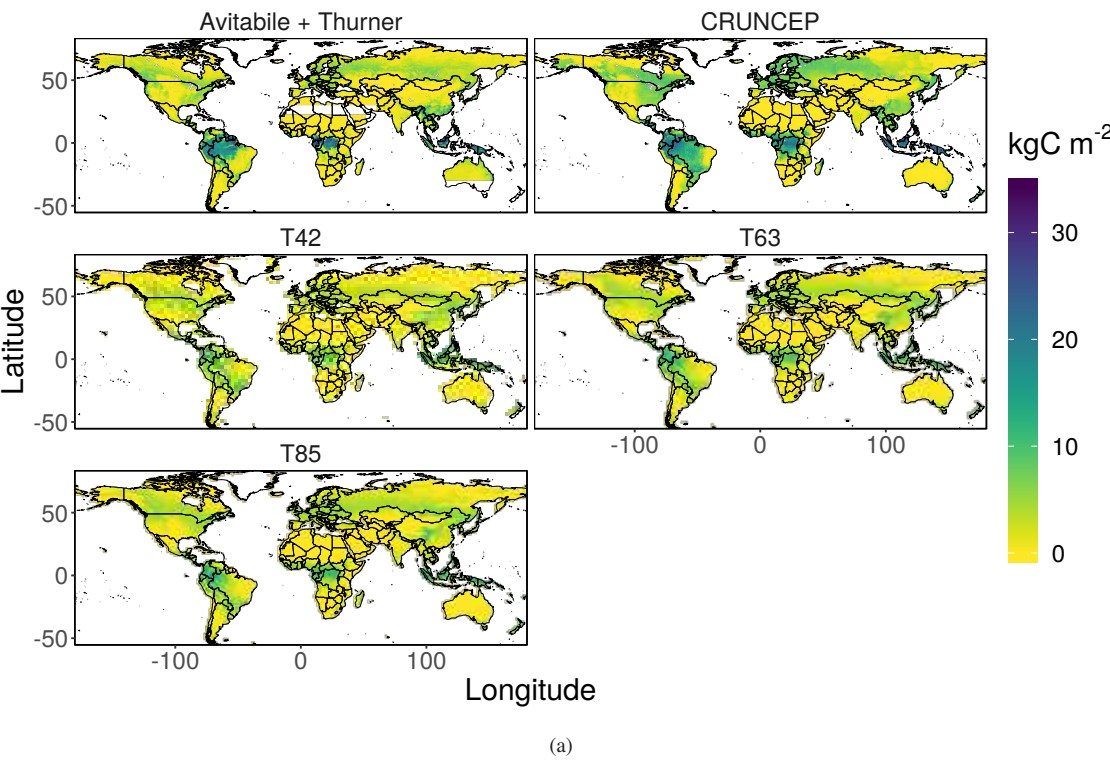

(a)

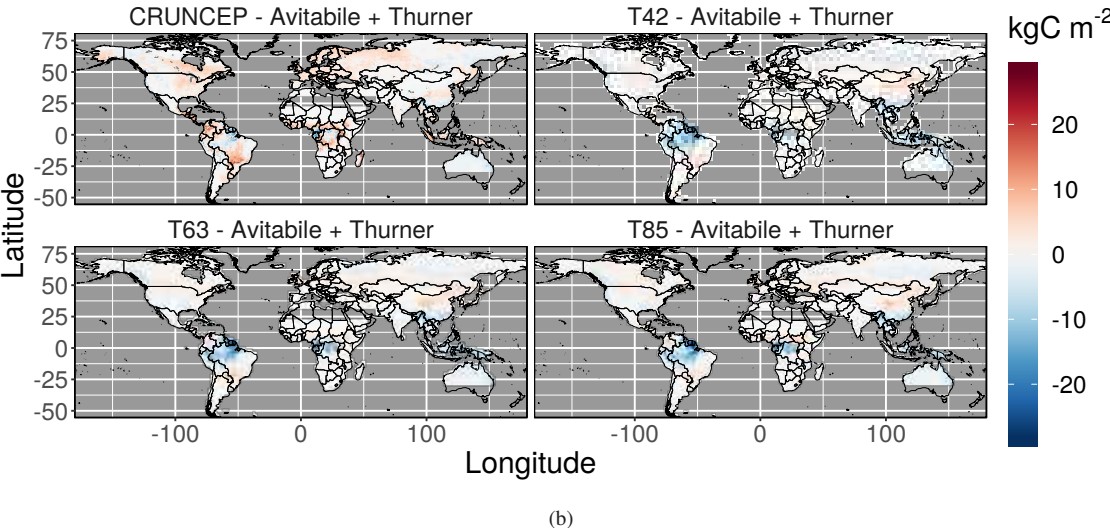

(b)

**Figure 5.** Comparison of biomass from the *T42*, *T63* and *T85* (EMAC climate) and *CRUNCEP* (observed climate) simulations with observed biomass from Avitabile et al. (2016) and Thurner et al. (2014), a) absolute values and b) the difference between simulations minus observation. Note that neither the Avitabile et al. (2016) nor the Thurner et al. (2014) biomass dataset provide biomass for a band across the Sahara desert and Arabian peninsula and in southern Australia, so no data are plotted there.

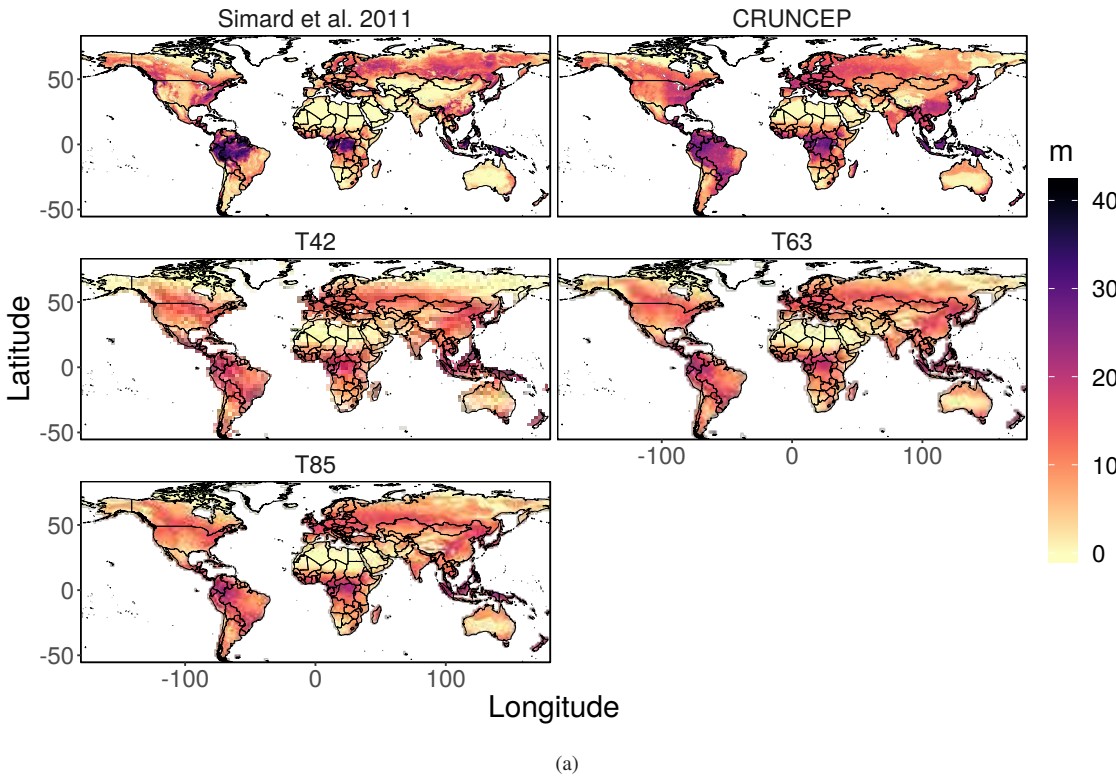

(a)

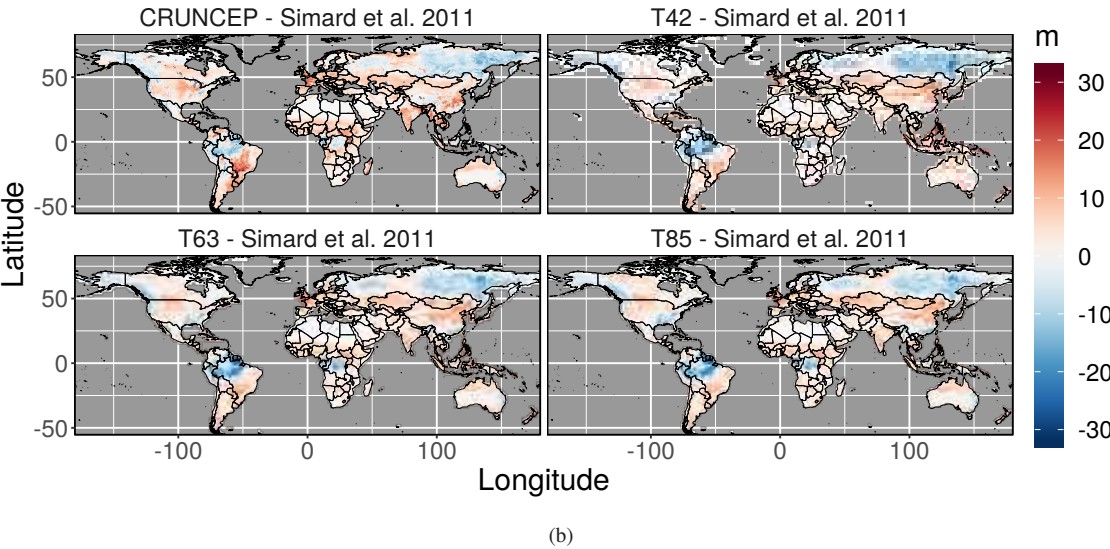

(b)

**Figure 6.** Comparison of canopy height from the *T42*, *T63* and *T85* (EMAC climate) and *CRUNCEP* (observed climate) simulations with observed canopy height from Simard et al. (2011), a) absolute values and b) the difference between simulations minus observation.

**Table 1.** NME scores for the vegetation produced by the *T42*, *T63*, *T85* and *CRUNCEP* simulations compared to four gridded global datasets both with and without a LUC (Land Use Correction) where applicable. Note that lower scores imply better agreement between simulation and observation.

| Dataset | without LUC | | | | with LUC | | | |
|---|---|---|---|---|---|---|---|---|
| | *T42* | *T63* | *T85* | *CRUNCEP* | *T42* | *T63* | *T85* | *CRUNCEP* |
| Tree cover (Dimiceli et al., 2015) | 0.93 | 0.85 | 0.84 | 1.1 | 0.78 | 0.67 | 0.63 | 0.62 |
| Biomass (Avitabile et al., 2016; Thurner et al., 2014) | 0.72 | 0.68 | 0.68 | 0.76 | 0.75 | 0.68 | 0.66 | 0.56 |
| Canopy height (Simard et al., 2011) | n/a | n/a | n/a | n/a | 1.0 | 0.87 | 0.84 | 0.77 |
| GPP (Beer et al., 2010) | 0.55 | 0.54 | 0.53 | 0.60 | n/a | n/a | n/a | n/a |

The canopy height data was produced in such a way that no land use correction is necessary, and the land use cannot be meaningfully applied to the modelled GPP.

global NME scores for the coupled model, we see a level of agreement in the ball park of DGVMs used with offline climate data (Kelley et al., 2013) and the offline *CRUNCEP* LPJ-GUESS simulations performed in this study (Table 1). The fact that the GPP scores for the coupled model are actually better than stand-alone LPJ-GUESS can be dismissed as either an artefact of the coarser spatial resolution of the EMAC simulations or a 'fortuitous cancellation of errors' between the EMAC-produced
climate and the LPJ-GUESS parameterisation. It should be noted that in this work we are evaluating only the first milestone of model development, and at this point the model is known to be incomplete (particularly with regard to human land use) and no tuning has been performed to either model component. As such, these summary metrics are not meant to demonstrate a particular level of agreement better than some arbitrary threshold, but rather they are included to quantitatively evaluate the differences between models runs at difference resolutions and to assess the effect of human land use via a land cover correction
factor. They also give a first overview of how the model simulates key features of the vegetated land surface and quantitatively indicate (on a normalised scale) which properties are well simulated and which may require additional tuning in future work.

Applying the LUC has a marked improvement on the tree cover NME scores (in terms of percentage reduction of error: 16% for *T42*, 22% for *T63*, 33% for *T85* and 43% for *CRUNCEP*), implying that much of the discrepancy seen between sim-
ulation and observation apparent in Fig 4 is indeed due to human land use as expected. Whilst this could be expected, it does demonstrate that a large part of the mismatch between the tree cover simulated by the coupled model and the observed tree cover is due to human land use which is not present in the current model. Indeed it highlights the fact that applying a land use scheme or correction will be important when enabling feedbacks from the land surface to the atmosphere in the future, particularly as tree cover plays a direct role in the determining the biophysical properties of the land surface as seen by the
atmospheric model. Whilst applying the land use correction does improve biomass agreement in the *CRUNCEP* simulation (by 26%) and the *T85* simulation (by only 3%) , it leaves the agreement in *T63* simulations essentially unchanged (0% change) and actually it worsens agreement in the *T42* simulation by 4%. This can be understood in the context of Fig 5, which shows that the combined model underestimates biomass (particularly at low resolution), and so further reducing the biomass (through

the land use correction) will worsen agreement. This indicates the importance of including land use effects in a consistent and realistic way in the coupled model, and that improved simulation of biomass is also critical due its status as a key state variable in the land surface representation. As LPJ-GUESS biomass has been shown to be sensitive to disturbance rates (Hickler et al., 2004; Pugh et al., 2019), the average global patch-destroying disturbance rate of 0.01 yr$^{-1}$ could be re-evaluated and the rather simplistic mortality could be further developed in LPJ-GUESS. In summary, and especially in light of the caveats above, these good NME scores give confidence that LPJ-GUESS is a suitable choice for coupling to EMAC; that the implementation is working correctly; and that on global scale, LPJ-GUESS is not critically sensitive to biases in the climate produced by EMAC.

For the coupled simulations, increasing spatial resolution improved the agreement between simulations and observations for all variables, with the exception of biomass at the higher resolutions. The GPP agreement improved consistently by 2% with increasing spatial resolution. The canopy height NME improved by 13% from *T42* to *T63*, with a smaller increase of 3% for *T85* compared to *T63*. For the tree cover the improvements going from *T42* to *T63* are appreciable at 14% and 9% with and without the LUC, respectively; however more modest improvements of 6% and 1% (with and without LUC) are seen going from *T63* to *T85*. Biomass is more realistic going from *T42* to *T63*, with improvements of 9% (with LUC) and 6% (without LUC), but agreement did not improve notably going from *T63* to *T85*. Averaging over all the data sets gives an improvement of 10% when going from *T42* to *T63* spatial resolution (with the land use correction applied where appropriate). Going from *T63* to *T85* yields a smaller average improvement of 4%. Whilst this is a decent improvement which may be important in some applications (particularly in the case of the fully coupled model runs), the law of diminishing returns clearly applies, so going to higher resolutions than T85 may not be worth the additional computation burden. As LPJ-GUESS processes are completely independent of spatial resolution, these improvements must be coming from the EMAC-simulated climate. It is also worth considering that these gains are particularly noteworthy because conducting benchmarking at higher resolutions is more rigorous and would generally be expected to result in lower benchmark scores. This can be understood mathematically as a consequence of a larger degree of spatial aggregation (of both the evaluation data and the model input data) at coarser resolutions leading to more homogenised values and therefore more agreement. While the result that the resolution of the atmospheric simulation has such a significant effect on the vegetation is not surprising in itself, it does highlight that when considering the bidirectionally coupled model with dynamic (as opposed to prescribed) vegetation, thorough investigation must be made of the effect of the resolution of the atmospheric model on both model components, particularly considering feedbacks between them.

In light of these results, we would recommend against using T42 resolution given the high latitude growing season temperature and radiation bias and it's effect on the vegetation and, potentially, the resulting feedback to the atmosphere (in the case of the fully coupled model). These effects can be mitigated to a large extent by using T63 resolution without incurring too much additional computation cost. Stepping up to T85 resolution or higher may or not be beneficial depending on the details of the simulation and the study.

### 3.3 Future work

The work and simulations presented here are only the first milestone on a planned model integration roadmap. Figure 1 shows the various processes and feedbacks to be enabled as part of this roadmap. The next step is to use albedo and roughness length schemes and the vegetation and forest fractions (which are used in the standard land surface scheme to determine the hydrolog-
ical fluxes) to form a bidirectional coupling of interactive vegetation and climate. The work is underway and parameterisations for determining albedo and roughness length and the exchange of the relevant variables are already implemented in the EMAC code base (Tost et al., 2018).

Following this, the next critical step will be to enable land use and agriculture in LPJ-GUESS within EMAC. This will have
be beneficial not just in terms of improving the representation of the land surface as a boundary condition to the atmospheric circulation model, but also because model evaluation and benchmarking will become easier to perform and interpret. However, this step will involve a significant amount of development work to modify the LPJ-GUESS code to receive land cover, state transition and management data from EMAC rather than through the existing channels in LPJ-GUESS. In contrast, the calculation of biogenic volatile organic compounds in LPJ-GUESS will be fairly simple as the only additional variables that are
required are daily maximum and minimum temperatures (Arneth et al., 2007a, b).

Another further developmental step is to improve the representation of fire and associated emissions in LPJ-GUESS. The GlobFIRM fire model (Thonicke et al., 2001) is included as a module in LPJ-GUESS and is enabled by default. However, GlobFIRM is very a simple model which simulates wildfires based on soil moisture (as a proxy for fuel moisture) and a
minimum fuel (litter) threshold for burning. Other fire models of greater complexity have been used with LPJ-GUESS: SPIT-FIRE (Lehsten et al., 2009; Thonicke et al., 2010; Rabin et al., 2017), SIMFIRE (Knorr et al., 2016) and SIMFIRE-BLAZE (Rabin et al., 2017). Whilst these models are not currently in the main LPJ-GUESS code version used here, efforts to integrate SIMFIRE-BLAZE are underway, and it is anticipated that it will be available soon in the main LPJ-GUESS version and subsequently in EMAC. Similarly, a representation of tundra, arctic wetlands and permafrost has been developed within a separate
branch of LPJ-GUESS (Miller and Smith, 2012) which is now being re-integrated into the main model version. A potential longer-term aim is to include a representation of the phosphorus cycle which strongly limits terrestrial productivity (Elser et al., 2007) and is currently in development for LPJ-GUESS.

Initially, LPJ-GUESS was developed as a stand-alone DGVM featuring biogeochemical cycling and vegetation dynamics.
It was not designed as a land surface scheme and so some physical properties of the vegetation, such as canopy height, were not high priorities during development. Furthermore, many remotely-sensed data sets, such as the canopy height data used here, were not available during the model's initial development and calibration. It is therefore not surprising that in this study we found that GPP was the best-simulated quantity and canopy height was the least well simulated. Given the direct effect of canopy height on the atmosphere via roughness length, it may be appropriate to adjust the parameterisation in LPJ-GUESS to

improve the simulation of canopy height. Candidate parameters include PFT-specific coefficients in the allometric equations (Smith et al., 2001; Sitch et al., 2003), which directly control tree height, and the maximum crown area of trees. Maximum crown area for all trees is currently set to 50 m$^2$ in LPJ-GUESS, which is rather low for tropical trees (see, for example the maximum reported in Seiler et al. (2014)), and may result in an under-weighting of the contribution of mature individuals to canopy height (see Appendix D). A systematic tuning exercise for LPJ-GUESS may yield appreciable improvements in the representation of canopy height and other important aspects of the global LPJ-GUESS vegetation state.

Longer term and more ambitious goals on the roadmap are to fully replace the soil-vegetation part of the hydrological cycle in EMAC with that of LPJ-GUESS and to use LPJ-GUESS to close the land surface energy balance. Such developments may benefit from synergies with other on-going coupling work in the LPJ-GUESS community. When completed, these developments will extend the EMAC model into a full Earth system model including atmosphere (ECHAM5) with full chemistry (see Jöckel et al., 2010), vegetation and land surface processes (LPJ-GUESS) and an ocean component (MPIOM) (see Pozzer et al., 2011) with ocean biogeochemistry (see Kern, 2013). Furthermore, coupling efforts to COSMO (Baldauf et al., 2011) via the MESSy framework are undergoing. Additionally, linking LPJ-GUESS to ICON/MESSy in a similar way as for EMAC is straightforward.

## 4 Conclusions

Here we have reported the first steps towards to producing a new atmospheric-chemistry enabled ESM by combining an atmospheric-chemistry enabled AOGCM with a DGVM. The technical coupling work is now complete and has been achieved in a manner which respects both the integrity and philosophy of the two modelling frameworks, and will therefore allow relatively straightforward updates to both components.

Results from one-way coupled simulations (in which climate information generated by EMAC is used to force LPJ-GUESS but no land-surface information is relayed back to EMAC) showed that the vegetation patterns produced from EMAC climate are reasonable on a global scale. However some regional deviations from the observed vegetation are apparent. Some of these are due to the simple fact that in this configuration LPJ-GUESS produces PNV (potential natural vegetation with no human impacts) while the observed vegetation implicitly includes human impact. This effect was confirmed by performing a correction to account for human land use which improved agreement between simulation and observation. Human land use can be included in future model versions by utilising the recently developed crop and managed land module in LPJ-GUESS (Lindeskog et al., 2013), the use of which should mitigate these issues to a large extent.

A second class of deviations is due to biases in the simulated climate, particularly precipitation biases. This is a more difficult problem to solve; improving climate simulations is the subject of much on-going research. However, it is clear that using higher spatial resolution mitigates climate biases which results in tangible improvements in the simulated vegetation. Based

on the three spatial resolutions, we recommend using T63 resolution as a minimum to due climate biases in the high latitudes in the T42 simulation which resulted in insufficient growth of vegetation. However, using dynamically simulated land surface boundary conditions (in this case from LPJ-GUESS) in a bidirectionally coupled model will alter the atmospheric state and therefore the climate biases. This will be the subject of future studies.

Finally, there are some discrepancies arising as an inevitable consequence of the approximations, missing processes and parameter uncertainties inherent in a process-based model such as LPJ-GUESS. These may be reduced by on-going improvements occurring as LPJ-GUESS is further developed and refined. Given the rather rigorous requirements placed on a biosphere model when bidirectionally coupled to an atmospheric model, it may also be necessary to perform some focused model development work with the goal of improving vegetation functioning and structure so that key biophysical quantities (such as albedo and roughness length) are better simulated. Of the variables evaluated here, canopy height was found to be the least well-simulated, suggesting that re-tuning tree height in LPJ-GUESS might be an important step to ensure good performance of the fully coupled model.

Whilst further work remains before the full ESM is completed, we have demonstrated that coupling LPJ-GUESS into the EMAC/MESSy modelling framework has been accomplished, and that LPJ-GUESS provides a suitable basis for an improved and dynamic representation of the land surface in EMAC. Future development should focus on completing the two-way model coupling and investigate the effects of the atmosphere. Once the full coupling has been enabled and calibrated, the resulting model will be powerful tool for investigating atmosphere-biosphere interactions. In addition to the broad range of applications possible for any ESM, the particular strength of EMAC with LPJ-GUESS vegetation will be applications studying interactions and feedbacks at the atmosphere-biosphere boundary, for example: the nitrogen cycle, trace gas emissions from fire, the atmospheric dynamics of reduced carbon including biogenic volatile organic compound emissions from vegetation and methane from fires, ozone dynamics and the resulting damage to vegetation, and the effects of a wide spectrum of terrestrially emitted trace gases on cloud and aerosol formation and dynamics.

*Code availability.* The Modular Earth Submodel System (MESSy) is continuously further developed and applied by a consortium of institutions. The usage of MESSy and access to the source code is licensed to all affiliates of institutions which are members of the MESSy Consortium. Institutions can become a member of the MESSy Consortium by signing the MESSy Memorandum of Understanding. More information can be found on the MESSy Consortium Website (http://www.messy-interface.org). As the MESSy code is only available under license, no DOI is possible for MESSy code versions. However, the code for coupling to LPJ-GUESS used in this manuscript has already been included in the latest official MESSy version (v2.54).

LPJ-GUESS is used and developed world-wide, but development is managed and the code maintained at Department of Physical Geography and Ecosystem Science, Lund University, Sweden. Model code can be made available to collaborators on entering into a collaboration agreement with the acceptance of certain conditions. The MESSy-coupled version of LPJ-GUESS will be maintained as a derivative of

LPJ-GUESS. Because access to LPJ-GUESS is also restricted, no DOI can be assigned to LPJ-GUESS versions. The specific code version used here to enable the MESSy coupling the LPJ-GUESS code in EMAC, code is archived on the LPJ-GUESS subversion server with tag "_publications/MESSY_1.0_20180108" in the catalogue "MESSy". For more details and contact information please see the LPJ-GUESS website (http://web.nateko.lu.se/lpj-guess) or contact the corresponding author.

For review purposes, the code used here is available to the editor and reviewers via a password protected link on condition that the code is for review purposes only, it cannot be used for any other purposes and must be deleted afterwards.

## Appendix A:  Details of coupling implementation

One of the main priorities during the coupling implementation was to change the LPJ-GUESS source code as little as possible. As such, only the following modifications were made to the LPJ-GUESS code:

– Creation of three new functions to be called externally by the MESSy framework to: initialise an LPJ-GUESS simulation (or restart from a saved state if appropriate); perform one day of LPJ-GUESS simulation given one day of EMAC climate data and return the relevant data; and save the LPJ-GUESS state to disk. These key functions encapsulate the interactions between MESSy and LPJ-GUESS.

– Creation of a new input module (an instantiation of the LPJ-GUESS *C++* class `InputModule`) to handle model
initialisation in the MESSy framework, and the inclusion of one extra member function of the `InputModule` class (to read the gridlist file) which was implemented as a dummy function in the other LPJ-GUESS input modules.

– Creation of one additional internal function to calculate the daily values to be handed back to EMAC (such as vegetation cover for a particular PFT).

– Inclusion of an additional output module to save model output useful for benchmarking.

– Minor modifications to the standard output module such that the MPI rank number of each process is added to the file output names allowing the output from each process to be stored in the same directory.

– Minor modifications to the standard LPJ-GUESS restart code to allow the MESSy restart cycle number to be added to the names of the state files to be saved or read by LPJ-GUESS.

– Removal of some of code for the LPJ-GUESS real-time visualisations which is incompatible with the MESSy framework.

No changes to the scientific modules were made, and the directory structure and compilation machinery were untouched. Wherever new code conflicted with the standard offline version, a preprocessor directive was used to ensure that the model still compiled in the standard way outside the MESSy framework. Thus the integrity of LPJ-GUESS was maintained so that updates from the LPJ-GUESS trunk version can be applied relatively easily and the code can still be compiled and run offline.

On the MESSy side, the `Makefile` has been modified to compile the complete LPJ-GUESS code into a single library file using `CMake`, which is LPJ-GUESS's native compilation machinery[5]. This was necessary because LPJ-GUESS is written in *C++* whereas EMAC is written in *FORTRAN*. The LPJ-GUESS library is linked to the rest of the EMAC code with the standard linker of EMAC (also including a link to *C++* standard library). LPJ-GUESS provides functionality to new EMAC submodel (VEG) with its individual submodel interface layer (see Jöckel et al., 2005), which is controlled by a namelist and invokes the above mentioned *C++* functions to communicate with LPJ-GUESS.

In the initialisation phase, the grid from EMAC is transferred into LPJ-GUESS. Note, that currently there is only a geographic decomposition induced by EMAC, which could lead to some processors not having a single land box and cause idle time for that specific CPU. In future an additional, individual decomposition of the land gridcells to optimise CPU balance is desired, which could make use of the *UniTrans* library developed within the ScalES project[6], which shall also be used for load balanced distribution of chemical gaseous reactions. However, currently the LPJ-GUESS code with its daily timestep consumes very little computing time compared to the climate calculations of EMAC.

In its interface layer, the VEG submodel accumulates the required input fields (daily temperature, precipitation, incoming solar radiation and atmospheric $CO_2$ concentration) for the vegetation and, depending on the time step length of the LPJ-GUESS code, triggers the call of the LPJ-GUESS routines using the *TIMER*-MESSy interface structure routines (see Jöckel et al., 2005).

The combined model uses the pre-existing restart facilities of the LPJ-GUESS code, such that when EMAC triggers a restart, a restart is triggered for LPJ-GUESS. When a simulation is continuing from a restart, a flag is sent to the LPJ-GUESS code and the restart files of LPJ-GUESS state are read in allowing a seamless, continuous simulation. This feature may also be used to start a simulation with already well established vegetation from LPJ-GUESS restart (state) files, potentially significant saving significant amounts of CPU time that would otherwise be required to spin up the vegetation (typically the order of 500 simulation years).

---

[5]The current development version of EMAC including LPJ-GUESS is equipped with a new "Makefile" for a standard linux "make (gmake)" with the same functionality as the original "cmake" compilation; the updated compilation process does not require an up-to-date installation of cmake.

[6]https://www.dkrz.de/Klimaforschung/dkrz-und-klimaforschung/infraproj/scales/scales

## Appendix B: Net Ecosystem Exchange test for equilibrium

The net ecosystem change plots shown in Figure B1 display no systematic variation from zero in either space or time indicating that the vegetation from LPJ-GUESS is in equilibrium with the climate from EMAC. The small variations from zero that are visible are due to the stochastic processes in LPJ-GUESS and internal climate variability in EMAC.

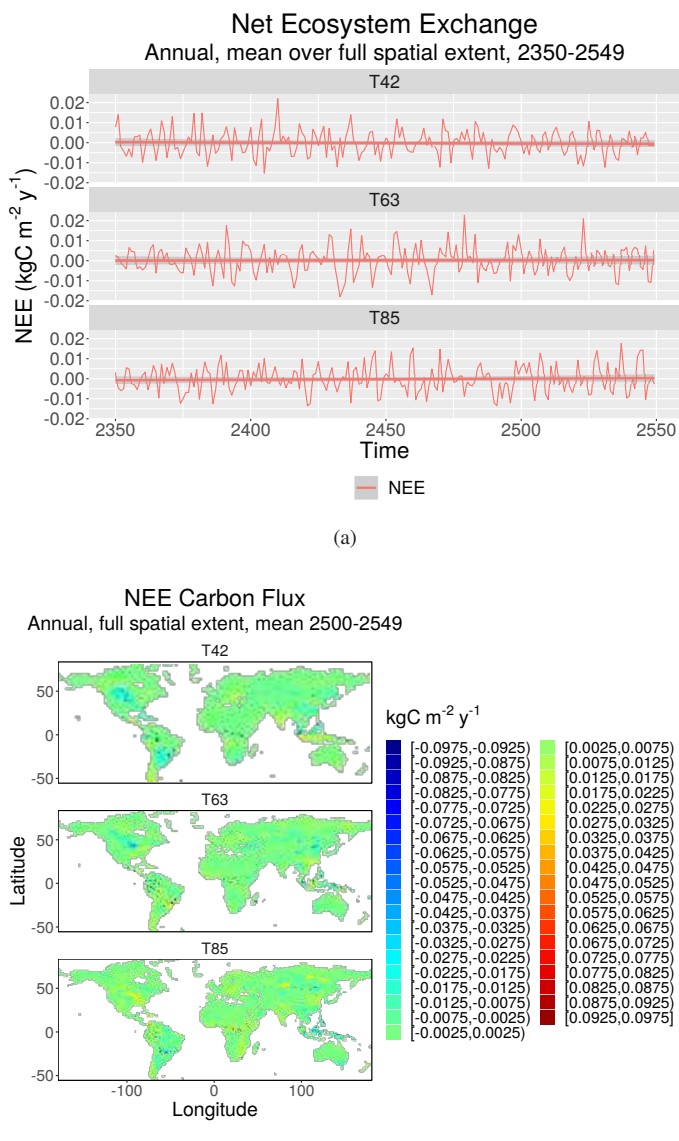

**Figure B1.** Net ecosystem exchange (NEE) for all EMAC simulations a) averaged globally and plotted for the last 200 years of the simulation and b) averaged over the last 50 years of the simulation and plotted globally. The straight red lines with grey uncertainty bands in panel a) are linear fits to the data.

## Appendix C: EMAC climate biases

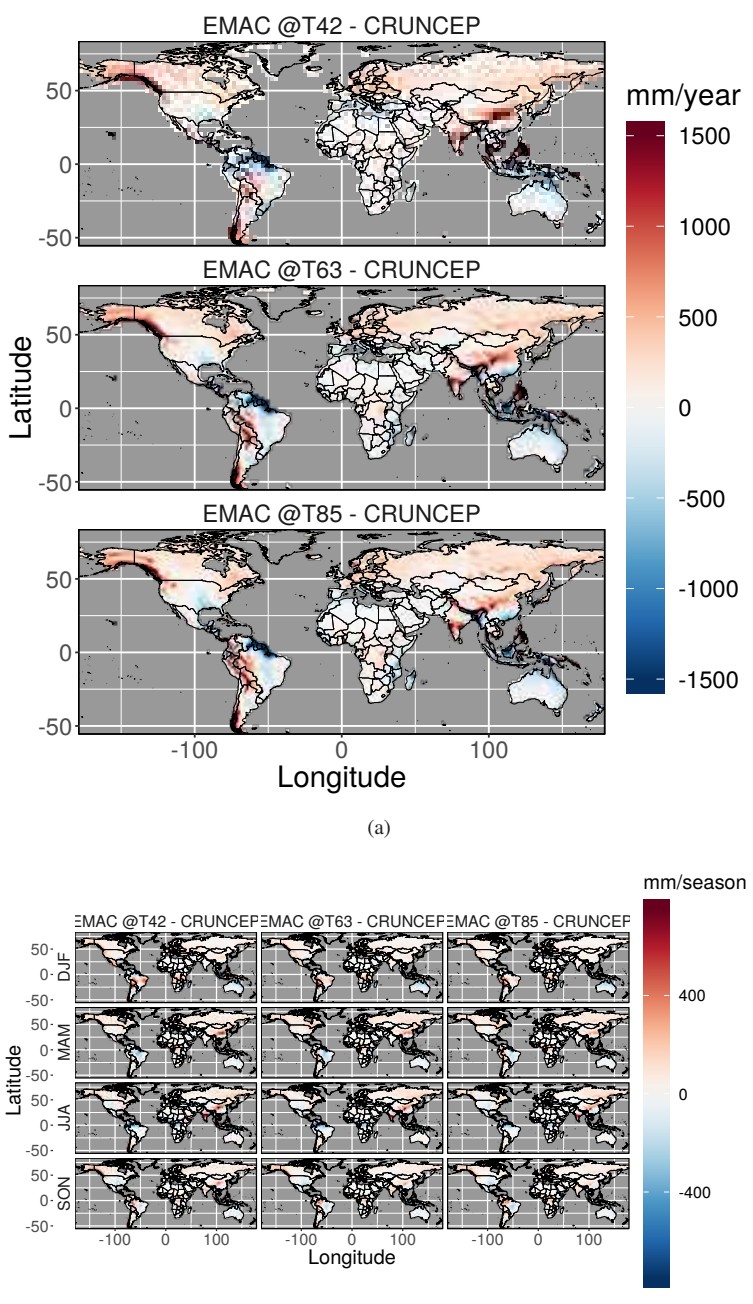

(a)

(b)

**Figure C1.** The a) mean annual precipitation bias and b) mean seasonal precipitation bias between the observed CRUNCEP dataset (1981-2010) and EMAC simulations (last 50 years of simulation). Note that ensure visibility of relatively low precipitations biases, the plotted values are capped at 750 mm/season and 1500 mm/year in the seasonal and annual plots, respectively.

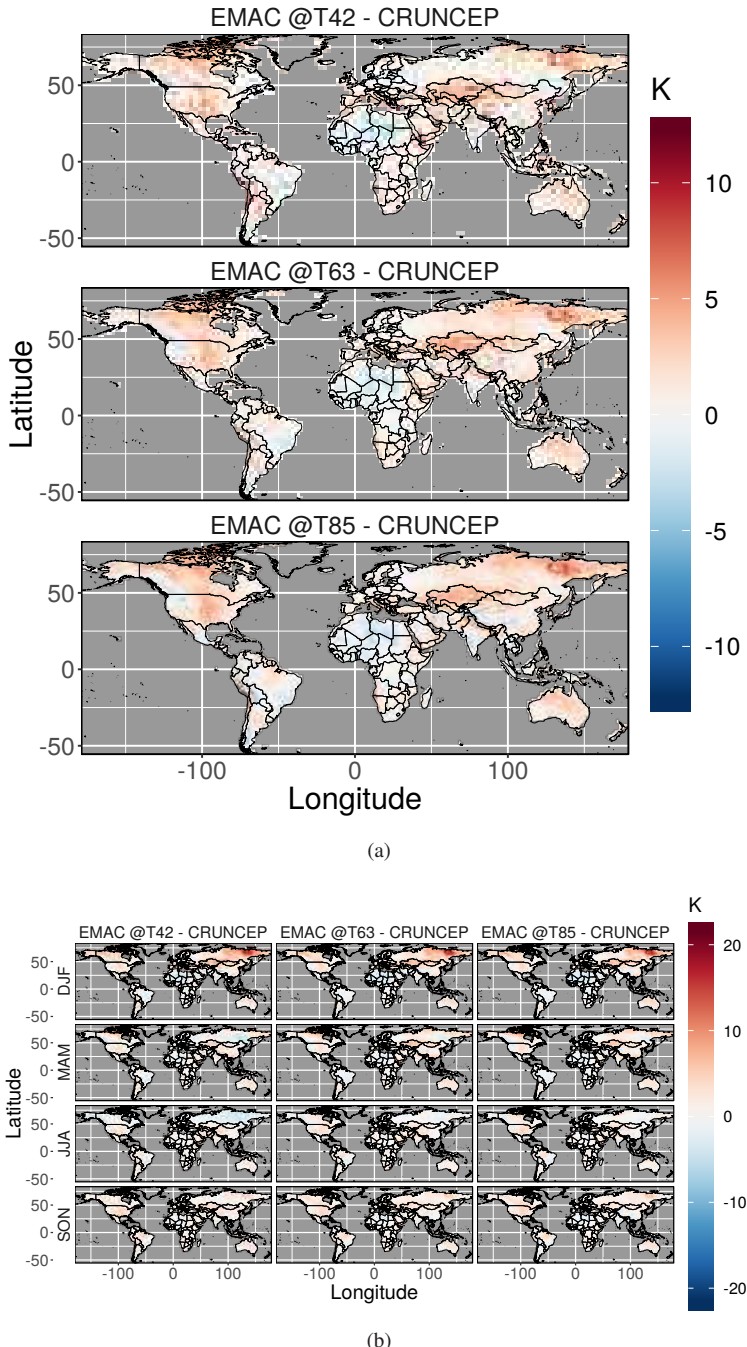

**Figure C2.** The a) mean annual temperature bias and b) mean seasonal temperature bias between the observed CRUNCEP dataset (1981-2010) and EMAC simulations (last 50 years of simulation).

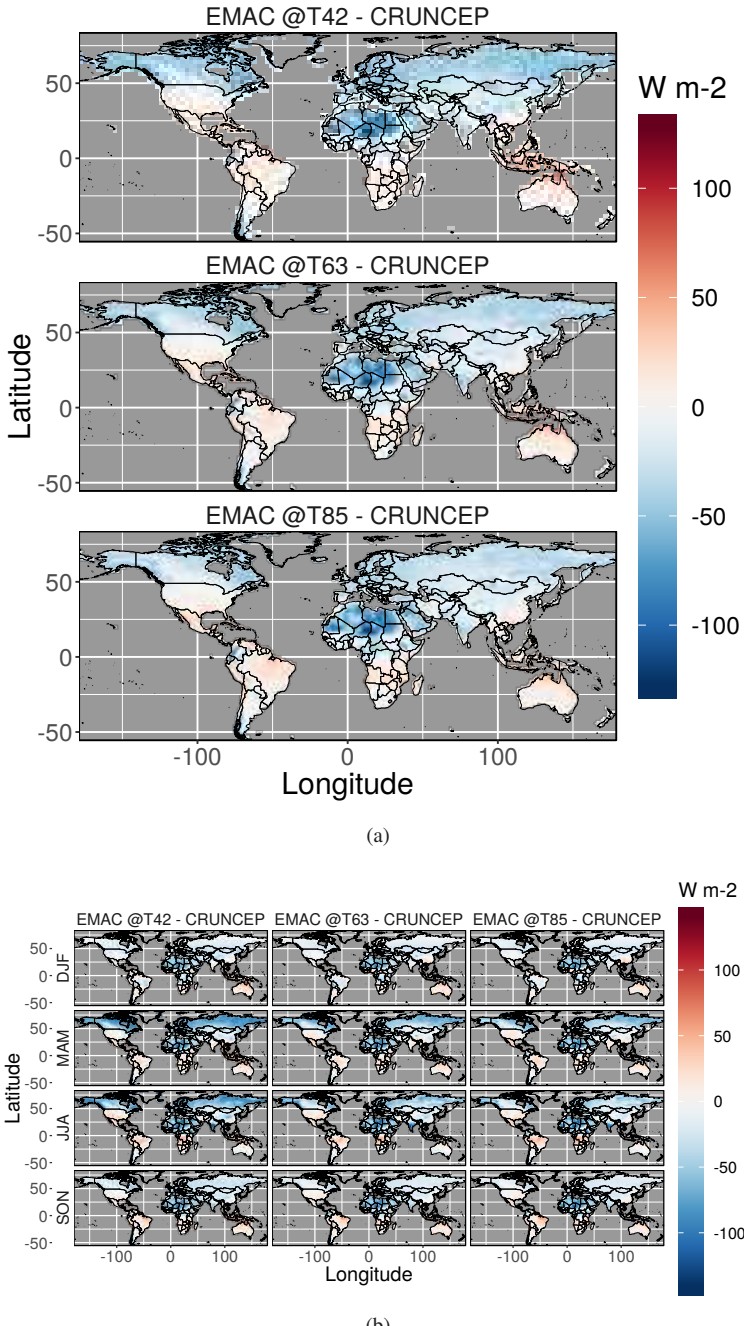

(a)

(b)

**Figure C3.** The a) mean annual net shortwave radiation bias and b) mean seasonal net shortwave radiation bias between the observed CRUNCEP (1981-2010) and EMAC simulations (last 50 years of simulation). Note that these plots compare shows the radiation available for vegetation in the CRUNCEP and EMAC forced LPJ-GUESS simulations, and consequently the gross shortwave radiation has been adjusted by different albedo values. The CRUNCEP gross shortwave flux has had the standard LPJ albedo value of 0.17 applied (temporally and spatially invariant), and the EMAC gross shortwave flux has had the spatially and temporally varying albedo values in the land surface scheme applied.

## Appendix D:  Canopy height calculation

Canopy height of a patch was calculated from individual tree cohort heights by a simple algorithm that attempts to reconstruct top of canopy height as it would be viewed from above, for example by a satellite. It utilises the modelled quantity Foliar Projective Cover (FPC), which is the ground area covered by the crowns of trees of a cohort expressed as a fraction of the patch area. LPJ-GUESS allows limited overlapping of trees and hence the sum of tree cohort FPC can be greater than unity. In this case cohorts are selected in descending order of height until the sum of their FPC reaches 1, i.e. smaller cohorts are assumed to be under the taller cohorts and so do not contribute to top of canopy height. Cohorts smaller than 5 m don't contribute to canopy height as the remotely-sensed dataset does not include canopies lower than 5 m. Having selected the contributing tree cohorts, the canopy height is simply the FPC-weighted sum of the contributing cohort heights.

## Appendix E:  Land use correction

In order to correct the model output for 'missing' tree cover and biomass due to human land cover modification, a simple correction was derived from the Globcover2009 land cover product (Arino et al., 2012). For each simulated gridcell the fraction of naturally vegetated land pixels from the Globcover2009 product was calculated. This fraction was then used to scale the model outputs of tree cover and biomass to give a simple, first order reduction based on remotely-sensed data. For these purposes, natural vegetated land cover was defined by the following classes in the Globcover 2009 dataset:

- 40 Closed to open broadleaved evergreen or semi-deciduous forest

- 50 Closed broadleaved deciduous forest

- 60 Open broadleaved deciduous forest/woodland

- 70 Closed needleleaved evergreen forest

- 90 Open needleleaved deciduous or evergreen forest

- 100 Closed to open mixed broadleaved and needleleaved forest

- 110 Mosaic forest or shrubland/ grassland

- 120 Mosaic grassland/forest or shrubland

- 130 Closed to open (broadleaved or needleleaved, evergreen or deciduous) shrubland

- 140 Closed to open herbaceous vegetation (grassland, savannas or lichens/mosses)

*Author contributions.*  HT and MF performed the model coupling. MF performed the simulations and analysis. All authors contributed to the overall model coupling strategy and to the manuscript.

*Competing interests.* The authors declare no competing interests.

*Acknowledgements.* Parts of this research were conducted using the supercomputer Mogon and/or advisory services offered by Johannes Gutenberg University Mainz (hpc.uni-mainz.de), which is a member of the AHRP (Alliance for High Performance Computing in Rhineland Palatinate, www.ahrp.info) and the Gauss Alliance e.V. Further development and the main simulations were performed using the LOEWE-CSC supercomputer at the High Performance Computing initiative. The authors gratefully acknowledge the computing time granted on the supercomputer Mogon at Johannes Gutenberg University Mainz (hpc.uni-mainz.de) and the LOEWE-CSC supercomputer at Goethe University Frankfurt (csc.uni-frankfurt.de).

The authors acknowledge the support of the MESSy core development team and are grateful for hints and discussions. Similarly, the authors recognise and appreciate the many improvements to LPJ-GUESS by the LPJ-GUESS development team which made this work possible, and thank the team (particularly Johan Nord) for their support. We also thank and acknowledge Allan Spessa for his contributions at the conception stage of this project.

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
