# Peer review of "Including vegetation dynamics in an atmospheric chemistry-enabled GCM: Linking LPJ-GUESS (v4.0) with EMAC modelling system (v2.53)"

_Geoscientific Model Development, 2018_

## Referee Comment (RC1) · Anonymous Referee #1 · 9 Oct 2018

Review of the article "Towards an advanced atmospheric chemistry-enabled ESM with dynamic land surface processes: Part I - Linking LPJ-GUESS (v4.0) with EMAC modelling system (v2.53) by Forrest et al. 9. Oct. 2018

General comments The article describes a one-way coupling of the vegetation model LPJ-GUESS with the ECHAM5 atmospheric model that is implemented in the modelling system EMAC. Several aspects of the resulting simulated vegetation are displayed and evaluated to be in good agreement with observations. Additionally, it is pointed out that this is a first important step on the way to build an Earth System Model

(ESM) including both atmospheric chemistry as well as dynamic vegetation. I appreciate this initiative as this ESM will be a very helpful tool to approach many important scientific questions like those listed in the abstract and the introduction. In view of the large effort it takes to construct such an ESM it is also appropriate to report already the first development steps to the modelling community. Also the text is well written. There are only two aspects, which (in my opinion) should be improved in the manuscript before publication. First, it should be clear from the title and the abstract that no detailed plan to construct an ESM nor any results based on an ESM are presented and that the only content of the article is the coupling of LPJ-GUESS to EMAC and the evaluation of the resulting vegetation. The reader is confused by the structure of the abstract. At the beginning of the abstract ESMs are explained, then it's mentioned that the coupling of LPJ-GUESS to EMAC is presented, then the development of ESMs is motivated that include dynamic vegetation and atmospheric chemistry, to finish with a sentence that simulated vegetation patterns are in agreement with observations. It would be much more straightforward to describe the contents of the paper first and then to motivate this work or the other way around, but not to mix both aspects in the abstract.

Second, the description of the coupling is incomplete in some aspects. LPJ-GUESS has a daily time step. This should be mentioned in section 2.2 and not only in the appendix. As the atmosphere model resolves the daily cycle, I guess, EMAC is building daily averages at the end of the simulation day and then passing it to LPJ-GUESS. Please describe this. What does it mean in terms of photosynthesis and stomatal conductance? These variables have a strong daily cycle, depend on each other, and also depend on the daily cycle in atmospheric conditions, but they are calculated by LPJ-GUESS on a daily basis. What variables are passed from EMAC to LPJ-GUESS? Precipitation, snow, solar insolation (split in visible and NIR?), wind?, temperature (surface temperature, 2m temperature, temperature of the lowest atmosphere level?), atmospheric humidity?, etc. A list of variables could easily be added to section 2.3 and would give the reader much more insight, what coupling of LPJ-GUESS to an atmosphere model means. The paper shows results for a one-way coupling. No information

from LPJ-GUESS are used in the calculations within EMAC during the simulation. That means, that the atmosphere model still needs the old land surface representation (in particular for the surface energy balance calculation). Please mention this. Generally, I think, a diagram illustrating the data flow between EMAC and LPJ-GUESS (also in terms of output) would be very helpful.

Specific comments Please omit "advanced" in the title. It is not explained in the article in what manner the atmosphere chemistry model is advanced. Replace in the title "land surface processes" by "vegetation". The article is only concerned with vegetation. Many other dynamic land surface processes that are relevant in Earth System modelling (as lakes/wetlands, permafrost, erosion, hydrological discharge) are not mentioned. page 1 line 10: please skip "fully". I don't believe that really everything in your surface description is computed porgnostically and nothing is prescribed (e.g. hydrological soil properties etc.). page 5 line 10: What is the chemical input from EMAC to the vegetation model? Constant CO2, constant N deposition? Perhaps it's better to specify it as the atmospheric chemistry model in EMAC is not used for this study. page 5 line 12: What is SMIL? page 6 line 2: Why do you kill the vegetation in the spin-up run? What does this mean? page 6 line 26: It would be nice to have some more information about NME scores. page 7 line 12: Here it is speculated that a bias in vegetation is caused by a precipitation bias in EMAC. Please mention, how large this precipitation bias in EMAC is (with respect to the bias corrected CRUNCEP data). page 7 line 30: Here it is speculated that imperfect process representation in LPJ-GUESS is responsible for a bias in tree cover over North Africa. Please, mention again, if there is also a precipitation bias of EMAC in this region. page 9 line 2: Again speculation. Here about N limitation in the tropics causing a bias in biomass. Do you have a nitrogen limitation factor? Is it possible to prove the N limitation in your model output? page 13 line 19: I'm very sceptical about a reduction of climate biases by vegetation dynamics. In most cases/regions there seems to be a positive feedback between climate and vegetation. That means, that climate biases will be enhanced by switching on vegetation dynamics.

Technical corrections page 1 line 9/10: "Then it" instead of "At this point, the full model" to shorten the abstract page 2 line 32: "a fully" instead of "an fully" page 3 line 4: "resulting damage" instead of "resulting to damage" page 5 line 11: "not affect the climate" instead of "not effect the climate" page 7 line 27" "are the climate" instead of "is the climate" page 8 line 9: "to ensure" instead of "to be ensure" page 11 line 12: "crown area of trees at 50 m2 in LPJ-GUESS" instead of "crown area of trees LPJ-GUESS is 50 m2" page 11 line 28: "affect" instead of "effect" page 13 caption of Tab.1: "the vegetation simulated" instead of "the vegetated simulated" page 13 line 10: "that in this" instead of "that this" page 13 line 10: What is the meaning of PNV? page 13 line 13: "utilising the recently" instead of "utilising a the recently" page 13 line 25: "bidirectionally" instead of "bi-bidirectionally" page 14 line 28: "Creation of three" instead of "Creation of a three" page 16 line 23: "biomass due to" instead of "biomass to" page 16 line 27: "was defined by the following classes in the Globcover 2009 dataset" instead of "was defined as classes"

---

## Referee Comment (RC3) · Anonymous Referee #2 · 9 Nov 2018

First, apologies for this last-minute review; I appreciate that it doesn't allow much time for online discussion, but other commitments prevented an earlier response. Referee #1 has made a number of very good points and I agree with all, although I have bigger difficulties with many aspects of this paper. I here only give additional comments to those made by Ref #1.

[Figure]

Major comments

1. As far as I can tell, the authors are running a non-bias corrected GCM together with a version of LPJ-GUESS that uses pre-industrial nitrogen levels, and only considers 'natural' vegetation (i.e. something is non-existent across large parts of the globe). It is mentioned that the GCM has temperature and precipitation biases (though little information is given), and of course it also has biases in other climate variables. The abstract concludes that 'initial results show that the one-way, on-line coupling from EMAC to LPJ-GUESS gives a good description of the global vegetation patterns ...'.

   If a vegetation model which predicts artificial vegetation is fed with wrong data and anyway gives a good description of global vegetation patterns, isn't something seriously wrong? Or is the comparison just not very discerning?

2. The paper states that a human land use and agricultural framework is included in LPJ-GUESS, but not enabled in this study. I cannot understand this. The authors are all from Europe and all the vegetation and land-cover they can see is affected by humans.

3. Similarly, N deposition rates are set for the decade 1850-1859, and seem to be kept constant. Given that nitrogen is a key nutrient, that values have changed enormously since the 1850s, and that LPJ-GUESS can account for this, why proceed with such an artificial assumption?

4. Much of the paper is vague about biases in EMAC and their importance. The authors explain (p5, L21) that 'it is expected that such biases will be reduced at higher spatial resolutions', but no evidence or quantification is provided. This is a serious weakness of the paper, and surprising as I most model groups know the biases of their GCMs pretty well these days.

[Figure]

5. On p6 we read that the simulations correspond loosely to the last couple of decades, in order to gain some insight into biases that may be present when LPJ-GUESS is forced by EMAC climate. But LPJ-GUESS runs over centuries, so how can results from a 20 year simulation (which used constant SST) give much insight into anything?

6. Actually, the LPJ-GUESS setup as given on p6, L1-5, is confusing. Here we read about a 400 year run after the vegetation has been killed off, and with nitrogen limitation accounted for. Does this mean the authors used the N-deposition of the 1850s across some period from 1600 to 2000? I think human populations have increased by more than a factor of 10 over this period, and N-deposition should reflect this to some extent.

   And which meteorology was used for the 100+400 years of simulation. Was this the constant SST, non-bias corrected EMAC, or was it CRU? When trying to interpret e.g. Fig 1 or indeed all results I really missed this information.

7. The fair evaluation of LPJ-GUESS would have been in its 'offline' mode, driven by CRU data. These results should also have been presented in Figs. 1-3, so we see how much influence 20 years of EMAC has on the simulations.

Although I appreciate that GMD is a place to report interim results, I am left with the feeling that this particular work is premature. I think the authors should run their model setup with the various anthropogenic impacts enabled (since they seem to have this capability), and they should properly account for GCM biases, before they compare with today's vegetation maps. Given that they seem to have all the model pieces in place, I cannot see why this wasn't done. And they need to compare LPJ-GUESS+EMAC with LPJ-GUESS+CRU in order to get a better sense of where discrepancies in vegetation cover and characteristics are coming from.

The whole manuscript also needs to be tightened up, with evidence offered in place of speculation. The example above, where impacts of resolution on a GCM were 'expected' is a good example. GCM modellers should know and demonstrate such results, not rest upon guesswork.

Other comments

p2, L11. It would be good to mention some of the key cycles that 'dynamic' vegetation models often lack too, e.g. not all have N-cycle, and few have P-cycles.

p2. Seems strange not to mention the EC-Earth ESM, which seems to have come much further in linking LPJ-GUESS inside an ESM model. Are there any links between the work described in this paper and the EC-Earth efforts? What are the similarities and differences in the approaches?

p3, L11 - refers to a 'companion' publication. As no real reference is given I assume they mean 'future' publication? In my experience these sometimes never appear (even if high-priority), and if one cannot already present an author list and title that can be cited I would re-phrase.

p3, L27. The phrase 'tree-individual' sounds odd and is not helpful. Re-phrase.

p4, L22. Why 'de facto'. Aren't all components of LPJ-GUESS or EMAC de facto components?!

p15, L15. Shouldn't you say 'This will extend' rather than 'This extends'? If the model is already a full ESM I don't see why you are reporting on the very limited and artificial setup you have here.

p5, L26-27. Why keep a constant CO2 when feeding a vegetation model; the seasonal cycle is well known and documented. Any why 367 ppm? Values are over 400 ppm these days.

p6. Also, isn't LPJ-GUESS a stochastic model? If so, how was the randomness of the results accounted for?

p6. Define and give references for CRUNCEP.

p7, L8. The authors conclude that 'the simulations reproduce the global patterns of vegetation cover well'. There are several point here. Firstly, the Sahara is largely missing, and that is rather a big deal. Secondly, I guess these patterns are mainly determined by the 100+400 years of simulation, rather than the last 20 years, but as noted above I don't understand which climate driver was behind these 500 years.

p7 and elsewhere. The authors sometimes say 'conservation remapping', sometimes 'conservative remapping', and neither version is explained.

p7, L27. Here it states 'The second source of disagreement is the climate biases in the EMAC derived climate, most obviously the underestimation of tree cover ...'. Usually one evaluates climate biases with reference to a temperature data set, not by looking at tree-lines. Again, I really miss any quantification of the EMAC errors going into this simulation, and without that I have no bases to judge the impact of LPJ-GUESS coupling.

p8, L2. The authors claim that biomass isn't directly relevant for land-atmosphere exchanges, but useful for evaluating DGVM performance. Well, canopy height is mentioned, and LAI (and hence BVOC and deposition parameters) could have been, but isn't biomass also one of the key outputs of ESMs? They are supposed to account for C-sequestration, NPP, etc. It is essential that an ESM can predict these outputs very well, but here they seem to be forgotten.

p11. Canopy height is here evaluated, but N-availability is a key driver for this, and here the N-deposition component is from the 1850s.

p11, L25-29. Here again I am not sure what to make of the paper and the setup. For biomass inclusion of a land use correction makes the results worse, but the authors say

that 'this is not a major concern .. as biomass does not directly affect the atmosphere'. How can they say this? Biomass is directly linked to water flows, energy balances, LAI, BVOC emissions, deposition rates, canopy height, momentum exchange, vegetation extent and a host of related parameters. A failure to model biomass reflects a failure to model the vegetation.

p11. Again the authors suggest that LPJ-GUESS needs to be changed to perform better, but since offline simulations perform better (p7, L10), I would look to EMAC first. (I wonder if any co-authors from Lund, Potsdam or Jena would have made the same conclusions!)

p12. The text states that 'scores improve when moving to a higher resolution implies that .. leads to a tangible increase in model performance'. Again, I have trouble with the loose arguments. A change in GCM resolution will result in a change in GCM performance and biases. There is no need to 'imply'. The EMAC bias results should have been presented and analysed to establish problems with EMAC, and the offline LPJ-GUESS results should have been presented as the only true benchmark against which the linking can be assessed.

(This sentence was also rather circular by the way. Higher scores implies better performance? I though that that was definition of the score?)

---

## Author Comment (AC1) · 7 Dec 2018

**Author's response to reviewer comments:**

**"Towards an advanced atmospheric chemistry-enabled ESM with dynamic land surface processes: Part I – Linking LPJ-GUESS (v4.0) with EMAC modelling system (v2.53)"**

by

Matthew Forrest et al.

We thank the reviewers giving time to review the manuscript and for their insightful comments which will significantly improve the manuscript. Here we reproduce the reviewers' comments in full and address them in turn. The reviewers' comments are in black, our responses are in light blue. We include proposed alterations to the manuscript to address the reviewers concerns in green. Page and line numbers refer to the original manuscript.

**Reviewer 1**

Review of the article "Towards an advanced atmospheric chemistry-enabled ESM with dynamic land surface processes: Part I - Linking LPJ-GUESS (v4.0) with EMAC modelling system (v2.53) by Forrest et al.

General comments

The article describes a one-way coupling of the vegetation model LPJ-GUESS with the ECHAM5 atmospheric model that is implemented in the modelling system EMAC. Several aspects of the resulting simulated vegetation are displayed and evaluated to be in good agreement with observations. Additionally, it is pointed out that this is a first important step on the way to build an Earth System Model (ESM) including both atmospheric chemistry as well as dynamic vegetation.

I appreciate this initiative as this ESM will be a very helpful tool to approach many important scientific questions like those listed in the abstract and the introduction. In view of the large effort it takes to construct such an ESM it is also appropriate to report already the first development steps to the modelling community. Also the text is well written. There are only two aspects, which (in my opinion) should be improved in the manuscript before publication.

First, it should be clear from the title and the abstract that no detailed plan to construct an ESM nor any results based on an ESM are presented and that the only content of the article is the coupling of LPJ-GUESS to EMAC and the evaluation of the resulting vegetation. The reader is confused by the structure of the abstract. At the beginning of the abstract ESMs are explained, then it's mentioned that the coupling of LPJ-GUESS to EMAC is presented, then the development of ESMs is motivated that include dynamic vegetation and atmospheric chemistry, to finish with a sentence that simulated vegetation patterns are in agreement with observations. It would be much more straightforward to describe the contents of the paper first and then to motivate this work or the other way around, but not to mix both aspects in the abstract.

Very good point, the abstract structure should be simplified and perhaps we over-sold the ESM aspect of the work. We propose a new and simpler title:

Including vegetation dynamics in an atmospheric chemistry-enabled GCM: Linking LPJ-GUESS (v4.0) with EMAC modelling system (v2.53)

And we have reformulated the abstract (also taking into account other comments) as follows:

"Central to the development of Earth System Models (ESMs) has been the coupling of previously separate model types, such as ocean, atmospheric and vegetation models, to provide interactive feedbacks between the system components. A modelling framework which combines a detailed representation of these components, including vegetation and other land surface processes, enables the study of land-atmosphere feedbacks under global change. This includes the methane cycle and lifetime and the atmospheric chemistry of reduced carbon; fire effects and feedbacks; future nitrogen deposition rates and fertilisation scenarios; ozone damage to plants; and the contribution of biogenic volatile organic compounds to aerosol load and, via cloud condensation nuclei activation, to cloud formation (e.g., precipitation cycles). Here we present the initial steps of coupling LPJ-GUESS, a dynamic global vegetation model, to the atmospheric chemistry enabled atmosphere-ocean general circulation model EMAC. The LPJ-GUESS framework includes a comparatively detailed individual based model of vegetation dynamics, a crop and managed-land scheme, a nitrogen cycle and a choice of fire models; and hence represents many important terrestrial biosphere processes and provides a wide range of prognostic trace gas emissions from vegetation, soil and fire. When development is complete, these trace gas emissions will form key inputs to the state-of-art atmospheric chemistry representations in EMAC allowing for bi-directional chemical interactions of the surface with the atmosphere. Then the full model will become a powerful tool for investigating land-atmosphere interactions. Initial results show that the one-way, on-line coupling from EMAC to LPJ-GUESS gives a reasonable description of the global potential natural vegetation distribution and reproduces the broad patterns of biomass, tree cover and canopy height when compared to remote sensing datasets. Based on this first evaluation, we conclude that the coupled model provides a suitable means to simulate dynamic vegetation processes into EMAC."

Second, the description of the coupling is incomplete in some aspects. LPJ-GUESS has a daily time step. This should be mentioned in section 2.2 and not only in the appendix. As the atmosphere model resolves the daily cycle, I guess, EMAC is building daily averages at the end of the simulation day and then passing it to LPJ-GUESS. Please describe this. What does it mean in terms of photosynthesis and stomatal conductance? These variables have a strong daily cycle, depend on each other, and also depend on the daily cycle in atmospheric conditions, but they are calculated by LPJ-GUESS on a daily basis.

Yes, we agree that this should be described, and yes, EMAC builds daily averages which are passed to LPJ-GUESS. We propose to add the following text as new paragraph after line 20 on page 4.

"Photosynthesis, respiration and hydrological processes operate on a daily time step and require daily temperature, precipitation and incident short wave radiation. However, monthly climate data may be provided, in which case the model interpolates daily values from the monthly values. In these circumstances, the number of precipitation days in the monthly periods may also be provided to disaggregate total precipitation into distinct rain events. In the case of unmanaged natural vegetation (as simulated here), vegetation dynamics (such as establishment and mortality),

disturbance, turnover of plant tissues and turnover between litter pools, and allocation of carbon and nitrogen to plant organs all occur on an annual basis."

What variables are passed from EMAC to LPJ-GUESS? Precipitation, snow, solar insolation (split in visible and NIR?), wind?, temperature (surface temperature, 2m temperature, temperature of the lowest atmosphere level?), atmospheric humidity?, etc. A list of variables could easily be added to section 2.3 and would give the reader much more insight, what coupling of LPJ-GUESS to an atmosphere model means.

Yes, we should explicitly list the variables and also describe how they are calculated (daily averages). We propose to replace the paragraph from line 10 to 17 on page 5 with the following paragraphs:

"To provide appropriate climate forcing for LPJ-GUESS, EMAC calculates the daily mean 2 m temperature, daily mean net downwards shortwave radiation and the total daily precipitation at the end of the simulation day and provided it to LPJ-GUESS.  Atmospheric $CO_2$ concentration and nitrogen deposition are also provided on a daily basis from EMAC to LPJ-GUESS.   Thus the LPJ-GUESS land-surface state is forced completely by the EMAC atmospheric state.

In turn, LPJ-GUESS provides fractional vegetation cover, leaf area index, daily net primary productivity and average height of each PFT to EMAC.  Parameterisations for determining albedo and roughness length are implemented in EMAC, however they are not enabled in the simulations presented here.  Thus, we are demonstrating only a one-way coupling where the land surface state does not affect the atmospheric state.  The boundary conditions for the atmospheric model (in particular the surface energy and water fluxes) come from the pre-existing land surface representation.

The overall strategy is to tighten the coupling between LPJ-GUESS and EMAC in well-defined, consecutive steps and to assess the effects of one model on the other in a step-wise and logical manner. Here we report the effect of EMAC climate on LPJ-GUESS.  The next step is to enable the albedo and roughness length schemes and to use the vegetation and forest fractions (which are used in the standard land surface scheme to determine the hydrological fluxes) to form a bidirectional coupling of interactive vegetation and climate.  The work is underway (Tost et al. 2018) and will be presented in a future publication.

Future planned development steps are to enable land use and agriculture in LPJ-GUESS within EMAC, to include a more process-based representation of fire and include the relevant emissions, to fully replace the soil-vegetation part of the hydrological cycle in EMAC with that in LPJ-GUESS and to use LPJ-GUESS to close the land surface energy balance.  When completed, these developments will extend the EMAC model into a full Earth system model including atmosphere (ECHAM5) with comprehensive chemistry (see Jöckel et al., 2010), vegetation and land surface processes (LPJ-GUESS) and an ocean component (MPIOM) (see Pozzer et al., 2011) with ocean biogeochemistry (see Kern, 2013). "

New reference:

Tost, H., Forrest, M., and Hickler, T. (2018) Interactive vegetation influences on climatological meteorological fields and trace gas emissions vol. 20, p. 12047, http://adsabs.harvard.edu/abs/2018EGUGA..2012047T.

The paper shows results for a one-way coupling. No information from LPJ-GUESS are used in the calculations within EMAC during the simulation. That means, that theatmosphere model still needs the old land surface representation (in particular for the surface energy balance calculation). Please mention this. Generally, I think, a diagram illustrating the data flow between EMAC and LPJ-GUESS (also in terms of output) would be very helpful.

Yes, the reviewer is right, information from LPJ-GUESS is not used by EMAC. We explicitly state this in the revised text for section 2.3 as described in the answer to the previous point. Further follow the suggestion that a diagram would be useful and propose to include we include the following figure:

[Figure]

[Figure]

"**Proposed New Main Text Figure 1**: The main processes and exchanges in the coupled model framework. Processes/exchanges with normal black text/black solid arrows are included in the framework and used in the simulations presented here; processes/exchanges with normal grey text/grey solid arrows are included in the framework but not used in the simulations presented here; and processes/exchanges with italic grey text/grey dotted arrows are not included in the framework but planned in future work. All exchanges happen on a daily basis, except for soil properties which happen only during the initialisation phase."

And we propose to refer to the diagram with the following text at the end of section 2.3 (overview of coupling implementation):

"The process and exchanges currently included in the modelling framework, as well as planned future additions, are shown in Fig **Proposed New Main Text Figure 1.**"

Specific comments

Please omit "advanced" in the title. It is not explained in the article in what manner the atmosphere chemistry model is advanced.

Done (see revised title above).

Replace in the title "land surface processes" by "vegetation". The article is only concerned with vegetation. Many other dynamic land surface processes that are relevant in Earth System modelling (as lakes/wetlands, permafrost, erosion, hydrological discharge) are not mentioned.

Done (see revised title above).

page 1 line 10: please skip "fully". I don't believe that really everything in your surface description is computed porgnostically and nothing is prescribed (e.g. hydrological soil properties etc.).

Done (see revised abstract above).

page 5 line 10: What is the chemical input from EMAC to the vegetation model? Constant CO2, constant N deposition? Perhaps it's better to specify it as the atmospheric chemistry model in EMAC is not used for this study.

Done, our revised text for section 2.3 explicitly discussed the chemical input ($CO_2$ and N dep) and states that they are provided daily. We propose to insert the following text at line 25 on page 5 to make it clear that atmospheric chemistry was not fully enabled:

"Note that for reasons of computational burden, the atmospheric chemistry calculations of which EMAC is capable were not activated in these simulations."

page 5 line 12: What is SMIL?

SMIL stands for 'Submodel interface layer', a component of MESSy. The reviewer was correct to flag it, this rather technical jargon has no place in the main text and has been removed during the reformulation of section 2.3.

page 6 line 2: Why do you kill the vegetation in the spin-up run? What does this mean?

LPJ-GUESS starts from 'bare-ground' i.e. no vegetation and no soil C or N pools. But to have vegetation in an N limited model one requires plant available N, it is a bit of a chicken-and-egg situation. To overcome this, the model is run without N limitation (so that vegetation can grow) but with N deposition (to allow N pools to accumulate) for 100 years. This builds up reasonable N pools, but the vegetation has not been limited by N during its development and so is not what we want. Therefore, we 'kill' the vegetation by removing it and putting its C and N into the soil pools and start the vegetation again, this time with N limitation and pre-existing N pools.

To explain this, we propose to change the sentence which runs from line 1 to 3 of page 6 with the following fuller and hopefully clearer explanation:

"Here we followed the standard LPJ-GUESS procedure of starting with 'bare ground', ie. no vegetation and no C or N in the soil and litter pools, and running for approximately 500 years to allow the vegetation to reach equilibrium. Having no plant available N present in the soil at the start of the simulation would inhibit and distort vegetation growth if N limitation was enabled. To overcome this, we follow the standard protocol and run LPJ-GUESS for 100 years without N

limitation but with normal N deposition to build up the N pools. After 100 years there is sufficient N in the pools, but the vegetation is inconsistent with the desired state as it has been growing without N limitation. Therefore, the vegetation is removed (and the C and N put into the litter pools), and the vegetation is allowed to regrow, this time with N limitation enabled, for a further 400 years.

page 6 line 26: It would be nice to have some more information about NME scores.

Yes, this a good point, particularly as the meaning of NME scores are inverted compared to $r^2$ values, ie. an $r^2$ value of zero means no correlation or explanatory power, but an NME of zero means perfect agreement between model and observation. We propose to insert the following text at page 6 line 27.

"It should be noted that the NME is rather different from a coefficient of correlation or a coefficient of determination. It does not attempt to derive a correlation but instead sums the differences between the model and the observation. It can be thought of as quantifying the deviation from the one-to-one line of perfect data-model agreement, rather than the deviation from a line of best fit. This means that is a rather direct and unforgiving metric, since every deviation of the model from the data is penalised (uncertainty is not included) and there is no possibility for the line of best fit to move to compensate for systematic biases. It also means the values are interpreted in the opposite direction to a correlation coefficient; an NME score of zero implies perfect agreement between observation and model, whereas an $r^2$ of zero would imply no correlation between the two. By the normalisation implicit in the method, using the mean value of the observations in place of the model gives an NME of unity."

page 7 line 12: Here it is speculated that a bias in vegetation is caused by a precipitation bias in EMAC. Please mention, how large this precipitation bias in EMAC is (with respect to the bias corrected CRUNCEP data).

To answer this and other comments by both reviewers, we propose to include an appendix with plots showing the bias between the EMAC produced climate and the CRUNCEP corrected climate. We would like to stress that this is not intended to be a thorough investigation of the biases in EMAC, but rather supporting information to enable a more concrete and less speculative interpretation of our results. In particular, the bias in the net shortwave radiation quantifies the difference in radiation available to the plants using the relevant albedo values, not the gross flux.

Furthermore, we now propose to include results from an 'offline' LPJ-GUESS simulation driven by the CRUNCEP data (but using the same code and settings as the EMAC simulations) following the recommendation of reviewer 2. This also enables a better attribution data-model disagreement in the EMAC simulations. These simulations are referred to as 'CRUNCEP' in the following proposed text.

We propose to include the following bias climate plots:

[Figure]

a)

[Figure]

b)

**New Appendix Figure 1**. The a) mean annual precipitation bias and b) mean seasonal precipitation bias between the observed CRUNCEP (1981-2010) and EMAC simulations (last 50 years of simulation). Note that ensure visibility of relatively low precipitations biases, the plotted values are capped at 750 mm/season and 1500 mm/year in the seasonal and annual plots, respectively.

[Figure]

a)

b)

**Proposed New Appendix Figure 2.** The a) mean annual temperature bias and b) mean seasonal temperature bias between the observed CRUNCEP (1981-2010) and EMAC simulations (last 50 years of simulation).

[Figure]

a)

[Figure]

b)

**Proposed New Appendix Figure 3.** The a) mean annual net shortwave radiation bias and b) mean seasonal net shortwave radiation bias between the observed CRUNCEP (1981-2010) and EMAC simulations (last 50 years of simulation).  Note that these plots compare shows the radiation available for vegetation in the CRUNCEP and EMAC forced LPJ-GUESS simulations, and consequently the gross shortwave radiation has been adjusted by different albedo values.  The CRUNCEP radiation has been adjusted using the standard LPJ value of 0.17 applied (temporally and spatially invariant), and the EMAC radiation has been adjusted by the spatially and temporally varying albedo values in the land surface scheme.

We propose to include the following text to introduce these plots as a new final paragraph of section 4 (Model evaluation).

"Whilst it is not within the scope of this work to evaluate the biases of climate state produced by EMAC, knowledge of these biases is very useful for disentangling the causes of model-data disagreement in the simulated vegetation. To this end, we include bias plots of seasonal and annual biases in surface temperature, precipitation and net (plant-available) short wave radiation of the EMAC T42 and T63 climate with respect to the CRUNCEP bias-corrected, re-analysis climate dataset (Appendix B)."

To answer the particular point, we propose to replace the paragraph on page 7 from line 8-18 with the following:

"The simulations reproduce the global patterns of vegetation cover well (Fig. 1), although some regional discrepancies are visible. The most obvious mismatch between all the simulations and the reconstructed megabiomes is the underestimation of the abundance of vegetation in the high northern latitudes. The tendency is relatively small in the *CRUNCEP* simulation, but larger for the EMAC simulations. The higher resolution *T63* simulation is better than the *T42*, as it shows a greater tundra and boreal forest extent. Therefore, this mismatch can be attributed to a high-latitude growing season low temperature and low plant available radiation bias in the EMAC climate (**Proposed new appendix figures 2 and 3**) at low resolution, which is somewhat mitigated at higher resolution due to a better representation of the synoptic scale systems in *T63* (Roeckner et al., 2006).

The extent of the temperate forest of the east coasts of the USA and China was underestimated in the simulations using the EMAC climate, which is not seen in the *CRUNCEP* simulations. Inspection of the climate bias plots (**Proposed new appendix figures 1-3**) shows that this can be attributed to a negative bias in precipitation in the southern areas, and negative bias in the plant available radiation in the northern area of coastal China. Large negative precipitation biases reduce the extent of the tropical forest in Indonesia and Papua New Guinea, and in the north-east coast of South America (not seen in the *CRUNCEP* simulations). In central Africa and the interior of Brazil, the extent of the tropical forests is also much reduced compared to *CRUNCEP*. However, in this case the reasons are not immediately clear from examination of the bias plots, although some seasonal biases in precipitation and plant available radiation are not clearly apparent. The extent of the Savanna and Dry Woodlands vegetation type is also under-simulated in Australia and eastern Africa as a result of a negative precipitation bias, although this extent is also not well represented in the *CRUNCEP* simulation. The extent of the Sahara Desert is also underestimated in the EMAC and *CRUNCEP* simulations, as some areas grow sufficient grass cover to be classified as short grasslands. As this is present in the *CRUNCEP* simulation, it is not related to a climate biases, but rather the over-simulation of grasses in very arid regions by LPJ-GUESS.

Given that the *T42* and *T63* results are broadly similar, for brevity of the presentation the subsequent *T42* results will be omitted from the subsequent spatial benchmarks, although the *T42* summary metric results will be tabulated and discussed."

page 7 line 30: Here it is speculated that imperfect process representation in LPJ-GUESS is responsible for a bias in tree cover over North Africa. Please, mention again, if there is also a precipitation bias of EMAC in this region.

It can be seen in the plots above that the precipitation bias in this region is not large. Furthermore, in the new plot showing the tree cover bias in the CRUNCEP simulation, this bias is also observed. We therefore propose the change the last sentence on page 7 to read:

"A final source of disagreement is due to the inevitable imperfect process representations in LPJ-GUESS. This is exemplified by the over-estimation of tree cover north of the central African tropical forest in both the EMAC and *CRUNCEP* forced simulations."

page 9 line 2: Again speculation. Here about N limitation in the tropics causing a bias in biomass. Do you have a nitrogen limitation factor? Is it possible to prove the N limitation in your model output?

The newly included CRUNCEP simulations shows that LPJ-GUESS with observed climate underestimates tropical biomass, but to a much smaller extent. Furthermore, is it possible to determine the N limitation from our model output. To answer this point, and the point of reviewer two, we propose a new figure to be put in an appendix. We propose the following figure, which show the nitrogen limitation as the ratio of the N limited photosynthetic rate ($V_{c,max}$) to the non-N limited photosynthetic rate (weighted across the vegetation in a gridcell by leaf biomass).

[Figure]

**Proposed New Appendix Figure 4.** The ratio of nitrogen limited to nitrogen non-limited photosynthetic rates (weighted by leaf biomass) in all simulation runs.

This figure shows that in fact N limitation does not appear to be any more limiting in the tropics in the EMAC simulations than in the CRUNCEP simulations. We propose to amend the speculative (and incorrect) sentence on page 9 line 2 to read:

"This underestimation is seen to a lesser extent in the *CRUNCEP* simulation. The different degrees of underestimation of tropical biomass by the *CRUNCEP* and EMAC *T63* simulations cannot be explained by the different nitrogen deposition forcing data (**Proposed New Appendix Figure 4**), however negative seasonal biases in plant available radiation and precipitation are visible in these regions (**Proposed New Appendix Figure 1,3**)."

page 13 line 19: I'm very sceptical about a reduction of climate biases by vegetation dynamics. In most cases/regions there seems to be a positive feedback between climate and vegetation. That means, that climate biases will be enhanced by switching on vegetation dynamics.

We understand the reviewer's point, but we argue that a bias can be reduced by a positive feedback, in the case that bias was caused by the underestimation of a particular process or interaction. For example, a low precipitation bias could be reduced by a positive feedback on the process of precipitation cycling by vegetation.

None-the-less, we accept that our wording was perhaps naïve and over-optimistic, and propose to change the text on page 13, lines 18-20 to use more neutral language and to read:

"Furthermore, using dynamically simulated land surface boundary conditions (in this case from LPJ-GUESS) in a bidirectionally coupled model will alter the atmospheric state and therefore the climate biases. This will be the subject of future studies."

Technical corrections

page 1 line 9/10: "Then it" instead of "At this point, the full model" to shorten the abstract

Done.

page 2 line 32: "a fully" instead of "an fully"

Done.

page 3 line 4: "resulting damage" instead of "resulting to damage"

Done.

page 5 line 11: "not affect the climate" instead of "not effect the climate"

Done.

page 7 line 27" "are the climate" instead of "is the climate"

Done.

page 8 line 9: "to ensure" instead of "to be ensure"

Done.

page 11 line 12: "crown area of trees at 50 m2 in LPJ-GUESS" instead of "crown area of trees LPJ-GUESS is 50 m2"

page 11 line 28: "affect" instead of "effect"

Done.

page 13 caption of Tab.1: "the vegetation simulated" instead of "the vegetated simulated"

Done.

page 13 line 10: "that in this" instead of "that this"

Done.

page 13 line 10: What is the meaning of PNV?

Potential Natural Vegetation. This acronym is defined on page 6, line 12. But we also now clarify the term in the conclusions (page 13 line 10) be saying:

"PNV (potential natural vegetation with no human impacts)"

page 13 line 13: "utilising the recently" instead of "utilising a the recently"

Done.

page 13 line 25: "bidirectionally" instead of "bi-bidirectionally"

Done.

page 14 line 28: "Creation of three" instead of "Creation of a three"

Done.

page 16 line 23: "biomass due to" instead of "biomass to"

Done.

page 16 line 27: "was defined by the following classes in the Globcover 2009 dataset" instead of "was defined as classes"

Done.

**Reviewer 2**

First, apologies for this last-minute review; I appreciate that it doesn't allow much time for online discussion, but other commitments prevented an earlier response. Referee #1 has made a number of very good points and I agree with all, although I have bigger difficulties with many aspects of this paper. I here only give additional comments to those made by Ref #1.

Major comments

1. As far as I can tell, the authors are running a non-bias corrected GCM together with a version of LPJ-GUESS that uses pre-industrial nitrogen levels, and only considers 'natural' vegetation (i.e. something is non-existent across large parts of the globe). It is mentioned that the GCM has

temperature and precipitation biases (though little information is given), and of course it also has biases in other climate variables.  The abstract concludes that 'initial results show that the one-way, on-line coupling from EMAC to LPJ-GUESS gives a good description of the global vegetation patterns ...'.

If a vegetation model which predicts artificial vegetation is fed with wrong data and anyway gives a good description of global vegetation patterns, isn't something seriously wrong? Or is the comparison just not very discerning?

Yes, we now realise that we over-stated our results and were also unclear in the intention of our comparison.  Our intention was not to show the we have a perfect description of the land surface, but rather to show that the implemented coupling works as expected and that forcing LPJ-GUESS with EMAC-produced climate gives sensible continental to global scale patterns in some key vegetation indicators for which global observations are available. Similarly, the inclusion of NME scores was not meant to say that we had sufficient agreement (or better agreement than other models), but instead to quantify the effects of changing resolution and accounting for human land use.

To rectify this, we propose the following changes. The sentence in the abstract flagged by the reviewer would read:

"Initial results show that the one-way, on-line coupling from EMAC to LPJ-GUESS gives a reasonable description of the global potential natural vegetation distribution and the reproduces the broad patterns of biomass, tree cover and canopy height when compared to remote sensing datasets."

We hope this clarifies that for the direct comparison (expert-reconstructed potential natural vegetation) we have reasonable results and that for other comparison (which, as you point, are not apples-to-apples because of the lack of human land use) we are satisfied with the broad patterns.

We also propose to include the following text at line 27 on page 6.

"These summary metrics are not meant to demonstrate a particular level agreement better than some arbitrary threshold.  Nor are they mean for strict evaluation or comparison to other models, since here we evaluating only the first milestone of model development and so the model is known to be incomplete (particularly with regard to human land use) and no tuning has been performed. They are included to quantitively evaluate the differences between models runs at difference resolutions and for assessing the effect of human land use via a land cover correction factor (see below)."

Furthermore, we are aware of land use issue and attempt to quantify it by including a land use correction in Table 1 (described in Appendix C).  In doing so, we attempted to show that if account for human land use (albeit in a very simple way which assumes that dominant effect of human land use is deforestation) then the agreement between the model and observations improves.

2.  The paper states that a human land use and agricultural framework is included in LPJ-GUESS, but not enabled in this study. I cannot understand this. The authors are all from Europe and all the vegetation and land-cover they can see is affected by humans.

The reviewer makes a good point.  As mentioned above, we are aware of the land use issue and attempt to quantify it with the land use correction.  The changes to the text described above attempt to acknowledge that the model is incomplete in this regard.  We absolutely agree that including land use and agriculture would be essential for scientific results derived from the model for

the present day, future or recent past (paleo applications would of course be different). However, for the proof-of-concept (or perhaps better said 'proof-of-functioning') demonstration of the model coupling presented here, we don't believe that this is necessary.

3. Similarly, N deposition rates are set for the decade 1850-1859, and seem to be kept constant. Given that nitrogen is a key nutrient, that values have changed enormously since the 1850s, and that LPJ-GUESS can account for this, why proceed with such an artificial assumption?

This was an unfortunate oversight on our part that we only noticed after the model runs were complete. We regret this. It was not a bug in the code exactly, just a mis-specification in the settings file. But similar to the land use point above, we don't believe that the use of pre-industrial N deposition data invalidates our proof-of-functioning, as the results are as expected. Of course, for future studies (including the evaluation of the two-way coupling) we will ensure that temporally-correct N deposition data is used.

To illustrate that using pre-industrial N deposition does not invalidate out results we propose to include a new figure to quantify the degree of N limitation. This figure shows the ratio of the N limited photosynthetic rate to the photosynthetic rate with limitation. We have introduced the figure in answer to a comment by reviewer one (please see **Proposed New Appendix Figure 4** above) and we propose to include this figure as an appendix plot in the revised manuscript.

To answer the reviewer's current point, this figure shows that the patterns of nitrogen limitation are broadly similar with both transient N deposition (CRUNCEP) and pre-industrial N deposition (T42, T63). In fact, in some regions the T63 and T42 simulations actually show less N limitation than the CRUNCEP simulations, showing that climatological factors (such as other constraints on growth and different N mineralisation rates due to different temperatures) have a larger effect than N deposition rates. Therefore, we can conclude that whilst nitrogen is critical for understanding and describing ecosystems, N deposition is not so critical that with only pre-industrial levels one would get wildly different or inaccurate results.

We intend to include the following test to briefly discuss this point and introduce the N limitation figure:

"The degree of nitrogen limitation experienced by the vegetation in the simulations (quantified by the ratio of nitrogen limited to non-nitrogen limited photosythetic rates) is shown in Fig **Proposed New Appendix Figure 4**, which shows that pre-industrial nitrogen deposition does not result in additional nitrogen limitation. However, in future work, temporally-appropriate nitrogen deposition data will be used."

4. Much of the paper is vague about biases in EMAC and their importance. The authors explain (p5, L21) that 'it is expected that such biases will be reduced at higher spatial resolutions', but no evidence or quantification is provided. This is a serious weakness of the paper, and surprising as I most model groups know the biases of their GCMs pretty well these days.

Yes, and we apologise for the vague and sloppy language. We propose to provide bias plots of seasonal temperature, precipitation and net shortwave radiation in an appendix. Please see our response to reviewer one for the bias plots (**New Appendix Figures 1-3**) and accompanying changes to the text.

5. On p6 we read that the simulations correspond loosely to the last couple of decades, in order to gain some insight into biases that may be present when LPJ-GUESS is forced by EMAC climate. But LPJ-GUESS runs over centuries, so how can results from a 20 year simulation (which used constant SST) give much insight into anything?

We apologise for the poor description here (as the reviewer also mentions in point 6.) To clarify here for the reviewers: throughout the simulation period (for all 500 years), LPJ-GUESS was driven by the climate variables that were being dynamically simulated by EMAC. As the reviewer points out, LPJ-GUESS is normally run for centuries and the procedure was no different here. The reference to the 'last couple of decades' was simply referring to the fact that we have SSTs from 1998, $CO_2$ concentration of 367 ppm (which also correspond to approximately 1998) and we are comparing to satellite data from which the observations were taken in the 2000s.

In order to clear up the confusion we propose the following changes. Addition of a sentence on page 5, line 27 which reads:

"Throughout both the simulations, LPJ-GUESS was driven exclusively by climate variables from EMAC, at no point were external climate datasets used."

Also, modification the sentence starting on page 6 line 21 to read:

"Instead, our goal with this evaluation is to perform steady state simulations where the climate and $CO_2$ forcing are constant and correspond approximately to conditions in the recent past. Thus, after 500 years of simulation, we can compare the equilibrium vegetation to satellite products based on observations in the early 2000s. Whilst we can't expect perfect agreement since (among other reasons) this is not a full transient simulation, the simulations should be sufficient to check if the model coupling is working as intended, and to gain some insight into biases that may be present when LPJ-GUESS is forced using EMAC climate."

6. Actually, the LPJ-GUESS setup as given on p6, L1-5, is confusing. Here we read about a 400 year run after the vegetation has been killed off, and with nitrogen limitation accounted for. Does this mean the authors used the N-deposition of the 1850s across some period from 1600 to 2000? I think human populations have increased by more than a factor of 10 over this period, and N-deposition should reflect this to some extent.

Yes, we realise that the description of the setup was incomplete. We have proposed revised text in answer to a comment from reviewer one which hopefully makes the details and logic of the spin-up clearer.

However, this alone does not answer all of the concern raised here by the reviewer. To answer the point about N-deposition, yes, it was constant at 1850s levels through the simulation. To explain this, we propose to extend the sentence on page 5 line 31 to read:

"Nitrogen deposition rates were prescribed using data from Lamarque et al. (2013) for the decade 1850-1859 throughout the *T42* and *T63* simulations."

We would also like to emphasise that the simulations presented here are intended to be proof-of-implementation, not transient simulations with the intention to reconstruct a particular time period with great accuracy. We hope that second piece of revised text to the reviewer's point 5 will make this clear in the manuscript. Viewed in this light, the 400 years of spin-up don't correspond to calendar years 1600-2000, but rather they are a simulation period required by the model to allow the modelled vegetation to come into equilibrium with the climate. In such a context, the exact amount of N deposition is not critical, but rather it is important that there is N deposition occurring with reasonable spatial patterns.

And which meteorology was used for the 100+400years of simulation. Was this the constant SST, non-bias corrected EMAC, or was it CRU? When trying to interpret e.g. Fig 1 or indeed all results I really missed this information.

Yes, we apologise that this important information was unclear. Throughout the 100+400 years constant SST, non-bias corrected EMAC meteorology was used. We have added an explicit statement of this fact as described in our answer to the review's point 5.

7. The fair evaluation of LPJ-GUESS would have been in its 'offline' mode, driven by CRU data. These results should also have been presented in Figs. 1-3, so we see how much influence 20 years of EMAC has on the simulations.

We apologise once more for the inexact language and see how the text may have misled the reader into believing that we intended to evaluate stand-alone LPJ-GUESS. This is categorically not our

intention, although we can see how the confusion can have occurred with following text at line 10 page 6:

"As LPJ-GUESS has been evaluated in detail in previous work, it is beyond the scope of this work to perform a full model evaluation and propose improvements for the dynamic vegetation model. Instead we performed basic evaluation using …"

This was poorly worded and would lead the reader to believe that we are evaluating standalone LPJ-GUESS. In fact, we want to stress that LPJ-GUESS has already been evaluated extensively and we want to evaluate the functioning of the coupled model system. To clear this up we propose the following text to replace the two sentences starting at page 6 line 10:

"As LPJ-GUESS has been evaluated in detail in previous work, for example, net primary production (e.g. Zaehle et al., 2005; Hickler et al., 2006), modelled potential natural vegetation (Hickler et al., 2006; Smith et al., 2014), stand-scale and continental-scale evapotranspiration (AET) and runoff (Gerten et al., 2004), vegetation greening trends in high northern latitudes (Lucht et al., 2002) and the African Sahel (Hickler et al., 2005), stand-scale leaf area index (LAI) and gross primary productivity (GPP; Arneth et al., 2007), forest stand structure and development (Smith et al., 2001, 2014; Hickler et al., 2004), global net ecosystem exchange (NEE) variability (Ahlström et al., 2012, 2015) and $CO_2$ fertilisation experiments (e.g. Hickler et al., 2008; Zaehle et al., 2014; Medlyn et al., 2015), it is beyond the scope of this work to perform a full model evaluation and propose improvements for the dynamic vegetation model. Instead we evaluated the coupled model setup to consider how LPJ-GUESS responds when EMAC climate is used as the forcing data and to investigate any biases in the vegetation produced. For this we used an expert-derived Potential Natural Vegetation (PNV) map and using remotely-sensed data sets of tree cover (Dimiceli et al., 2015), canopy height (Simard et al., 2011) and biomass (Avitabile et al., 2016; Thurner et al., 2014)"

New references:

Ahlström, A., G. Schurgers, A. Arneth, and B. Smith (2012) Robustness and Uncertainty in Terrestrial Ecosystem Carbon Response to CMIP5 Climate Change Projections. Environmental Research Letters 7(4): 044008.

Ahlström, Anders, Michael R. Raupach, Guy Schurgers, et al. 2015The Dominant Role of Semi-Arid Ecosystems in the Trend and Variability of the Land CO2 Sink. Science 348(6237): 895–899.

Arneth, Almut, Paul A. Miller, Marko Scholze, et al. (2007) CO2 Inhibition of Global Terrestrial Isoprene Emissions: Potential Implications for Atmospheric Chemistry. Geophysical Research Letters 34(18): L18813.

Hickler, Thomas, Benjamin Smith, Martin T. Sykes, et al. (2004) Using a Generalized Vegetation Model to Simulate Vegetation Dynamics in Northeastern Usa. Ecology 85(2): 519–530.

Hickler, Thomas, Lars Eklundh, Jonathan W. Seaquist, et al. (2005) Precipitation Controls Sahel Greening Trend. Geophysical Research Letters 32(21).

Hickler, Thomas, I. Colin Prentice, Benjamin Smith, Martin T. Sykes, and Sönke Zaehle (2006) Implementing Plant Hydraulic Architecture within the LPJ Dynamic Global Vegetation Model. Global Ecology and Biogeography 15(6): 567–577.

Hickler, Thomas, Benjamin Smith, I. Colin Prentice, et al. (2008) CO 2 Fertilization in Temperate FACE Experiments Not Representative of Boreal and Tropical Forests. Global Change Biology 14(7): 1531–1542.

Lucht, Wolfgang, I. Colin Prentice, Ranga B. Myneni, et al. (2002) Climatic Control of the High-Latitude Vegetation Greening Trend and Pinatubo Effect. Science 296(5573): 1687–1689.

Medlyn, Belinda E., Sönke Zaehle, Martin G. De Kauwe, et al. (2015) Using Ecosystem Experiments to Improve Vegetation Models. Nature Climate Change 5(6): 528–534.

Zaehle, S., S. Sitch, B. Smith, and F. Hatterman (2005) Effects of Parameter Uncertainties on the Modeling of Terrestrial Biosphere Dynamics. Global Biogeochemical Cycles 19(3): GB3020.

Zaehle, Sönke, Belinda E. Medlyn, Martin G. De Kauwe, et al. (2014) Evaluation of 11 Terrestrial Carbon–Nitrogen Cycle Models against Observations from Two Temperate Free-Air CO2 Enrichment Studies. New Phytologist 202(3): 803–822.

However, we feel that the reviewer makes an excellent point, there is no reason not show standalone LPJ-GUESS driven by CRUNCEP data (and the other standard input data and settings) and doing so would be illuminating. So, in addition, to the above clarification, we have performed such a simulation and propose to include it in the manuscript. To describe this simulation, we propose to replace the lines 6-7 on page 6 with the following text:

To aid the interpretation of the EMAC simulations, we also performed an 'offline' LPJ-GUESS simulation using observed climate data from the CRUNCEP bias-corrected, re-analysis dataset (Wei et al 2014). The simulation was performed using exactly the same code and parameter settings as the EMAC *T42* and *T63* simulations, but code was compiled as a stand-alone model. The spin-up procedure, $CO_2$ concentration and nitrogen deposition follow Smith et al. (2014). Note that this is full a transient simulation, after the 500 year spin-up, a further 113 years were simulated using CRUNCEP data. The plots presented here show model output averaged over the years 1981-2010 and are referred to simply as the *CRUNCEP* simulation.

New citation:

Wei, Y., Liu, S., Huntzinger, D. N., Michalak, A. M., Viovy, N., Post, W. M., Schwalm, C. R., Schaefer, K., Jacobson, A. R., Lu, C., Tian, H., Ricciuto, D. M., Cook, R. B., Mao, J., and Shi, X.: The North American Carbon Program Multi-scale Synthesis and Terrestrial Model Intercomparison Project – Part 2: Environmental driver data, Geoscientific Model Development, 7, 2875–2893, https://doi.org/https://doi.org/10.5194/gmd-7-2875-2014

We then propose to include the *CRUNCEP* simulation in Figures 1-4, as follows:

[Figure]

**Proposed Modified Figure 1:** Distribution of PNV megabiomes simulated by LPJ-GUESS within EMAC *T42* and *T63*) and with observed climate data (*CRUNCEP*) compared to an expert-derived PNV map.

[Figure]

a)

b)

**Proposed Modified Figure 2:** Comparison of tree cover from the *T63* (EMAC climate) and *CRUNCEP* (observed climate) simulations with observed tree cover from Dimiceli et al. (2015) a) absolute values and b) the difference between simulations minus observations.

[Figure]

a)

b)

**Proposed Modified Figure 3:** Comparison of biomass from the *T63* (EMAC climate) and *CRUNCEP* (observed climate) simulations with observed biomass from Avitabile et al. (2016) and Thurner et al. (2014), a) absolute values and b) the difference between simulations minus observation. Note that neither the Avitabile et al. (2016) nor the Thurner et al. (2014) biomass dataset provide biomass for a band across the Sahara, so no data are plotted there.

[Figure]

a)

b)

**Proposed Modified Figure 4:** Comparison of canopy height from the *T63* (EMAC climate) and *CRUNCEP* (observed climate) simulations with observed canopy height from Simard et al. (2011), a) absolute values and b) the difference between simulations minus observation.

We also propose to include the NME results for the *CRUNCEP* simulation in the results table, giving:

| Dataset | without LUC | | | with LUC | | |
|---|---|---|---|---|---|---|
| | *T42* | *T63* | *CRUNCEP* | *T42* | *T63* | *CRUNCEP* |
| Tree cover (Dimiceli et al., 2015) | 0.94 | 0.85 | 1.1 | 0.81 | 0.69 | 0.62 |
| Biomass (Avitabile et al., 2016; Thurner et al., 2014) | 0.7 | 0.8 | 0.76 | 0.67 | 0.7 | 0.56 |
| Canopy height (Simard et al., 2011) | n/a | n/a | n/a | 0.96 | 0.81 | 0.77 |

**Proposed Modified Table 1:** NME scores for the vegetation produced by the *T42*, *T63* and *CRUNCEP* simulations compared to three remotely-sensed global datasets both with and without a LUC (Land Use Correction) where applicable. Note that lower scores imply better agreement between simulation and observation.

Although I appreciate that GMD is a place to report interim results, I am left with the feeling that this particular work is premature. I think the authors should run their model setup with the various anthropogenic impacts enabled (since they seem to have this capability), and they should properly account for GCM biases, before they compare with today's vegetation maps. Given that they seem to have all the model pieces in place, I cannot see why this wasn't done. And they need to compare LPJ-GUESS+EMAC with LPJ-GUESS+CRU in order to get a better sense of where discrepancies in vegetation cover and characteristics are coming from.

The reviewer makes several important points here which will we address in turn.

We are disappointed that the reviewer feels that this work is premature.  We agree that we may have over-stated the degree to which model development is completed and will happily revise the text to further emphasize that the coupled-model presented here is only the first milestone on a development path which will lead to an ESM capable of addressing a broad range of research topics. However, the development work involved in combining these two modelling frameworks was a significant undertaking and included but was not limited to: factoring out the original land surface scheme code in EMAC, re-ordering the space and time loops in LPJ-GUESS , modifying the LPJ-GUESS code to allow restarts within the EMAC framework and at arbitrary points in time (not just the end of the year) whilst ensuring binary identical results between restarted and non-restarted simulations, and writing new input and return functions for LPJ-GUESS.  As part of achieving this milestone, it also seemed prudent to perform a broad-strokes evaluation to check that the resulting model was working as expected, to determine if the resulting vegetation state forms a suitable basis for which to modify the land surface properties of EMAC and identify any prominent discrepancies in the vegetation state compared to observations which may inform future development work.  Thus, we are making an LPG-GUESS – EMAC coupled system available that can be used and further developed by a larger community.  We believe that this development work, combined the first evaluation, is sufficient for publication as a GMD 'Development and technical paper' manuscript (to use GMD's own classification).

To better describe our progress down a larger roadmap we propose the following changes.

On page 2 line 16 change "To construct an ESM with dynamic vegetation and fire," to:

"To take the first steps towards constructing an ESM with dynamic vegetation, anthropogenic influences and fire,"

The reformulated abstract, including closing sentence:

"Based on this first evaluation, we conclude that the coupled model provides a suitable means to simulate dynamic vegetation processes into EMAC."

Furthermore, **Proposed New Figure 1** (see response to reviewer 1) and the accompanying text also attempts to make clear what has been done, and what is still to be done.

On the next point, we fully agree that anthropogenic impacts are critical when simulating and studying the Earth system, and including their impacts is essential for any scientific conclusions concerning the anthropocene. Of course, for paleo-applications, where anthropogenic impact is nil, the model would not require these inclusions. We fully intend to perform further development to integrate these processes into the EMAC modelling system via LPJ-GUESS. However, although the processes are included in LPJ-GUESS, enabling them with EMAC is non-trivial. The reasons are somewhat technical but in summary it is because the land use, land use transition and management data are provided to LPJ-GUESS via significantly different input streams than the other inputs, and passing this information through EMAC (and the associated re-gridding) requires significant amounts of further development. Performing a study with various anthropogenic impacts is undoubtedly an excellent and exciting idea but would require significant amounts of development and computing time (a full 500 year T63 simulation requires approximately 2 months on 144 CPU cores) and would amount to an entire study in itself. Such a study would go far beyond the scope of a 'Development and technical paper' manuscript presented here.

To answer the reviewer's comment about 'accounting for GCM biases', we point out (and apologise again for not make this clear) that in the manuscript we aim for a simple evaluation of the coupled system, of which GCM biases are an implicit part. Correcting for GCM biases is not the goal, although understand the origins of such biases will inform future development work. We agree that such biases are a critical issue and should be discussed more rigorously, and to this end we propose to include LPJ-GUESS+CRU simulations as suggested (see above).

The whole manuscript also needs to be tightened up, with evidence offered in place of speculation. The example above, where impacts of resolution on a GCM were 'expected' is a good example. GCM modellers should know and demonstrate such results, not rest upon guesswork.

We acknowledge that the speculation and imprecise language in the submitted draft is unacceptable. As well as including the climate bias plots and LPJ-GUESS+CRU plots suggested by the reviewers to allow discussion of quantified biases wherever possible, we also to propose to tighten up the manuscript as suggested by the reviewer by removing such speculation or supporting it with citations.

With regards to the example quoted above we propose to amend the speculative sentence on page 5 at line 21 to read:

"It is well-known that these biases are dependent on spatial resolution; see Roeckner et al. (2006) for a study of the biases at different resolutions in ECHAM5 (the GCM upon which EMAC was original based). Whilst it is not within the scope of this study to perform a detailed analysis of the biases in EMAC or their dependence on spatial resolution, the impact of the horizontal spatial resolution of the atmospheric simulation on the vegetation simulation is relevant."

Other comments

p2, L11. It would be good to mention some of the key cycles that 'dynamic' vegetation models often lack too, e.g. not all have N-cycle, and few have P-cycles.

Yes, good point. We propose to amend the sentence to read:

"However, whilst all ESMs by definition have a carbon cycle, not all have truly dynamic vegetation or a nitrogen cycle, fewer still have prognostic representations of fire or a phosphorous cycle."

p2. Seems strange not to mention the EC-Earth ESM, which seems to have come much further in linking LPJ-GUESS inside an ESM model. Are there any links between the work described in this paper and the EC-Earth efforts? What are the similarities and differences in the approaches?

Yes, whilst we cited studies involving both the EC-Earth global ESM and RCA-GUESS regional ESM, these initiatives merit further discussion in this context. We propose replace the sentence at page 2 line 21 which currently reads:

"Furthermore, LPJ-GUESS has already been used with atmospheric models, both global (Weiss et al., 2014; Alessandri et al., 2017) and regional (Wramneby et al.,2010; Zhang et al., 2014) "

With the text:

"Furthermore, LPJ-GUESS has already been used both a global ESM, EC-Earth (Weiss et al., 2014; Alessandri et al., 2017), and a regional ESM, RCA-GUESS (Wramneby et al.,2010; Smith et al., 2011; Zhang et al., 2014). In both modelling systems, climate variables and daily soil moisture from the atmospheric component and its land surface scheme are aggregated over one simulation day and provided to LPJ-GUESS (Weiss et al., 2014; Smith et al., 2011). In the EC-Earth framework, LPJ-GUESS provides only time-varying leaf area index (LAI) to the atmospheric component which initially only affected physiological resistance (Weiss et al., 2014). This link was recently extended to include effects on albedo, surface roughness length, soil water exploitable by roots and snow shading by vegetation (Alessandri et al., 2017). The land surface scheme in RCA-GUESS splits the gridcell surface into two tiles, one of forest and one herbaceous vegetation, and LPJ-GUESS is used to dynamically adjust the LAI within those tiles and relative fractional coverage of needle-leaved and broad-leaved trees in the forest tile. These LAI and fractional cover values affect albedo, surface roughness and heat fluxes in the land surface scheme (Smith et al., 2011).

In the work reported here we have adopted a broadly similar approach with regards to forcing LPJ-GUESS with daily aggregated climate fields from the atmospheric model, although daily soil moisture values are calculated by LPJ-GUESS and not the land surface. In the model version described here, LPJ-GUESS return LAI, vegetation cover fractions, canopy heights and net primary production to EMAC, which allows dynamic calculation of transpiration (by using the vegetation cover provided by LPJ-GUESS as opposed to prescribed vegetation cover) and of albedo and surface roughness (using newly added parameterisations). However, this information is not (thus far) used by the land surface scheme. In other words, although there is two-way information flow and calculation of land surface properties, the model is only effectively coupled in one direction. Enabling the effect of LPJ-GUESS's dynamic vegetation on the atmosphere (via the land surface scheme) is still under development and will be reported in a future publication (for preliminary results see Tost et al., 2018). The integration of LPJ-GUESS into EMAC is independent of the development of the EC-Earth and RCA-GUESS, but we believe that there are possible synergies in terms of future model development. Furthermore, we consider this parallel development to be complementary in terms of

scientific applications, in particular the representation of atmospheric chemistry processes in EMAC allows study of land-atmosphere interactions mediated by trace gas exchanges."

New references:

Smith, Benjamin, Patrick Samuelsson, Anna Wramneby, and Markku Rummukainen (2011) A Model of the Coupled Dynamics of Climate, Vegetation and Terrestrial Ecosystem Biogeochemistry for Regional Applications. Tellus A 63(1): 87–106.

Tost, Holger, Matthew Forrest, and Thomas Hickler (2018) Interactive Vegetation Influences on Climatological Meteorological Fields and Trace Gas Emissions. In P. 12047. http://adsabs.harvard.edu/abs/2018EGUGA..2012047T, accessed November 13, 2018.

p3, L11 - refers to a 'companion' publication. As no real reference is given I assume they mean 'future' publication? In my experience these sometimes never appear (even if high-priority), and if one cannot already present an author list and title that can be cited I would re-phrase.

Yes, the reviewer makes a good point. Although the work on the bidirectional coupling for the 'companion publication' is well under way (see above response including reference to Tost et al., 2018), due to unfortunate personal circumstances this work has been delayed. We therefore propose to remove all references to a 'companion publication' and instead refer to a 'future publication' and to remove 'Part 1' from the title (see new title in response to reviewer 1).

p3, L27. The phrase 'tree-individual' sounds odd and is not helpful. Re-phrase.

In saying 'tree-individual' model we wished to inform the read that individual trees are distinguished in the model but individual grasses are not. We accept that this phrasing is not helpful and instead just say 'individual' here and in the revised abstract.

p4, L22. Why 'de facto'. Aren't all components of LPJ-GUESS or EMAC de facto components?!

We apologise for the unclear phrasing. We wanted to convey that GlobFIRM was first developed and used in the distinct but related LPJ-DGVM and can be considered to be an independent model embedded in LPJ-GUESS, but which is enabled by default in most LPJ-GUESS global simulations. We propose to instead simply state:

"The GlobFIRM fire model (Thonicke et al., 2001) is included in LPJ-GUESS."

p15, L15. Shouldn't you say 'This will extend' rather than 'This extends'? If the model is already a full ESM I don't see why you are reporting on the very limited and artificial setup you have here.

The reviewer is absolutely correct, we will follow the suggestion to say 'This will extend'.

p5, L26-27. Why keep a constant CO2 when feeding a vegetation model; the seasonal cycle is well known and documented. Any why 367 ppm? Values are over 400 ppm these days.

To answer the reviewer's questions: $CO_2$ was kept constant because we are producing a steady state, rather than a transient, simulation. The value of 367 ppm was chosen to be broadly consistent with the prescribed SST of 1998. For transient simulations $CO_2$ will of course not be kept constant. Values of over 400 ppm are not relevant in this study as we are not seeking to reproduce the last few years of climate. Finally, LPJ-GUESS normally takes a single global annual value of $CO_2$, and that is the approach repeated here. However, the reviewer is correct to point the seasonal cycle of $CO_2$ with amplitude of about 5 ppm. Since $CO_2$ is provided daily from EMAC to LPJ-GUESS, future simulations can include the seasonal cycle (and also spatial variation) for increased accuracy and its effect can be quantified.

We hope that the revised text included in our response to the reviewer's point 5 adequately explains that we are performing steady state simulations and motivates our choice of $CO_2$ value.

p6. Also, isn't LPJ-GUESS a stochastic model? If so, how was the randomness of the results accounted for?

The reviewer is correct to point out that LPJ-GUESS is stochastic (although the stochasticity is generating using a pseudo random number generator with a defined starting seed to give reproducibility between model runs). Randomness was handled in the standard fashion for LPJ-GUESS; by simulating multiple patches and averaging them. We propose to extend the sentence on page 5 lines 31-32 to read:

"… and for each gridcell 50 replicate patches were simulated and averaged to account for model stochasticity."

p6. Define and give references for CRUNCEP.

Yes, this is included in our proposed text describing the offline CRUNCEP simulation.

p7, L8. The authors conclude that 'the simulations reproduce the global patterns of vegetation cover well'. There are several point here. Firstly, the Sahara is largely missing, and that is rather a big deal. Secondly, I guess these patterns are mainly determined by the 100+400 years of simulation, rather than the last 20 years, but as noted above I don't understand which climate driver was behind these 500 years.

To answer the first point, yes, we were remiss not to discuss the Sahara in the original text. We propose to discuss the under-estimation of Sahara extent in the revised text as follows:

"The extent of the Sahara Desert is also underestimated in the EMAC and *CRUNCEP* simulations, as some areas grow sufficient grass cover to be classified as short grasslands. As this is present in the *CRUNCEP* simulation, it is not related to a climate biases, but rather the over-simulation of grasses in very arid regions by LPJ-GUESS."

To answer the second, the reviewer is correct that the patterns will have been determined in the main part by the full 500 years of simulation. As this was exclusively EMAC-produced climate

variables (now made clear in our answer to point 5), we believe our interpretations and conclusions to be valid.

p7 and elsewhere. The authors sometimes say 'conservation remapping', sometimes 'conservative

We apologise for the typo and the lack of explanation. The correct term in 'conservative remapping' this has been corrected in the manuscript. We also removed the work 'bilinearly' which erroneously appeared on page 8 line 20.

To explain the method we propose to include a citation of the algorithm:

Jones, Philip W. (1999) First- and Second-Order Conservative Remapping Schemes for Grids in Spherical Coordinates. Monthly Weather Review 127(9): 2204–2210.

And a citation to the implementation:

CDO (2018) Climate Data Operators. http://www.mpimet.mpg.de/cdo.

p7, L27. Here it states 'The second source of disagreement is the climate biases in the EMAC derived climate, most obviously the underestimation of tree cover ...'. Usually one evaluates climate biases with reference to a temperature data set, not by looking at tree-lines. Again, I really miss any quantification of the EMAC errors going into this simulation, and without that I have no bases to judge the impact of LPJ-GUESS coupling.

We apologise for the unclear wording and the lack of quantification of EMAC errors. We propose to include plots of the bias in the EMAC climate to aid interpretation (see our answer to the reviewer's point 4).

Regarding the wording, we stress again that in this work it is was not our intention to comprehensively evaluate climate biases in EMAC (as the reviewer appears to have understood from our wording) but rather to examine the behaviour of LPJ-GUESS when forced with EMAC climate. In this case looking at tree lines is appropriate because we are assessing vegetation, not climate. We propose to change the sentence to:

"The second possible reason for discrepancies in modelled tree cover compared to observed tree cover is climate biases in the EMAC-produced climate. For example, this is apparent in the underestimation of tree cover on the north-east coast of South America, as already indicated in the biome plots (Fig. 1), which is clearly the result of a large negative precipitation in the region (see **New Appendix Figure 1**). "

p8, L2. The authors claim that biomass isn't directly relevant for land-atmosphere exchanges, but useful for evaluating DGVM performance. Well, canopy height is mentioned, and LAI (and hence BVOC and deposition parameters) could have been, but isn't biomass also one of the key outputs of ESMs? They are supposed to account for C-sequestration, NPP, etc. It is essential that an ESM can predict these outputs very well, but here they seem to be forgotten.

We apologise for the careless language. We fully agree that biomass is an important state variable and output in ESMs. Our statement was merely meant to reflect that from a technical perspective, total biomass (in terms of $kgC/m^2$) is not used directly by the land surface (and therefore not passed

back to EMAC), whereas the *distribution* of biomass in terms of height (ie. canopy height) and area (ie. vegetative cover) are used directly in the land surface scheme.  However, we realise that this was poorly worded and perhaps an unimportant point, and we propose to replace the sentence starting on page 8 line 2 with the following text to highlight to importance of biomass:

"Standing biomass is a key state variable in ESM and DGVMs as it is connected to productivity, carbon sequestration, evapotranspiration, vegetation cover, canopy height and other critical processes and variables which are relevant to vegetation functioning and land-atmosphere exchanges.  As such, it is a useful quantity for evaluating DGVM/ESM performance."

p11. Canopy height is here evaluated, but N-availability is a key driver for this, and here the N-deposition component is from the 1850s.

Yes, we understand that N-availability is a driver of canopy height and other vegetation properties, and that using N-deposition from the 1850s will not perfectly reconstruct 20$^{th}$ Century vegetation, it was an unfortunate oversite on our part to use these values.  However, the difference (in terms of N limitation) compared to the 'offline' CRUNCEP simulation with transient N deposition is not that large (see previous answers and **Proposed New Appendx Fig 4**). Therefore, we don't believe that this negates the point that we are trying to make with these simulations, which is that the coupled model functions as expected, reproduces the broad pattern of global vegetation and so the vegetation simulated provides a reasonable basis for an updated land surface parameterisation in EMAC.

p11, L25-29.  Here again I am not sure what to make of the paper and the setup.  For biomass inclusion of a land use correction makes the results worse, but the authors say that 'this is not a major concern .. as biomass does not directly affect the atmosphere'.

How can they say this? Biomass is directly linked to water flows, energy balances, LAI, BVOC emissions, deposition rates, canopy height, momentum exchange, vegetation extent and a host of related parameters. A failure to model biomass reflects a failure to model the vegetation.

This was a careless choice of words on our part, we simply wanted to point out that quantity 'biomass' (in terms of kgC/m$^2$) is not explicitly used by the land surface scheme (whereas quantities relative to biomass, such vegetation height and cover, are).  We propose to replace the two sentences on page 11 line 27-29 with:

"This indicates the importance of including land use effects in a consistent and realistic way in the coupled model, and that improved simulation of biomass is also critical due its status as a key state variable in the land surface representation.   In particular, the average global patch-destroying disturbance rate of 0.01 yr$^{-1}$ could be re-evaluated and rather simplistic mortality and tissue turnover rates could be further developed in LPJ-GUESS."

p11.  Again the authors suggest that LPJ-GUESS needs to be changed to perform better, but since offline simulations perform better (p7, L10), I would look to EMAC first. (I wonder if any co-authors from Lund, Potsdam or Jena would have made the same conclusions!)

We wholeheartedly agree with the reviewer that if we make statements about where LPJ-GUESS should be improved, it follows that we should support such statements with information about the performance of stand-alone LPJ-GUESS.

The inclusion of stand-alone LPJ-GUESS results makes is clear that the canopy height biases are independent of the forcing climate data used (see **Proposed Modified Figure 4**), so attempts to improve canopy height must therefore focus on the LPJ-GUESS representation of tree height, biomass and allometry.

p12. The text states that 'scores improve when moving to a higher resolution implies that .. leads to a tangible increase in model performance'. Again, I have trouble with the loose arguments. A change in GCM resolution will result in a change in GCM performance and biases. There is no need to 'imply'. The EMAC bias results should have been presented and analysed to establish problems with EMAC, and the offline LPJ-GUESS results should have been presented as the only true benchmark against which the linking can be assessed.

(This sentence was also rather circular by the way. Higher scores implies better performance? I though that that was definition of the score?)

Once again we apologise for the poor word choice and inexact meaning. We also realise that this sentence was not only circular but somewhat superfluous. We propose to replace the sentence with the following:

"While the result that the resolution of the atmospheric has such a significant effect on the vegetation is not surprising in itself, it does highlight that when considering the bidirectionally coupled model with dynamic (as opposed to prescribed) vegetation, thorough investigation must be made of the effect of the resolution of the atmospheric model on both model components, particularly considering feedbacks between them."

To answer the reviewers second point, we fully recognise this and now include both EMAC bias plots and offline LPJ-GUESS results as discussed above.

---

## Referee Report (RR1)

**Review of "Including vegetation dynamics in an atmospheric chemistry-enabled GCM: Linking LPJ-GUESS (v4.0) with EMAC modelling system (v2.53)"**

The paper describes the atmosphere-to-land coupling of EMAC to the LPJ-GUESS dynamic vegetation model. Resultant vegetation properties that will become important during the two-way coupling step are benchmarked against observations for offline and two resolutions of the newly-coupled LPJ-GUESS in order to determine if discrepancies are due to the underlying vegetation model or climate biases from EMAC. I find it quite refreshing that the authors have focussed on the one-way coupling of part of the land surface as an initial step of ESM development, rather than including the full coupling in one paper. The paper includes a substantial description of future development priorities, which helps frame this analysis in the wider ESM development. It will be interesting to see if this step-by-step approach helps aid these future developments and evaluation of the final ESM. The models' performance seems pretty impressive in the comparisons in this study, which is definitely a good sign for thing to come.

Whilst individual parts of the paper are well explained and easy to understand, the structure of the paper as a whole is quite confusing and could do with some work. I also share some of reviewers 2 concerns around attribution of model discrepancies to the vegetation model, simulated climate or model resolution, as well as the modelling protocol description. However, after going through the manuscript a number of times, I think most of these concerns could also be down to the paper structure. I also have additional questions about the benchmarking methodology which might just need clarification, but could potentially require new analysis. Due to the suggested restructuring and number of general questions, I'm afraid the review is quite long.

I notice that I seem to be replacing reviewer 2, so I start by briefly addressing responses to reviewer 2 (none of which actually need a fresh response from the authors) before some suggestions regarding paper structure, followed by general methodological comments and ending with specific suggestions to the text.

**Responses to reviewer 2.**
One of the main concerns raised by reviewer 2 was that the papers focus on natural vegetation at the exclusion of land use coupling. The reviewer included a rather odd comment, suggesting that the authors should consider land use because they are from Europe - almost as if the development of **global** models should be based on our own circumstances or immediate surroundings?! It is perfectly fine to focus on natural dynamic vegetation in this paper, whilst noting (as the authors do) that land use will need including in future developments towards a completed ESM. Dynamics in natural vegetation behaves very differently from agricultural systems, and often require separate consideration (e.g. (Burton et al., 2019)). The paper also accounts for the discrepancy between the simulation of natural vegetation and observations including land use during benchmarking in what seems an appropriate way that is consistent with previous benchmarking studies (Kelley et al., 2013). I, therefore, feel the authors have addressed this point adequately.

I do, however, share the reviewer's concern about attribution of simulated discrepancies to LPJ-GUESS model structure, EMAC climate and model resolution; as well as the description of the model protocol, including the prescription of nitrogen, even in the revised manuscript. I have incorporated these into the remainder of my review.

**Paper structure**
The paper mixes future developments into the introduce and model description and has combined benchmarking methods with results, which at times makes the manuscript hard to follow. On the first read, the paper seems very short on results and discussion, and it's only on subsequent reads that I realised the authors do actually present enough results and make some very interesting discussion points. But these are lost throughout other sections of the paper. It would be much easier to follow the paper's narrative if it was restructured into a more traditional format (i.e an introduction; methods split between model description, modelling protocol, benchmark data and benchmark metric; results and discussion including future work, and then the conclusion). Below, I've highlighted some point that could be moved to more appropriate sections of the manuscript. But I have probably missed some, and I hope the authors can identify some more in the next iteration.

***Abstract***

The parts of LPJ-GUESS that are and are not being used in this study should be explicitly separated in the abstract. Lines 5-8 list a lot of processes that, despite being important in the Earth System, are not being considered for evaluation of this study. This list of processes could be moved into the introduction. Only some processes/couplings in the description of LPJ-GUESS on lines 10-14 are switched on, and not all of those are evaluated.

The abstract is very short on benchmarking methods and results. It should describe what variables are being benchmarked, and give a brief description of attributions of model performance between LPJ-GUESS itself, climate model biases and resolution.

***Introduction***

The introduction should mention the importance of attributing biases in simulated vegetation to the vegetation model deficiencies, GCM biases or resolution effects - especially as this probably the most important piece of analysis during benchmarking.

A lot of the model description should be moved to later sections. For example:

- How climate is aggregated and passed from EMAC to LPJ-GUESS (i.e page 2, line 32 "In both modelling systems…" to end of next paragraph on page 3 line 21) is more relevant to the methods (ie current section 2.3?) and most could be moved there, although it will be worth briefly mentioning that LPJ-GUESS has been used in ESMs before. Also, I'm not entirely sure it's that relevant for this study to describe in detail how LPJ-GUESS is coupled to other GCMs (although maybe the authors disagree on this point…?), and I would suggest making only **brief** comments on past couplings when they use the same technique as the one described in this study.
- Page 3, line 31 "In addition…" to the end of the introduction could be summarized with the details moved to the discussion/future work.

***Model descriptions and protocol (section 2 and 3)***

Much of the model descriptions include a description of processes/coupling that have either not been implemented or aren't switched on and therefore not relevant for the results that follow. These are useful to give a wider context to the coupling presented here but should be moved to other parts of the text to help make it much clearer what the authors are actually using in this study. Also, some of the modelling description, coupling implementation and simulation protocol seem misplaced within the methods. For example:

- Unless the authors use fire in their results, all of the paragraph at the bottom of page 5 could be moved to a "future work" section in the discussion. Maybe the authors do use (and during revision, go on to evaluate/discuss) GlobFIRM. In which case the first sentence can be kept.
- The last bit of the 1st paragraph on page 6 (line 7 onwards) could be moved to Appendix A, as it has more to do with modification and how to run the model and isn't necessary information to evaluate the quality of the coupling or benchmarking (which the rest of the paper is dedicated too).
- The first half of the paragraph starting on line 24 on page 6 could be merged with the line 3-5 on the same page to avoid repetition.
- The "next steps" to the end of the following paragraph on page 6, line 26 to page 7, line 7 could be moved to the discussion.
- The 1st sentence of the paragraph starting line 16, page 8 should be moved to the model description (section 2.2)

***Model evaluation.***

The second half of the 2nd paragraph on page 9 feels like a discussion on poor model performance... before the results are actually presented! The NME scores actually turn out to be pretty reasonable so this slightly negative statement doesn't just seem misplaced but also takes away from the results that follow. This should, therefore, be moved to the discussion, and phrased in a slightly happier way.

Dataset descriptions and comparisons are combined for each variable. I can see the logic here, but it makes the m/s feel very jumpy, especially given the interactions between e.g. carbon to tree cover, tree cover/height to biome and biomass etc. The authors should consider putting descriptions of comparison datasets and benchmarking metrics into an earlier, more traditional methods section. That way, you can also describe how benchmark datasets are processed in one place (I'm sure I saw "conservative remapping" more than once.)

This should be followed by the comparisons, presented in an order which helps link how model errors in different variables affect one another and are affected by resolution and climate.

**Discussion and Future work and development plans**
There's no discussion section yet, but much of the text identified above could be moved into one. When rewriting, it should be made clear in the text which coupling/model developments are implemented but not yet assessed and what is still to be implemented (this shouldn't require too much effort as the authors have already demonstrated this quite nicely in Figure 1). Once this is done, development and evaluation priorities could be better linked to model deficiencies identified in the results/discussion (as has been done with things like disturbance rates etc).

**General comments**
**Model descriptions**
More description of the land surface scheme (outside of LPJ-GUESS) would be helpful. Specifically, as the paper only deals with one-way coupling, what vegetation cover/distributions does EMAC actually see in these simulations? And where are they obtained from? Are there any other, none-dynamic land surface properties that are relevant?

For the stochastic processes described, are these processes truly random? Or do they use semi-stochastic seeded random number generators? I.e if you performed the exact same simulation twice, would you get the same answer?

**Modelling protocol**
On line 18 of page 6, what is meant by "LPJ-GUESS provides fractional vegetation cover, leaf area index, daily net primary productivity and average height of each PFT to EMAC"? From Figure 1, I'm pretty sure this means that the coupling is technically implemented but not turned on for this study. But the text sounds a bit like EMAC is using information in LPJ-GUESS. The authors should make it clear what is and isn't turned on.

I am a little lost as to what the simulation actually represents? The solar forcing and $CO_2$ concentration of 367ppm suggest present day (and the authors should state which years this concentration is from). However, nitrogen deposition is from the 1850s - suggesting pre-industrial/early historic. What is the reason for the mismatch? I know the authors have said why in the responses, but a better definition of what these runs represent might help explain the mismatch in the paper. And how does Figure C1 show that there is no impact of nitrogen limitation?

What time period are the sea surface temperatures from?

On the whole, the run sounds like an equilibrium run. Was CRU-NCEP detrended to match (both for the spin-up and the final 113-year run)? And overall, what do the runs represent? An equilibrium version of the present day? Or a pragmatic spin up that could be used for further transient runs? Pragmatic is fine - we're all climate modellers and we know computer resources are too limited to run all the perfect runs we might want. But it would help when interpreting the results to better define the runs.

What resolution was the CRU-NCEP run? If it was different than the T42 and T63 runs, might this have a difference?

Was the 500-year spin up for the coupled EMAC-LPJ-GUESS, or was EMAC spun up using a separate protocol before being coupled to LPJ-GUESS? Either is a valid protocol to follow given EMAC doesn't actually see simulated vegetation properties, and it is not entirely evident from the text which is used. Either way, the spin-up protocol for the EMAC part of the model should be described. Did the 100 year period without N limitation follow an initial 500-year spin up? Or does the 100 years with N limitation + 400 years with N limitation constitute the full 500 years spin up?

How was the trend in PFT extension and height tested at the end of the spin-up? And does the inter-annual variability in vegetation refer to extension and height, or other vegetation properties as well? Was the trend in carbon pools tested?

**Null models and metric interpretation**

I can't see any reference to null models described in (Kelley et al., 2013; Kloster and Lasslop, 2017). I *think* a potential reason these have not been included is because the benchmarking is used to compare performance across models, rather than quantifying model performance itself. If this is this case, then it should be clearly stated. The 2nd paragraph in section 4, page 9 is probably a good spot to add this. However, I would point out that scores taking into account land use (in Table 2) look pretty good, and using null models may help highlight this.

As NME is basically absolute mean error, changes in scores are directly proportional to the distance away from the observations, so can be interpreted as % improvement/degradation in model performance. Maybe this could be explained when introducing NME and used when describing the scores? i.e for ree cover with LUC, T63 represents a degradation in performance of (0.69 - 0.62)/0.69 10.14% compared to CRUNCEP runs.

**Choice of comparisons**

Before introducing the datasets, the authors should justify their choices of variables for comparison, particularly why they are important to assess when coupling vegetation to a GCM.

There is no comparison of important biogeochemical earth system fluxes (i.e NPP/GPP, respiration, ET, methane, aerosols etc). The authors should justify why fluxes aren't considered or else consider adding these comparisons to their analysis. Especially as the title promises coupling to a "chemistry-enabled GCM" where fluxes will be important.

As the model should be in equilibrium given the modelling protocol (?) then the authors could also consider demonstrating the model's carbon is in equilibrium as well, i.e net ecosystem exchange is zero - a good basic test for an ESM in equilibrium runs.

From the description of biome reconstructions in section 4.1, it seems like biome comparisons are being used partially as a way of benchmarking several vegetation properties, including LAI. There are LAI products which could be used for benchmarking if LAI is an important coupling variable (Myneni et al., 2002)?

**Vegetation cover comparisons**

I know little about biome maps, but (Haxeltine and Prentice, 1996) seems a little bit old. (Oak Ridge National Laboratory, n.d.; Olson et al., 2001) maybe a bit newer? The use of (Haxeltine and Prentice, 1996) over other products should probably be justified somehow.

(Kelley et al., 2013) used the Manhattan Metrics (MM) for vegetation cover comparisons. Why was NME used instead? For two item comparisons (ie tree cover vs none-tree cover, as in this paper), I *think* scores obtained using MM and NME would be proportional to one another. If the authors can confirm this is the case (probably with a bit of maths in the response to this review), then using NME will be okay, and could be a better choice as NME is normalised by the variance around the mean of the observations, providing a more intuitive score. But the change in metric should be explained.

MODIS MODD44B Collection 6 measures woody cover of a height > 5m. Was LPJ-GUESS tree cover of less than 5m removed? If so, how? Does LPJ-GUESS use shrub PFTS? If so, were they included in tree cover comparisons?

Previous cover-based benchmark comparisons also compare herbaceous/total vegetation cover and details on leaf type and phenology (Burton et al., 2019; Kelley et al., 2013; Rabin et al., 2017). Some of these might not all be so relevant for ESM benchmarking, but the authors should either perform them or briefly justify the omission of at least total vegetation cover.

**Fire**
The manuscript describes what I *think* is the current fire model (GlobFIRM), as well as plans for future fire model development, at several points in the introduction and model description section. Yet there is no mention of fire in the results or discussion, except a brief comment about possible underestimation of fires effects on canopy height. If fire is important for any of the variables tested (e.g for veg distribution, as it is for (Bond et al., 2005; Burton et al., 2019; Hantson et al., 2016)) or for further development of ESM coupling, then surely it should be benchmarked as well? Simple burnt area and emission benchmarking is described in (Kelley et al., 2013; Rabin et al., 2017) and could be applied.

**Impact of spatial resolution**
The authors suggest at several points that degradation of performance of vegetation properties between T63 and T42 runs demonstrate poorer performance in EMAC at coarser resolutions. It's well established that changes in GCM resolution has an impact on model performance (e.g. (Kuhlbrodt et al., 2018)). However, I'm not sure the comparisons presented in this study lend any evidence to support this. It could be LPJ-GUESS performs worse at coarse resolution, and/or is sensitive to the aggregation of inputted climate information. Unless the authors can justify this statement another way (i.e driving LPJ-GUESS with CRU-NCEP gridded to the two scales of grid - which I'm sure would be much more effort than it's worth? Or making more use of the climate maps in the appendix), I would rephrase this argument to suggest the model as a whole (i.e EMAC+LPJ-GUESS) performs better at the increased spatial resolution, particularly in the first paragraph on page 17 and lines 4 and 5 on page 18.

To explore resolution effects further, I do actually think it would be interesting to see T42 in the other maps. There are clearly some big differences in biome cover in some parts of the world between model resolutions, including some interesting changes in the biomes in carbon-rich Amazon, Indonesian and South Asian forests (changes which I don't think are mentioned in the text?). It would be useful to see where these differences are coming from (including a T42 tree cover, height, and biomass maps and linking these to Figure B1-B3 climate where T42 is already plotted out).

**Figures**
Why is the background in different maps in figures 3-5 blue? Shouldn't they be white to match the part (a) figures?

There are streaks of missing data running across the Sahara and Arabian Peninsula and southern Australia what I think is in the (Avitabile et al., 2016) region of the biomass observations map. What's causing this?

**Specific comments**
Given the timing of the submission, I wonder if this model will be contributing to CMIP6. If so, the authors may wish to point this out somewhere?

Remove most of the instances of "state of the art". The authors either clearly demonstrate that the models they are coupling are the latest version, in which case the phrase is redundant, or use it when there is no extra justification as to why the model is "state of the art", in which case we are left wondering why it warrants the description.

Page 2, line 17 is the phosphorus cycle considered in EMAC or LPJ-GUESS? If so, add to the model description.

Page 2, line 26 add "being" between "actively" and "developed"

Page 4, line 23/24: "comparatively detailed". Compared to what?

Page 4, line 29: replace "phenology" with "phenological response"

Page 4, line 29: C4 is just for grasses right? If so, state.

Page 4, line 30: Are any of the woody PFTs shrubs? If so, say so. If not, don't worry.

Page 5, paragraph starting line 5: remove the information of version 3 of LPJ-GUESS. Unless version 3 is going to be used somehow (?), it is not relevant to this study. Apart from that, this is actually a very clear and concise overview of how LPJ-GUESS works!

Page 5, line 20-22 "However, monthly climate data… distinct rain events". Is this climate interpolation actually used in coupling to EMAC? Or is it used when driving with the CRU-NCEP in this study? If not, remove.

Page 7, line 6-7: remove "Whilst it's not within the scope … dependence on spatial resolution" and if necessary, merge the remainder of the sentence with the next. The paper does later test biases/resolution that affects dynamic vegetation. And it's self-evident that your aim is to test the coupling and veg dynamics, not the wider GCM.

Page 8 line 1: state what resolution or T42 and T63 are. (ie no. lat x lon. Deg at the equator or something like that). I know spectral resolutions are a little confusing in these regards, but it would be useful to give an idea of how much coarser T42 is relative to T63. And maybe discuss somewhere what applications each resolution is likely to be used for.

Page 8, line 8 remove "at no point were external climate datasets used". I think the authors mean that no external climate datasets were used in the T42 and T63 runs. But that's pretty self-evident, so stating it here sounds like at no point in this analysis was climate data used, right before describing how climate data was used.

Page 8, lines 19-22: Explain how Fig C1 shows that PI N deposition does not result in additional N limitation. And additional when compared to what?

Page 9, line 4: This paragraph starts off with a very long list of previous benchmarking of LPJ-GUESS. It should be reduced to just state that LPJ-GUESS has a long history of development and evaluation, and then pick a couple of key references. And recent ones preferably using the version of LPJ-GUESS used here (if I remember correctly, (Gerten et al., 2004) for example, was pre-GUESS?)

Page 9, line 10-11: Remove "it is beyond the scope…. dynamic vegetation model". I'm not sure it is beyond the scope to suggest changes to the dynamic vegetation model that might affect/be affected by the coupling. Especially because the authors do later describe some changes (i.e to disturbance rates, fire modules etc) that will change the dynamic vegetation model.

Page 9, line 31 replace "unity" with "1"

Page 10, line 6: The sentence "Whilst we can't expect … forced using EMAC climate" seems to be making the same point as the start of the paragraph, and should be moved/merged into somewhere around lines 3 and 4.

Page 10, line 15: The paragraph starts off a little negative. Maybe just remove everything before the first comma and start the sentence by saying "Knowledge of EMAC biases…".

Page 10, line 32: Add "Fig" before "2"

Page 11, line 24: Briefly explain conservative remapping.

Page 11, line 25: removed "as would be expected by a state-of-the-art DGVM". I'm sure I would expect some DGVMs to do a rubbish job. The fact that you have a combined model which does kind of okay is pretty impressive and shouldn't be understated.

Page 12 Please add dataset reference to figure 2 caption.

Page 15, line 22: remove "(lower is better)". It's already been described and the reader is reminded in the table caption.

Page 15, line 26, 27, sentence stating "For biomass…" states that biomass performance gets worse when accounting for LUC. I might be reading this wrong, but from Table 1, it seems the biomass results are actually getting better with the LUC modification, not worse…?

Page 15, line 31, sentence starting "In particular". The authors state that the background disturbance rate could be changed to increase vegetation carbon in the T42 run. However, as far as I can tell, this disturbance rate is also used in the CRU-NCEP run as well (?), where biomass is generally too high. This suggests the problem is actually climate biases, and that EMAC should be improved to get a better representation of biomass. The authors may be putting this forward as a pragmatic way of getting the right carbon balance within an atmosphere model which is much harder to fix. If this is the case, then please say so in the text.

Also, how might the simplistic turn over rates be developed?

Avitabile, V., Herold, M. and Heuvelink, G. B. M.: An integrated pan-tropical biomass map using multiple reference datasets, Glob. Chang. Biol. [online] Available from: https://onlinelibrary.wiley.com/doi/abs/10.1111/gcb.13139, 2016.

Bond, W. J., Woodward, F. I. and Midgley, G. F.: The global distribution of ecosystems in a world without fire, New Phytol., 165(2), 525–537, 2005.

Burton, C., Betts, R., Cardoso, M., Feldpausch, R. T., Harper, A., Jones, C. D., Kelley, D. I., Robertson, E. and Wiltshire, A.: Representation of fire, land-use change and vegetation dynamics in the Joint UK Land Environment Simulator vn4.9 (JULES), , doi:10.5194/gmd-12-179-2019, 2019.

Gerten, D., Schaphoff, S., Haberlandt, U., Lucht, W. and Sitch, S.: Terrestrial vegetation and water balance—hydrological evaluation of a dynamic global vegetation model, J. Hydrol., 286(1), 249–270, 2004.

Hantson, S., Arneth, A., Harrison, S. P., Kelley, D. I., Prentice, I. C., Rabin, S. S., Archibald, S., Mouillot, F., Arnold, S. R., Artaxo, P., Bachelet, D., Ciais, P., Forrest, M., Friedlingstein, P., Hickler, T., Kaplan, J. O., Kloster, S., Knorr, W., Laslop, G., Li, F., Melton, J. R., Meyn, A., Sitch, S., Spessa, A., van der Werf, G. R., Voulgarakis, A. and Yue, C.: The status and challenge of global fire modelling, Biogeosciences, 13(11), 3359–3375, 2016.

Haxeltine, A. and Prentice, I. C.: BIOME3: An equilibrium terrestrial biosphere model based on ecophysiological constraints, resource availability, and competition among plant functional types, Global Biogeochem. Cycles, 10(4), 693–709, 1996.

Kelley, D. I., Prentice, I. C., Harrison, S. P., Wang, H., Simard, M., Fisher, J. B., Willis, K. O. and Others: A comprehensive benchmarking system for evaluating global vegetation models, Biogeosciences, 10(5), 3313–3340, 2013.

Kloster, S. and Lasslop, G.: Historical and future fire occurrence (1850 to 2100) simulated in CMIP5 Earth System Models, Glob. Planet. Change, 150, 58–69, 2017.

Kuhlbrodt, T., Jones, C. G., Sellar, A., Storkey, D., Blockley, E., Stringer, M., Hill, R., Graham, T., Ridley, J., Blaker, A., Calvert, D., Copsey, D., Ellis, R., Hewitt, H., Hyder, P., Ineson, S., Mulcahy, J., Siahaan, A. and Walton, J.: The Low-Resolution Version of HadGEM3 GC3.1: Development and Evaluation for Global Climate, J. Adv. Model. Earth Syst., 10(11), 2865–2888, 2018.

Myneni, R. B., Hoffman, S., Knyazikhin, Y., Privette, J. L., Glassy, J., Tian, Y., Wang, Y., Song, X., Zhang, Y., Smith, G. R., Lotsch, A., Friedl, M., Morisette, J. T., Votava, P., Nemani, R. R. and Running, S. W.: Global products of vegetation leaf area and fraction absorbed PAR from year one of MODIS data, Remote Sens. Environ., 83(1), 214–231, 2002.

Oak Ridge National Laboratory: Olson's Major World Ecosystem Complexes Ranked by Carbon in Live Vegetation: An Updated Database Using the GLC2000 Land Cover Product, [online] Available from: https://cdiac.ess-dive.lbl.gov/epubs/ndp/ndp017/ndp017b.html (Accessed 26 April 2019), n.d.

Olson, D. M., Dinerstein, E., Wikramanayake, E. D., Burgess, N. D., Powell, G. V. N., Underwood, E. C., D'amico, J. A., Itoua, I., Strand, H. E., Morrison, J. C., Loucks, C. J., Allnutt, T. F., Ricketts, T. H., Kura, Y., Lamoreux, J. F., Wettengel, W. W., Hedao, P. and and Kassem, K. R.: Terrestrial Ecoregions of the World: A New Map of Life on Earth A new global map of terrestrial ecoregions provides an innovative tool for conserving biodiversity, Bioscience, 51, 933–938, 2001.

Rabin, S. S., Melton, J. R., Lasslop, G., Bachelet, D., Forrest, M., Hantson, S., Li, F., Mangeon, S., Yue, C., Arora, V. K. and Others: The Fire Modeling Intercomparison Project (FireMIP), phase 1: Experimental and analytical protocols, Geoscientific Model Development, 20, 1175–1197, 2017.

---

## Referee Report (RR2)

Apologies for the amount of time it has taken to do this review. I did warn the editors that it would take a while, but I'm not sure any of us realised quite how long.

The authors have made detailed and convincing responses to my initial review and have made extensive revisions to the m/s. I won't go over there responses unless it related to changes still required to the m/s, but I would like to point out that they really were very good. In the paper itself, the authors do a much better job of explaining the motivation of the work, the model setup and evaluation methods and how their results link to future development priorities. The introduction could still be slightly clearer on what is actual is being turned on and evaluated in the model (covered minor comments) , I am still not completely satisfied with the author response and changes with regard to scale dependency of the land surface, and I think there is a slight mismatch between some of the figures and text which might need a bit of clearing up. But these should only require slight changes to the manuscript, so I have marked this up as minor revisions.

There are also a few small corrections and specific comments.

Goodluck with future model development and (potential) CMIP7 involvement!

**Small but not quite minor comments**

***Land surface resolution dependence***

From the author's response:
*There is no inter-gridcell communication in LPJ-GUESS (ie it can be considered as a 'site model' that simulates an arbitrary list of sites) so spatial resolution does not directly affect the processes.*
You do not need inter-gridcell communication for resolution effects to be important. Many of the processes simulated by LPJ-GUESS are extremely non-linear, so simply aggregating inputs over larger scales could affect the quality of performance of LPJ-GUESS, even if using a fantastic coarse resolution model or perfect observational driving data. Infact the authors make this very point when considering likely changes in scores expected at different resolution due to homogenization of LPJ-GUESS output and observations on page 20, lines 14-16.

An example I think some of the authors might know about is fire - on fine scales you would need a model that includes fine scale processes (rate of spread etc), whereas on a coarse resolution, rate of spread becomes less important and a control based model (such as BLAZE suggested towards the end) is much easier to parameters for broadscale controls (Burton, 2019). Soils are perhaps another example, and I think somewhere buried in soil is information on the impact of averaging soils on model hydrology.

However, in the revised m/s, there are no only a couple of instances where this is a potential problem, but these should either be changed or removed. Specifically:

Page 3 line 31: The processes could also be resolution dependent.

Page 6, line 26-29: two sentences starting with "As LPJ-GUESS…" The logic here is probably wrong. But the extra resolution in the revised m/s, and climate plots in the appendix, probably provide the tools you need to attribute between climate biases and climate aggregating. And I *think* it will read okay if you add "...(i.e climate biases **or climate aggregating** )" at the end.

Page 13, line 15/16: "*indicating that this discrepancy is caused by biases in the EMAC climate at low resolution.*" If you can demonstrate this by examination of the climate plots in the appendix, then this is fine. If not, add the caveat about climate averaging again. I *think* you manage this a couple of sentences later so it might just involve some rearranging of these few sentences.

That's all I can spot. Other instances, Page 13, line 19/20 , for example, are fine as you have backed it up with previous assessment of climate simulation at different resolutions.

**Biome and tree cover bias attribution**
Specifically page 12, line 24-26.
Temperate tree cover in the EMAC simulations actually look more extensive than observations in Figure 4, though still less than the CRUNCEP simulations (Fig 4). This first sentence should be rewritten to make this clear. GPP does look slightly less than obs in some regions (Fig 3), but the difference between CRUNCEP and EMAC GPP is a lot more clear cut, so maybe use that again.

**Minor/specific comments**

Page 1 line 7: The sentence starting "The LPJ-GUESS…"   should provide a bit more detail about the processes that have been enabled and evaluated in this study. Something at the level of detail as the lines of the next sentence, which lists things not enabled.

Page 1, line 17: replace the average NME score with all three scores. Averaging across benchmarks is a bit of a controversial issue (see (Blyth et al., 2010; Kelley et al., 2013; Randerson et al., 2009)), so is best avoided seeing. There's only 3 numbers you'd need to quote so that should be fine in the abstract.

Page 2, line 10: remove "simulations"

Page 2, line 20-21:  mentioning fire and phosphorous here makes it sound like your going to include them in the model. Both are future developments though (I know GlobFIRMs in the model, but you later point out that this needs replacement). So I'd leave them out of the introduction and just provide a note about how important they are in the discussion (either future work or conclusions).

Page 3, line 1: It seems like there's an "in" missing. I.e., "has already been used **in** both a global ESM". Or maybe it's "has already been used both **in** a global ESM".  (My grammar isn't great).

Page 3, line 15: replace "both" with "all" (there are more than two ESM components and each probably claims it's own community ;) )

Page 3, line 15: The sentence beginning "When development is complete…." you should mention before this sentence  that you are just focussed on one-way coupling.

Page 3, line 17: "Then the full model will become a powerful tool", slightly more cautionary language would be better. Maybe replace "will" with "should" or "We aim for the full model to become…."

Page 5 line 20: add something like "In this study…" at the start of the paragraph, just to make it clear that you are inputting on a daily timestep for this study, rather than the monthly timestep mentioned in the following sentence

Page 5 line 22: Sentence starting "In these circumstances…" Does the disaggragator involve some stochastic implementation as well? If so, say so, and as you have an extra stochastic process, maybe move the two sentences starting "All stochastic processes ..." on line 12 to a new paragraph just before 2.1.3. If not, don't worry.

Page 6 line 6/7: Sentence starting "However in both these cases…" where does the soil moisture come from in this study? Was it simulated in LPJ-GUESS?

Page 6, line 29: Should "T63" now be replaced with "T89"?

Page 7 line 5: Should "1990-1990" be "1990-1999"?

Page 8, line 25: Figure 1 in the author responses is a good way of showing that the parts of the model being accessed here is in equilibrium -potentially quiet important to demonstrate given the short spin up time. The authors may want to consider including it as a supplement or appendix figures?

Page 8 line 16: should there be an "a" before "function"?

Page 14, line 4: refer to figure 3 and 4. Also, I'm not sure I see the overestimation of either GPP or biomass. Maybe there is some work needed on the colour bar of figures 3 and 4?

Page 14 line 22/23: "low competitiveness of grass PFTs vs tree PFTs" could also be due to other processes affecting competition, and not just fire frequency? E.g soil moisture, simplistic soil depth, drought response, PFT heretical setup, establishment rates etc. And I don't think you've offered evidence to suggest that it is caused fire frequency or why fire frequency should be singled out?

Page 14 line 26-28: I'm not sure I follow this. It sounds like NME scores in this study are comparable to scores in Kelley et al. 2013, which the text implies included LPJ-GUESS? A pre-GUESS version of LPJ was used in Kelley et al. 2013, so if I read this correctly, then the sentence needs adapting. If I didn't read this correctly, the sentence could do with some clarification.

Page 14, lines 30 - Page 19 line 3: It feels like this couple of sentences should be moved up to just after the first sentence of this section.

Page 15, Figure 3: The colour scale on the top 5 figures could be altered (i.e not linear) to make spatial patterns of GPP in EMAC maps clearer.

Page 19, line 4: The sentence starting "In summary" feels better places at the end of the section after discussion of metric scores accounting for land use, or maybe even removed as this is covered in the discussion.

Page 19, line 21: You'll have to explain a bit better how the sentence before suggests that the disturbance rate in particular needs re-evaluating.

Page 20, line 31: I think there's an "and" missing in "... fluxes), **and** to form…"

Page 21 line 14: I think it is worth briefly explaining why BLAZE was selected, rather than another model such as SPITFIRE or a re-parameterised GLOBFIRM.

Page 22 line 26: replace "will" with "should"

Page 23 line 6: "A future publication will present…" should be replaced with something like "Future development should focus on…", so as not to pre-empt what journals might publish.

Page 23 line 8 onwards: That's quite a nice way to finish.

Page 27, Figure B1: Swap the colours around so blue means more precip.

Page 29, Figure B3 caption. In not sure I understand, particularly the sentence starting "Note that these plots compare shows the radiation available..". Are all plots adjusted by 0.17? And what is meant by "adjusted"

**References**

Blyth, E., Gash, J., Lloyd, A., Pryor, M., Weedon, G. P. and Shuttleworth, J.: Evaluating the JULES Land Surface Model Energy Fluxes Using FLUXNET Data, J. Hydrometeorol., 11(2), 509–519, 2010.

Burton, C.: Impacts of fire, climate and land-use change on terrestrial ecosystems, University of Exeter, 1 April. [online] Available from:
https://ore.exeter.ac.uk/repository/handle/10871/36801 (Accessed 20 December 2019), 2019.

Kelley, D. I., Prentice, I. C., Harrison, S. P., Wang, H., Simard, M., Fisher, J. B., Willis, K. O. and Others: A comprehensive benchmarking system for evaluating global vegetation models, [online] Available from:
https://www.researchonline.mq.edu.au/vital/access/services/Download/mq:26863/DS01, 2013.

Randerson, J. T., Hoffman, F. M., Thornton, P. E., Mahowald, N. M., Lindsay, K., Lee, Y.-H., Nevison, C. D., Doney, S. C., Bonan, G., Stöckli, R. and Others: Systematic assessment of terrestrial biogeochemistry in coupled climate--carbon models, Glob. Chang. Biol., 15(10), 2462–2484, 2009.

---

## Author Response (AR2)

**Author's response to reviewer comments:**

**"Including vegetation dynamics in an atmospheric chemistry-enabled GCM: Linking LPJ-GUESS (v4.0) with EMAC modelling system (v2.53)" by**

**Matthew Forrest et al.**

We thank the reviewer for giving time to perform a very thorough review the revised version of our manuscript. Here we reproduce the reviewers' comments in full and address them in turn. As in the previous round, the reviewers' comments are in black, our responses are in blue. We include proposed alterations to the manuscript to address the reviewers concerns in green.

**Review of "Including vegetation dynamics in an atmospheric chemistry-enabled GCM: Linking LPJ-GUESS (v4.0) with EMAC modelling system (v2.53)"**

The paper describes the atmosphere-to-land coupling of EMAC to the LPJ-GUESS dynamic vegetation model. Resultant vegetation properties that will become important during the two-way coupling step are benchmarked against observations for offline and two resolutions of the newly coupled LPJ-GUESS in order to determine if discrepancies are due to the underlying vegetation model or climate biases from EMAC. I find it quite refreshing that the authors have focussed on the one-way coupling of part of the land surface as an initial step of ESM development, rather than including the full coupling in one paper. The paper includes a substantial description of future development priorities, which helps frame this analysis in the wider ESM development. It will be interesting to see if this step-by-step approach helps aid these future developments and evaluation of the final ESM. The models' performance seems pretty impressive in the comparisons in this study, which is definitely a good sign for thing to come.

Whilst individual parts of the paper are well explained and easy to understand, the structure of the paper as a whole is quite confusing and could do with some work. I also share some of reviewers 2 concerns around attribution of model discrepancies to the vegetation model, simulated climate or model resolution, as well as the modelling protocol description. However, after going through the manuscript a number of times, I think most of these concerns could also be down to the paper structure. I also have additional questions about the benchmarking methodology which might just need clarification, but could potentially require new analysis. Due to the suggested restructuring and number of general questions, I'm afraid the review is quite long.

We are happy to re-structure the manuscript as suggested and to include clarification of the methodology; we include details below. For convenience we summarise here the main changes in the revised manuscript:

- Restructuring to more standard structure including a "Results and Discussion" section
- Repeated the EMAC T42 and T63 simulations with appropriate N deposition data
- Inclusion of a higher resolution T85 simulation
- Inclusion of GPP comparisons
- Various clarifications and additional information, as requested by the reviewer

One further comment to the reviewer's use of the term 'benchmarking methodology' above. It was not our intention to rigorously 'benchmark' the coupled model exactly, but rather to perform some broad-stroke 'reality checks' (or perhaps better said 'initial evaluation'), identify the major biases (and attribute where possible) and investigate the effect spatial resolution. Indeed, we attempted to avoid the term 'benchmarking' in the manuscript, although it did creep in a few times which we have now re-phrased. Whilst this distinction is mostly semantic, it does indicate a slightly different intent.

I notice that I seem to be replacing reviewer 2, so I start by briefly addressing responses to reviewer 2 (none of which actually need a fresh response from the authors) before some suggestions regarding paper structure, followed by general methodological comments and ending with specific suggestions to the text.

**Responses to reviewer 2.**

One of the main concerns raised by reviewer 2 was that the papers focus on natural vegetation at the exclusion of land use coupling. The reviewer included a rather odd comment, suggesting that the authors should consider land use because they are from Europe - almost as if the development of **global** models should be based on our own circumstances or immediate surroundings?! It is perfectly fine to focus on natural dynamic vegetation in this paper, whilst noting (as the authors do) that land use will need including in future developments towards a completed ESM. Dynamics in natural vegetation behaves very differently from agricultural systems, and often require separate consideration (e.g. (Burton et al., 2019)). The paper also accounts for the discrepancy between the simulation of natural vegetation and observations including land use during benchmarking in what seems an appropriate way that is consistent with previous benchmarking studies (Kelley et al., 2013). I, therefore, feel the authors have addressed this point adequately.

**We appreciate the reviewer's comments on this matter.**

I do, however, share the reviewer's concern about attribution of simulated discrepancies to LPJ-GUESS model structure, EMAC climate and model resolution; as well as the description of the model protocol, including the prescription of nitrogen, even in the revised manuscript. I have incorporated these into the remainder of my review.

**Paper structure**

The paper mixes future developments into the introduce and model description and has combined benchmarking methods with results, which at times makes the manuscript hard to follow. On the first read, the paper seems very short on results and discussion, and it's only on subsequent reads that I realised the authors do actually present enough results and make some very interesting discussion points. But these are lost throughout other sections of the paper. It would be much easier to follow the paper's narrative if it was restructured into a more traditional format (i.e an introduction; methods split between model description, modelling protocol, benchmark data and benchmark metric; results and discussion including future work, and then the conclusion). Below, I've highlighted some point that could be moved to more appropriate sections of the manuscript. But I have probably missed some, and I hope the authors can identify some more in the next iteration.

Yes. The previous manuscript structure was admittedly somewhat unconventional as the paper was initially formatted along the lines of technical report, rather than a standard research paper. We have endeavoured to follow the suggestions and re-structure the manuscript appropriately.

**Abstract**

The parts of LPJ-GUESS that are and are not being used in this study should be explicitly separated in the abstract. Lines 5-8 list a lot of processes that, despite being important in the Earth System, are not being considered for evaluation of this study. This list of processes could be moved into the introduction. Only some processes/couplings in the description of LPJ-GUESS on lines 10-14 are switched on, and not all of those are evaluated.

The abstract is very short on benchmarking methods and results. It should describe what variables are being benchmarked, and give a brief description of attributions of model performance between LPJ-GUESS itself, climate model biases and resolution.

We propose the following revised abstract which we hope addresses the reviewer's concerns.

"Central to the development of Earth System Models (ESMs) has been the coupling of previously separate model types, such as ocean, atmospheric and vegetation models, to address interactive feedbacks between the system components. A modelling framework which combines a detailed representation of these components, including vegetation and other land surface processes, enables the study of land-atmosphere feedbacks under global change.

Here we present the initial steps of coupling LPJ-GUESS, a dynamic global vegetation model, to the atmospheric chemistry enabled atmosphere-ocean general circulation model EMAC. The LPJ-GUESS framework includes a comparatively detailed individual based model of vegetation dynamics including a nitrogen cycle. Although not enabled here, the model framework also includes crop and managed-land

scheme, a representation of arctic methane and permafrost, and a choice of fire models; and hence represents many important terrestrial biosphere processes and provides a wide range of prognostic trace gas emissions from vegetation, soil and fire.

We evaluated a one-way, on-line coupled model configuration (with climate variable being passed from EMAC to LPJ-GUESS but no return information flow) by conducting simulations at three spatial resolution (T42, T63 and T85). These were compared to an expert derived map of potential natural vegetation and four global gridded data products: tree cover, biomass, canopy height and gross primary productivity (GPP). We also applied a post-hoc land use correction to account for human land use. The simulations give a good description of the global potential natural vegetation distribution although there are some regional discrepancies. In particular, at the lower spatial resolutions, a combination of low cold and low-radiation biases in the growing season of the EMAC climate at high-latitudes causes an underestimation of vegetation extent.

Quantification of the agreement with the gridded datasets using the normalised mean error (NME) averaged over all datasets shows that increasing spatial resolution from T42 to T63 improved the agreement by 10%, and going from T63 to T85 improved agreement by a further 4%. The highest resolution simulation gave an average NME score of 0.67, just 4% worse agreement than an offline LPJ-GUESS simulation using observed climate data (NME = 0.64). However, it should be noted that the offline LPJ-GUESS simulation used a higher spatial resolution which makes the evaluation more rigorous, and that excluding GPP from the datasets (which was anomalously better in the EMAC simulations) gave 10% worse agreement for the EMAC simulation than the offline simulation. Gross primary productivity was best simulated by the coupled simulations, and canopy height the worst. Based on this first evaluation, we conclude that the coupled model provides a suitable means to simulate dynamic vegetation processes into EMAC."

**Introduction**

The introduction should mention the importance of attributing biases in simulated vegetation to the vegetation model deficiencies, GCM biases or resolution effects - especially as this probably the most important piece of analysis during benchmarking.

Yes, we agree. We now discuss different sources of biases in the Introduction, see the revised text below.

A lot of the model description should be moved to later sections. For example:

How climate is aggregated and passed from EMAC to LPJ-GUESS (i.e page 2, line 32 "In both modelling systems..." to end of next paragraph on page 3 line 21) is more relevant to the methods (ie current section 2.3?) and most could be moved there, although it will be worth briefly mentioning that LPJ-GUESS has been used in ESMs before. Also, I'm not entirely sure it's that relevant for this study to describe in detail how LPJ-GUESS is coupled to other GCMs (although maybe the authors disagree on this point...?), and I would suggest making only brief comments on past couplings when they use the same technique as the one described in this study.

Yes, this level of detail is excessive and was in part to answer previous reviewer comments. We have significantly shortened the discussion of the other couplings in the introduction and moved (and abbreviated) the details to the Methods section.

Page 3, line 31 "In addition..." to the end of the introduction could be summarized with the details moved to the discussion/future work.
 Yes, much of this is more appropriate for discussions/future work. We have moved many details to the new Results and Discussions section.

**The last two paragraphs of the Introduction now read:**

"By bringing together these two modelling systems, our intent is to produce a fully-featured ESM which benefits from the continuous development of both communities. We plan to follow a step-wise model integration roadmap, whereby the coupling between LPJ-GUESS and EMAC is tightened in well-defined, consecutive steps and processes (such as land use) are included or enabled in a consecutive manner. This will allow us to assess the effects of one model on the other, and the effects of the inclusion of new processes, in a step-wise and logical fashion. For our first step, we have chosen to simulate and evaluate the vegetation produced when LPJ-GUESS is forced by EMAC-simulated climate.

When evaluating the vegetation produced in this configuration, there are potentially three sources of error that may contribute to data-model mismatch: poorly constrained parameters values and inadequate representation of the processes in LPJ-GUESS; biases in the climate produced by EMAC (which are expected to have some dependency on the spatial resolution see Roeckner et al. (2006)) and missing processes in LPJ-GUESS (predominantly land use). The issue of missing land use was considered in the design of the evaluation method. To disentangle the mismatches resulting from LPJ-GUESS from those resulting EMAC, we consider a 'stand-alone' run of LPJ-GUESS in its standard configuration using observed climate data to assess LPJ-GUESS's implicit biases. To investigate resolution-dependent biases in EMAC, simulations with three spatial resolutions were performed and their performance relative to observed data compared."

**Model descriptions and protocol (section 2 and 3)**

Much of the model descriptions include a description of processes/coupling that have either not been implemented or aren't switched on and therefore not relevant for the results that follow. These are useful to give a wider context to the coupling presented here but should be moved to other parts of the text to help make it much clearer what the authors are actually using in this study. Also, some of the modelling description, coupling implementation and simulation protocol seem misplaced within the methods. For example:

 Unless the authors use fire in their results, all of the paragraph at the bottom of page 5 could be moved to a "future work" section in the discussion. Maybe the authors do use (and during revision, go on to evaluate/discuss) GlobFIRM. In which case the first sentence can be kept. GlobFIRM is enabled here (as it is a part of the 'standard' LPJ-GUESS set up) so we have retained a reference to it here (but integrated it into the previous paragraph). We now explicitly state in the Simulation Protocol section that fire is enabled in our simulations, where we also note that the output from GlobFIRM is not evaluated as it is a rather simple and (and soon to be out-dated fire model). We discuss alternative fire models in the future work section.

- The last bit of the 1st paragraph on page 6 (line 7 onwards) could be moved to Appendix A, as it has
  more to do with modification and how to run the model and isn't necessary information to evaluate
  the quality of the coupling or benchmarking (which the rest of the paper is dedicated too).
  Yes. As these lines were not important at this point in the manuscript, we have moved these lines to
  Appendix A.
- The first half of the paragraph starting on line 24 on page 6 could be merged with the line 3-5 on the same page to avoid repetition.

There may be some confusion here due our repeated use of the phrase "coupling strategy". This was clumsy on our part as we should have referred to the "model coupling strategy" (referring to the technical implementation) and the "model integration roadmap" (referring the step-wise plan to integrate the two models). We have amended the text with the new phrasing, and moved the mention on the model integration roadmap to the last section of the introduction (as it motivates the approach taken here), see our response in the Introduction comments above. We also discuss the integration roadmap further in the Further Work section.

- The "next steps" to the end of the following paragraph on page 6, line 26 to page 7, line 7 could be moved to the discussion.
   Done.
- The 1st sentence of the paragraph starting line 16, page 8 should be moved to the model description (section 2.2)
   Done.

**Model evaluation.**

The second half of the 2nd paragraph on page 9 feels like a discussion on poor model performance... before the results are actually presented! The NME scores actually turn out to be pretty reasonable so this slightly negative statement doesn't just seem misplaced but also takes away from the results that follow. This should, therefore, be moved to the discussion, and phrased in a slightly happier way.

Done. And we are of course pleased to phrase our results in a happier way.

Dataset descriptions and comparisons are combined for each variable. I can see the logic here, but it makes the m/s feel very jumpy, especially given the interactions between e.g. carbon to tree cover, tree cover/height to biome and biomass etc. The authors should consider putting descriptions of comparison datasets and benchmarking metrics into an earlier, more traditional methods section. That way, you can also describe how benchmark datasets are processed in one place (I'm sure I saw "conservative remapping" more than once.)

Yes. We have consolidated the descriptions of the datasets and processing to a new section "Methods/Evaluation data sets".

This should be followed by the comparisons, presented in an order which helps link how model errors in different variables affect one another and are affected by resolution and climate.

Yes, we have moved the description of the results and associated discussion to the new "Results and Discussion" section, consistent with the re-formatting of the manuscript.

With these changes, the former "Model evaluation" section has been entirely deleted and its contents have been split into conventional "Methods" and "Results and Discussion" section as requested.

**Discussion and Future work and development plans**

There's no discussion section yet, but much of the text identified above could be moved into one. When rewriting, it should be made clear in the text which coupling/model developments are implemented but not yet assessed and what is still to be implemented (this shouldn't require too much effort as the authors have already demonstrated this quite nicely in Figure 1). Once this is done, development and evaluation priorities could be better linked to model deficiencies identified in the results/discussion (as has been done with things like disturbance rates etc).

Yes, this is a very good and logical idea. We have endeavoured to consolidate the details of the results to the new "Results and Discussion" section, and all discussions of further developments and evaluation priorities (particularly in the context of the preceding results) to the new "Results and Discussion/Further work" section.

**General comments**

**Model descriptions**

More description of the land surface scheme (outside of LPJ-GUESS) would be helpful. Specifically, as the paper only deals with one-way coupling, what vegetation cover/distributions does EMAC actually see in these simulations? And where are they obtained from? Are there any other, none-dynamic land surface properties that are relevant?

We initially omitted the details because these relate more to the existing aspects of EMAC that we will eventually circumvent in the fully coupled model. However, we agree with the reviewer that these details might be helpful and propose to include the following text:

"As the simulations conducted here utilise only a one-way coupling, EMAC uses its standard land surface scheme which is taken from the ECHAM5 model and is described in detail by Roeckner et al. (2003). Prognostic surface and soil temperatures are calculated with a 5 layer soil model. For the hydrology component, a simple bucket model is assumed, and the water storage capacity is prescribed based on soil type data. A set of land surface data (vegetation ratio, leaf area index, forest ratio, background albedo) has been derived from a global 1 km-resolution dataset for the different horizontal resolutions of the ECHAM5 model (Hagemann, 2002). These data are used to prescribed a climatology of forest fraction (with a constant value) and of vegetation ratio and leaf area index (with a monthly temporal resolution). This prescribed land surface data is used in the model for the calculation of processes such as the interception of precipitation, the snow view in the case of snow-covered surfaces, and for evaporation (bare ground versus vegetated surfaces). Additionally, this data is used in the vertical diffusion scheme and to calculate the grid-mean surface albedo, which depends on a specified background albedo (provided as a constant input data field), a specified snow albedo (function of temperature), the area of the grid cell covered with forest, the snow cover on the ground (function of snow depth and slope of terrain) and the snow cover on the canopy (Roesch et al., 2001)."

For the stochastic processes described, are these processes truly random? Or do they use semistochastic seeded random number generators? I.e if you performed the exact same simulation twice, would you get the same answer?

These processes are semi-stochastic and we use the same seed for every simulation. During development we carefully verified that we get binary identical results from different runs with the same settings (including the case where the model is restarted and when it is not). We add the following text to the LPJ-GUESS model description section to clarify this:

"All stochastic processes are implemented 'semi-stochastically' using a random number generator with a starting seed. This means that for a fixed starting seed, model runs with the identical settings produce identical results."

**Modelling protocol**

On line 18 of page 6, what is meant by "LPJ-GUESS provides fractional vegetation cover, leaf area index, daily net primary productivity and average height of each PFT to EMAC"? From Figure 1, I'm pretty sure this means that the coupling is technically implemented but not turned on for this study. But the text sounds a bit like EMAC is using information in LPJ-GUESS. The authors should make it clear what is and isn't turned on.

Yes, the reviewer's interpretation is correct and can see the misunderstanding. We propose to simply expand the sentence to read:

"LPJ-GUESS provides fractional vegetation cover, leaf area index, daily net primary productivity and average height of each PFT to EMAC, but these values are not used by EMAC in the simulations presented here."

We also move the following sentence ("Parameterisations for determining albedo and roughness length are implemented in EMAC, however they are not enabled in the simulations presented here.") to the "Results and Discussion/Future Work" section where it is more appropriate.

I am a little lost as to what the simulation actually represents? The solar forcing and CO2 concentration of 367ppm suggest present day (and the authors should state which years this concentration is from). However, nitrogen deposition is from the 1850s - suggesting preindustrial/early historic. What is the reason for the mismatch? I know the authors have said why in the responses, but a better definition of what these runs represent might help explain the mismatch in the paper. And how does Figure C1 show that there is no impact of nitrogen limitation?

The atmospheric CO2 corresponds to approximately 1999. The mismatch with regard to N deposition data was purely due to miscommunication during the development. We have now repeated the simulations with appropriate N deposition data (from 1990-1999), rendering the figure C1 and discussions of consequences of the N deposition mismatch entirely moot. These have been removed from the revised version of the manuscript.

**What time period are the sea surface temperatures from?**

The SSTs (and SICs = sea ice coverage) are climatological values from the AMPI2 database, i.e. the mean monthly resolved SST from the years 1995-2000; they include the annual cycle, but do not represent any specific year (neither a special El Nino or La Nina event). This information is now included in the "Methods/Simulation setup" section.

On the whole, the run sounds like an equilibrium run. Was CRU-NCEP detrended to match (both for the spinup and the final 113-year run)? And overall, what do the runs represent? An equilibrium version of the present day? Or a pragmatic spin up that could be used for further transient runs? Pragmatic is fine - we're all climate modellers and we know computer resources are too limited to run all the perfect runs we might want. But it would help when interpreting the results to better define the runs.

Yes, these are an equilibrium runs representing approximately the period from late 1990s to the early 2000s, corresponding roughly to the period in which the satellite observations were made. Unfortunately, these runs are unlikely to be utilised in further studies as the saved model state (for both LPJ-GUESS and EMAC) will not be compatible with future model versions.

What resolution was the CRU-NCEP run? If it was different than the T42 and T63 runs, might this have a difference?

The CRUNCEP was run at 0.5 degrees, that default spatial resolution for LPJ-GUESS. It was remiss of us not to include that detail and now include it in the "Methods/Simulation setup" section.

There is no inter-gridcell communication in LPJ-GUESS (ie it can be considered as a 'site model' that simulates an arbitrary list of sites) so spatial resolution does not directly affect the processes. The results are therefore only directly sensitive to the input data at an individual gridcell (site) and therefore also to whatever method was used to re-grid/interpolate it.

Was the 500-year spin up for the coupled EMAC-LPJ-GUESS, or was EMAC spun up using a separate protocol before being coupled to LPJ-GUESS? Either is a valid protocol to follow given EMAC doesn't actually see simulated vegetation properties, and it is not entirely evident from the text which is used. Either way, the spin-up protocol for the EMAC part of the model should be described. Did the 100 year period without N limitation follow an initial 500-year spin up? Or does the 100 years with N limitation + 400 years with N limitation constitute the full 500 years spin up?

LPJ-GUESS and EMAC are coupled throughout the spin-up. And the total spin-up is 500 years: 100 without N limitation followed by 400 years with N limitation (i.e. there is no initial 500 year spinup). We have re-written the last paragraph of the "Simulation setup" section to better explain this as follows:

"In all model simulations a 500 years spin-up phase was used to allow the LPJ-GUESS vegetation to reach approximate equilibrium. The coupled simulations used the online EMAC climate during spin-up, and the CRUNCEP simulations the first 30 years (1901-1930) of the CRUNCEP dataset which were detrended and repeated. Simulations followed the standard LPJ-GUESS procedure of starting with 'bare ground', ie. no vegetation and no C or N in the soil and litter pools. Having no plant available N present in the soil at the start of the simulation would inhibit and distort vegetation growth if N limitation was enabled. To overcome this, we followed the standard protocol, which is to run LPJ-GUESS for 100 years without N limitation but with normal N deposition to build up the N pools. After 100 years there is sufficient N in the pools, but the vegetation is inconsistent with the desired state as it has been growing without N limitation. Therefore, the vegetation is removed (and the C and N put into the litter pools), and the vegetation is allowed to regrow, this time with N limitation enabled, for a further 400 years. At that time, no significant trends in PFT extension and PFT height were obvious, but the vegetation shows interannual variability as expected.

For the T42, T63 and T85 simulations, an additional 50 years were simulated which were averaged to produce the plots shown here. In the CRUNCEP simulation, a further 113 years (1901-2013) were simulated using full CRUNCEP transient time series. The plots presented here show CRUNCEP output averaged over the years 1981-2010." How was the trend in PFT extension and height tested at the end of the spin-up? And does the interannual variability in vegetation refer to extension and height, or other vegetation properties as well? Was the trend in carbon pools tested?

They were examined through simple visual inspection. Since LPJ-GUESS features stochastic processes (establishment of vegetation, mortality and disturbance) and the EMAC climate has internal variability, there will interannual variability in all properties of the vegetation.

To satisfy the reviewer's curiosity, we include time series plots of the height and coverage of the PFTs, and the C pools for the last 200 years of the simulations below in Response Figure 1.

---

## Author Response (AR3)

Author's response to reviewer comments:

**"Including vegetation dynamics in an atmospheric chemistry-enabled GCM: Linking LPJ-GUESS (v4.0) with EMAC modelling system (v2.53)"**

by

Matthew Forrest et al.

(Third round of revisions)

Once again, we thank the reviewer taking the time to review the revised version of our manuscript.   As before we reproduce the reviewers' comments in full and address them in turn, the reviewers' comments are in black, our responses are in blue.   We include proposed alterations to the manuscript to address the reviewers concerns in green.

Apologies for the amount of time it has taken to do this review. I did warn the editors that it would take a while, but I'm not sure any of us realised quite how long.
The authors have made detailed and convincing responses to my initial review and have made extensive revisions to the m/s. I won't go over there responses unless it related to changes still required to the m/s, but I would like to point out that they really were very good. In the paper itself, the authors do a much better job of explaining the motivation of the work, the model setup and evaluation methods and how their results link to future development priorities. The introduction could still be slightly clearer on what is actual is being turned on and evaluated in the model (covered minor comments) , I am still not completely satisfied with the author response and changes with regard to scale dependency of the land surface, and I think there is a slight mismatch between some of the figures and text which might need a bit of clearing up. But these should only require slight changes to the manuscript, so I have marked this up as minor revisions.

There are also a few small corrections and specific comments.
Goodluck with future model development and (potential) CMIP7 involvement!

We are pleased that the reviewer is generally content with the revisions and will address the outstanding points below.

**Small but not quite minor comments**
***Land surface resolution dependence***
From the author's response:
*There is no inter-gridcell communication in LPJ-GUESS (ie it can be considered as a 'site model' that simulates an arbitrary list of sites) so spatial resolution does not directly affect the processes.*

You do not need inter-gridcell communication for resolution effects to be important. Many of the processes simulated by LPJ-GUESS are extremely non-linear, so simply aggregating inputs over larger scales could affect the quality of performance of LPJ-GUESS, even if using a fantastic coarse resolution model or perfect observational driving data. In fact the authors make this very point when considering likely changes in scores expected at different resolution due to homogenization of LPJ-GUESS output and observations on page 20, lines 14-16.

An example I think some of the authors might know about is fire - on fine scales you would need a model that includes fine scale processes (rate of spread etc), whereas on a coarse resolution, rate of spread becomes less important and a control based model (such as BLAZE suggested towards the end) is much easier to parameters for broadscale controls (Burton, 2019). Soils are perhaps another example, and I think somewhere buried in soil is information on the impact of averaging soils on model hydrology.

Understood (finally!). Although it could be argued that spatial resolution affects the processes *indirectly* via its effects on the input data rather than processes themselves. But yes, the point is most definitely taken.

However, in the revised m/s, there are no only a couple of instances where this is a potential problem, but these should either be changed or removed. Specifically:
Page 3 line 31: The processes could also be resolution dependent.

Here we assume the reviewer is referring to the processes missing from LPJ-GUESS. This is potentially true, but as these processes are missing from the model, their potential resolution-dependence does not affect the analysis here, so we see no need to modify the text.

Page 6, line 26-29: two sentences starting with "As LPJ-GUESS…" The logic here is probably wrong. But the extra resolution in the revised m/s, and climate plots in the appendix, probably provide the tools you need to attribute between climate biases and climate aggregating. And I *think* it will read okay if you add "...(i.e climate biases **or climate aggregating** )" at the end.

Here we propose to change the text highlighted by the reviewer to:

"As LPJ-GUESS has no inter-gridcell interactions and no processes are gridcell size/spacing dependent, it has no direct sensitivity to the spatial resolution at which it is run. However, in the coupled setup, LPJ-GUESS will be sensitive to spatial resolution via the climate data received from EMAC. Thus, the changes in the vegetation produced by the EMAC-coupled simulations at different resolution can only be due to changes in the EMAC produced climate (i.e. altered climate biases or climate aggregating). "

Page 13, line 15/16: "*indicating that this discrepancy is caused by biases in the EMAC climate at low resolution.*" If you can demonstrate this by examination of the climate plots in the appendix, then this is fine. If not, add the caveat about climate averaging again. I *think* you manage this a couple of sentences later so it might just involve some rearranging of these few sentences.

Yes indeed, the sentence later explains with is terms of climate biases.  We have re-arranged the text so that it now reads:

"This is most apparent for the lowest resolution (T42) EMAC simulation but improves with increasing spatial resolution, with the T63 simulation being better substantially than T42. The EMAC simulation with the highest spatial resolution (T85) showed only a small tendency to underestimate high latitude vegetation, to a similar degree as the offline CRUNCEP simulation. Examination of the climate bias plots for temperature and radiation (Figs B2 and B3) reveals a high-latitude growing season low temperature bias and low plant available radiation bias at low resolution. This was somewhat mitigated at higher resolution as would be expected due to a better representation of the synoptic scale systems in T63 and T85 (Roeckner et al., 2006). Correspondingly, the GPP simulated in this area (Fig. 3) confirms this by revealing a broad tendency to underestimate GPP above 50∘ N in the T42 simulation. This tendency lessens at higher resolution and is not seen in the offline CRUNCEP simulation. The consequences of this high-latitude underestimation of productivity at lower resolutions are also visible when comparing to observed tree cover (Fig. 4), biomass (Fig. 5) and canopy height (Fig. 6), showing that this issue affected both forested and non-forested vegetation types."

That's all I can spot. Other instances, Page 13, line 19/20 , for example, are fine as you have backed it up with previous assessment of climate simulation at different resolutions.

***Biome and tree cover bias attribution***
Specifically page 12, line 24-26.
Temperate tree cover in the EMAC simulations actually look more extensive than observations in Figure 4, though still less than the CRUNCEP simulations (Fig 4). This first sentence should be rewritten to make this clear. GPP does look slightly less than obs in some regions (Fig 3), but the difference between CRUNCEP and EMAC GPP is a lot more clear cut, so maybe use that again.
The point here was to discuss temperate forest vegetation zone (in terms of potential natural vegetation) rather than tree cover (where the observations include extensive deforestation), so our reference to Fig 4 in that first sentence was slightly misleading and wasn't meant to imply comparison to the tree cover observation.  We have removed it.  We have also added the following sentence to correctly discuss tree cover in this context:
"This underestimation led to reduced tree cover in the EMAC simulations compared to the *CRUNCEP* simulation (Fig. 4) and hence reduced temperate forest extent."

**Minor/specific comments**
Page 1 line 7: The sentence starting "The LPJ-GUESS…" should provide a bit more detail about the processes that have been enabled and evaluated in this study. Something at the level of detail as the lines of the next sentence, which lists things not enabled.
Yes, this is helpful information to provide in the abstract.  The text now reads:
"The LPJ-GUESS framework is based on ecosphysiological processes, such as photosynthesis, plant and soil respiration, ecosystem carbon, nitrogen and water cycling and includes a comparatively detailed individual-based representation of resource competition, plant growth and vegetation dynamics as well as fire disturbance. Although not enabled here, the

model framework also includes crop and managed-land scheme, a representation of arctic methane and permafrost, and a choice of fire models; and hence represents…"

Page 1, line 17: replace the average NME score with all three scores. Averaging across benchmarks is a bit of a controversial issue (see (Blyth et al., 2010; Kelley et al., 2013; Randerson et al., 2009)), so is best avoided seeing. There's only 3 numbers you'd need to quote so that should be fine in the abstract.
The given references don't appear to argue against naively averaging *normalised* benchmarking metrics. However, there is no reason not to quote all the numbers as suggested so that is done in the revised manuscript. The new sentences in the abstract now read:
"The highest resolution simulation gave NME scores of 0.63, 0.66, 0.84 and 0.53 for tree cover, biomass, canopy height and GPP respectively (after correcting tree cover and biomass for human-caused deforestation which was not present in the simulations). These scores are just 4% worse on average than an offline LPJ-GUESS simulation using observed climate data and corrected for deforestation by the same method."

Page 2, line 10: remove "simulations"
Done.

Page 2, line 20-21: mentioning fire and phosphorous here makes it sound like your going to include them in the model. Both are future developments though (I know GlobFIRMs in the model, but you later point out that this needs replacement). So I'd leave them out of the introduction and just provide a note about how important they are in the discussion (either future work or conclusions).
We have removed those mention sof fire and phosphorous. Fire is already discussed explicitly in the future work section and for phosphorous we have added the follow senrence to the future work section (end of third paragraph):
"A potential longer-term aim is to include a representation of the phosphorus cycle which strongly limits terrestrial productivity (Elser et al.,2007) and is currently in development for LPJ-GUESS."

New reference:
Elser, J. J., Bracken, M. E. S., Cleland, E. E., Gruner, D. S., Harpole, W. S., Hillebrand, H., Ngai, J. T., Seabloom, E. W., Shurin, J. B., and Smith, J. E.: Global analysis of nitrogen and phosphorus limitation of primary producers in freshwater, marine and terrestrial ecosystems, Ecology Letters, 10, 1135–1142, https://doi.org/10.1111/j.1461-0248.2007.01113.x, https://onlinelibrary.wiley.com/doi/abs/10.1111/j.1461-0248.2007.01113.x, 2007.

Page 3, line 1: It seems like there's an "in" missing. I.e., "has already been used **in** both a global ESM". Or maybe it's "has already been used both **in** a global ESM". (My grammar isn't great).
Yes, we changed it to the first variant (although I think both are correct).

Page 3, line 15: replace "both" with "all" (there are more than two ESM components and each probably claims it's own community ;) )

Yes, in fact we changed it to "continuous development of all components.", to avoid the discussion of 'communities' entirely.

Page 3, line 15: The sentence beginning "When development is complete…." you should mention before this sentence that you are just focussed on one-way coupling.
Yes, for additional clarity we have re-worked the text by bringing some later text to before this and modified the "When development is complete" sentence to read:
"By bringing together these two modelling systems, our intent is to produce a fully-featured ESM which benefits from the continuous development of all components. We plan to follow a step-wise model integration roadmap, whereby the coupling between LPJ-GUESS and EMAC is tightened in well-defined, consecutive steps and processes (such as land use) are included or enabled in a consecutive manner. This will allow us to assess the effects of one model on the other, and the effects of the inclusion of new processes, in a step-wise and logical fashion. For our first step, we have chosen to simulate and evaluate the vegetation produced when LPJ-GUESS is forced by EMAC-simulated climate, ie. a one-way coupling without the feedback from the land surface to the atmosphere.

Upon completion of the full model integration process (including bidirectional coupling which is not presented here), the trace gas emissions from LPJ-GUESS will form key inputs to the atmospheric chemistry representations in EMAC allowing…"

Page 3, line 17: "Then the full model will become a powerful tool", slightly more cautionary language would be better. Maybe replace "will" with "should" or "We aim for the full model to become…."
Yes, we replaced "will" with "should"

Page 5 line 20: add something like "In this study…" at the start of the paragraph, just to make it clear that you are inputting on a daily timestep for this study, rather than the monthly timestep mentioned in the following sentence
Since LPJ-GUESS *always* works on a daily time step for these processes we prefer not to make this alteration as it is actually misleading.  The following sentence refers to input data (not process time-step).

Page 5 line 22: Sentence starting "In these circumstances…" Does the disaggragator involve some stochastic implementation as well? If so, say so, and as you have an extra stochastic process, maybe move the two sentences starting "All stochastic processes ..." on line 12 to a new paragraph just before 2.1.3. If not, don't worry.
Yes, it does have a stochastic component so we have moved the text as suggested.

Page 6 line 6/7: Sentence starting "However in both these cases…" where does the soil moisture come from in this study? Was it simulated in LPJ-GUESS?
Yes, it was simulated in LPJ-GUESS.  We changed the text to read:
"However in both these cases, daily soil moisture from the land surface model was also used to drive LPJ-GUESS (in this implementation LPJ-GUESS's internally calculated soil moisture was used)."

Page 6, line 29: Should "T63" now be replaced with "T89"?
The definition of "finer" spatial resolution is admittedly somewhat arbitrary but since T63 does show improvements compared to T42 we believe that the sentence is still factually correct so prefer to keep it as is.

Page 7 line 5: Should "1990-1990" be "1990-1999"?
Yes, changed.

Page 8, line 25: Figure 1 in the author responses is a good way of showing that the parts of the model being accessed here is in equilibrium -potentially quiet important to demonstrate given the short spin up time. The authors may want to consider including it as a supplement or appendix figures?
Yes, this is a nice idea.  These plots are now included as an additional appendix with the additional text:
"The net ecosystem change plots shown in Figure B1 display no systematic variation from zero in either space or time indicating
that the vegetation from LPJ-GUESS is in equilibrium with the climate from EMAC. The small variations from zero that are
visible are due to the stochastic processes in LPJ-GUESS and internal climate variability in EMAC."

Page 8 line 16: should there be an "a" before "function"?
Yes, corrected.

Page 14, line 4: refer to figure 3 and 4. Also, I'm not sure I see the overestimation of either GPP or biomass. Maybe there is some work needed on the colour bar of figures 3 and 4?
References added.  Regarding the colour bars, this is more obvious in the original plots, it would appear some clarity is lost when they are included in the .pdf.  We will ensure they are correctly rendered in the final document at the proofing stage.

Page 14 line 22/23: "low competitiveness of grass PFTs vs tree PFTs" could also be due to other processes affecting competition, and not just fire frequency? E.g soil moisture, simplistic soil depth, drought response, PFT heretical setup, establishment rates etc. And I don't think you've offered evidence to suggest that it is caused fire frequency or why fire frequency should be singled out?
Yes, as no evidence is offered this is entirely speculative so we will simply remove the text in the parentheses referring specifically to fire frequency.

Page 14 line 26-28: I'm not sure I follow this. It sounds like NME scores in this study are comparable to scores in Kelley et al. 2013, which the text implies included LPJ-GUESS? A pre-GUESS version of LPJ was used in Kelley et al. 2013, so if I read this correctly, then the sentence needs adapting. If I didn't read this correctly, the sentence could do with some clarification.
In Kelley et al. 2013 a different model from the LPJ 'family' was used, and from the sentence structure we can see how confusion could arise.  We propose to replace the test "including LPJ-GUESS (Table 1)." with the following sentence text:
"and the offline *CRUNCEP* LPJ-GUESS simulations performed in this study (Table 1)"

Page 14, lines 30 - Page 19 line 3: It feels like this couple of sentences should be moved up to just after the first sentence of this section.

As this text is in the combined Results and Discussion section, we prefer to leave the ordering as is, i.e. first introducing the results and then discussing their context and caveats.

Page 15, Figure 3: The colour scale on the top 5 figures could be altered (i.e not linear) to make spatial patterns of GPP in EMAC maps clearer.

Yes, we have logged the scale and reversed the colour palette which improves definition a little bit and include this revised figure in the manuscript:

[Figure]

We have also added the following text to figure caption:

"In the upper panel the colour scale in has been log-transformed and grey areas denote GPP values less than 5 gC m$^{-2}$."

Page 19, line 4: The sentence starting "In summary" feels better places at the end of the section after discussion of metric scores accounting for land use, or maybe even removed as this is covered in the discussion.

Yes, we have moved this to the end of the following section after the metric scores accounting for land use have been discussed.

Page 19, line 21: You'll have to explain a bit better how the sentence before suggests that the disturbance rate in particular needs re-evaluating.

Yes, we have modified the sentence to read:

As LPJ-GUESS biomass has been shown to be sensitive to disturbance rates (Hickler et al. 2004, Pugh et al. 2019.), the average global patch-destroying disturbance rate of 0.01 yr−1 could be re-evaluated and the rather simplistic mortality could be further developed in LPJ-GUESS.

References:

Hickler, T., Smith, B., Sykes, M. T., Davis, M. B., Sugita, S., and Walker, K.: Using a Generalized Vegetation Model to Simulate Vegetation Dynamics in Northeastern Usa, Ecology, 85, 519–530, https://doi.org/10.1890/02-0344, http://onlinelibrary.wiley.com/doi/10.1890/02-0344/abstract, 2004.

Pugh, T. A. M., Arneth, A., Kautz, M., Poulter, B., and Smith, B.: Important role of forest disturbances in the global biomass turnover and carbon sinks, Nature Geoscience, 12, 730–735, https://doi.org/10.1038/s41561-019-0427-2, https://www.nature.com/articles/s41561-019-0427-2, 2019.

Page 20, line 31: I think there's an "and" missing in "... fluxes), **and** to form…"
Grammatically speaking the is no need for an "and" there as that would imply the three clauses are a list but they are not intended to be. In fact, a better modification to the sentence is to remove both the commas, which we have done.

Page 21 line 14: I think it is worth briefly explaining why BLAZE was selected, rather than another model such as SPITFIRE or a re-parameterised GLOBFIRM.
This decision was not ours to make, and (although we could speculate) we feel it would be more appropriate that the responsible people discuss their reasoning in a publication focussing on the implementation of BLAZE within LPJ-GUESS.

Page 22 line 26: replace "will" with "should"
Yes, see below.

Page 23 line 6: "A future publication will present…" should be replaced with something like "Future development should focus on…", so as not to pre-empt what journals might publish.
Yes. Sentence changed to,
"Future development should focus on completing the two-way model coupling and investigate the effects of the atmosphere."

Page 23 line 8 onwards: That's quite a nice way to finish.
Thank you.

Page 27, Figure B1: Swap the colours around so blue means more precip.
Whilst we can understand the logic here for precipitation, in all other difference plots in the paper red means more and blue means less, so we prefer to keep consistency within the manuscript.

Page 29, Figure B3 caption. In not sure I understand, particularly the sentence starting "Note that these plots compare shows the radiation available..". Are all plots adjusted by 0.17? And what is meant by "adjusted"
Yes, the wording here is not clear. Only the CRUNCEP offline radiation values were adjusted by 0.17. And adjusted simply mean "apply albedo value". We have rephrased to:
 "The CRUNCEP gross shortwave flux has had the standard LPJ albedo value of 0.17 applied (temporally and spatially invariant), and the EMAC gross shortwave flux has had the spatially and temporally varying albedo values in the land surface scheme applied."

**References**

Blyth, E., Gash, J., Lloyd, A., Pryor, M., Weedon, G. P. and Shuttleworth, J.: Evaluating the JULES Land Surface Model Energy Fluxes Using FLUXNET Data, J. Hydrometeorol., 11(2), 509–519, 2010.

Burton, C.: Impacts of fire, climate and land-use change on terrestrial ecosystems, University of Exeter, 1 April. [online] Available from: https://ore.exeter.ac.uk/repository/handle/10871/36801 (Accessed 20 December 2019), 2019.

Kelley, D. I., Prentice, I. C., Harrison, S. P., Wang, H., Simard, M., Fisher, J. B., Willis, K. O. and Others: A comprehensive benchmarking system for evaluating global vegetation models, [online] Available from: https://www.researchonline.mq.edu.au/vital/access/services/Download/mq:26863/DS01, 2013.

Randerson, J. T., Hoffman, F. M., Thornton, P. E., Mahowald, N. M., Lindsay, K., Lee, Y.-H., Nevison, C. D., Doney, S. C., Bonan, G., Stöckli, R. and Others: Systematic assessment of terrestrial biogeochemistry in coupled climate--carbon models, Glob. Chang. Biol., 15(10), 2462–2484, 2009.